# Extremal Domain Translation with Neural Optimal Transport

**Milena Gazdieva**\*
Skolkovo Institute of Science and Technology
Moscow, Russia
`milena.gazdieva@skoltech.ru`

**Alexander Korotin**\*
Skolkovo Institute of Science and Technology
Artificial Intelligence Research Institute
Moscow, Russia
`a.korotin@skoltech.ru`

**Daniil Selikhanovych**
Skolkovo Institute of Science and Technology
Moscow, Russia
`selikhanovychdaniil@gmail.com`

**Evgeny Burnaev**
Skolkovo Institute of Science and Technology
Artificial Intelligence Research Institute
Moscow, Russia
`e.burnaev@skoltech.ru`

## Abstract

In many *unpaired* image domain translation problems, e.g., style transfer or super-resolution, it is important to keep the translated image similar to its respective input image. We propose the extremal transport (ET) which is a mathematical formalization of the theoretically best possible unpaired translation between a pair of domains w.r.t. the given similarity function. Inspired by the recent advances in neural optimal transport (OT), we propose a scalable algorithm to approximate ET maps as a limit of partial OT maps. We test our algorithm on toy examples and on the unpaired image-to-image translation task. The code is publicly available at

`https://github.com/milenagazdieva/ExtremalNeuralOptimalTransport`

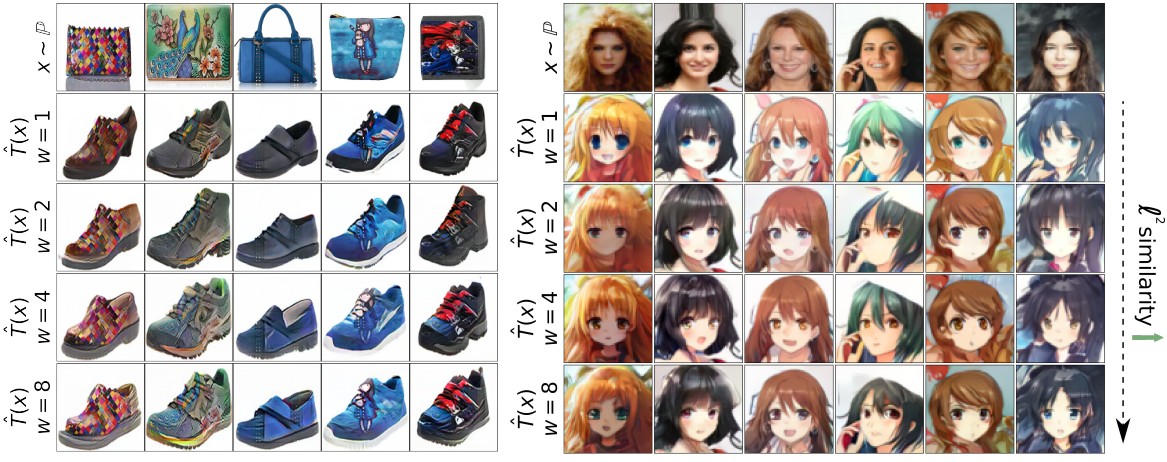

(a) *Handbag → shoes* (128×128).

(b) *Celeba* (female) *→ anime* (64×64).

Figure 1: (Nearly) extremal transport with our Algorithm 1.
Higher $w$ yields bigger similarity of $x$ and $T(x)$ in $\ell^2$.

## 1 Introduction

The **unpaired** translation task [72, Fig. 2] is to find a map $x \mapsto T(x)$, usually a neural network, which transports the samples $x$ from the given source domain to the target domain. The key challenge

---

\*Equal contribution

37th Conference on Neural Information Processing Systems (NeurIPS 2023).

here is that the **correspondence** between available data samples $x$ from the source and $y$ from target domains **is not given**. Thus, the task is ambiguous as there might exist multiple suitable $T$.

When solving the task, many methods regularize the translated samples $T(x)$ to inherit specific attributes of the respective input samples $x$. In the popular unpaired translation [72, Fig. 9] and enhancement [69, Equation 3] tasks for images, it is common to use additional *unsupervised* identity losses, e.g., $\|T(x) - x\|_1$, to make the translated output $T(x)$ be similar to the input images $x$. The same applies, e.g., to audio translation [51]. Therefore, the learning objectives of such methods usually have two components.

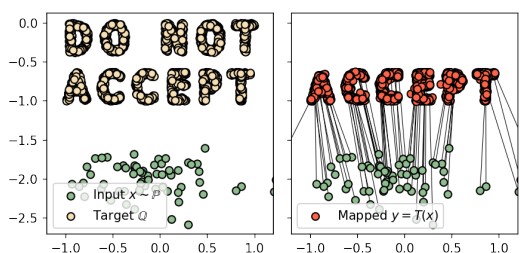

Figure 2: Learned transport map ($w = 2$) in *'Accept'* task.

The first component is the **domain loss** (main) enforcing the translated sample $T(x)$ to look like the samples $y$ from the target domain. The second component is the **similarity loss** (regularizer, *optional*) stimulating the translated $T(x)$ to inherit certain attributes of input $x$. A question arises: can one obtain the **maximal** similarity of $T(x)$ to $x$ but still ensure that $T(x)$ is indeed from the target domain? A straightforward "*yes, just increase the weight of the similarity loss*" may work but only to a limited extent. We demonstrate this in Appendix C.

**Contributions.** In this paper, we propose the *extremal transport* (ET, §3.1) which is a rigorous mathematical task formulation describing the theoretically best possible unpaired domain translation w.r.t. the given similarity function. We explicitly characterize ET maps and plans by establishing an intuitive connection to the nearest neighbors (NN). We show that ET maps can be learned as a limit (§3.3) of specific partial optimal transport (OT) problem which we call *incomplete transport* (IT, §3.2). For IT, we derive the duality formula yielding an efficient computational algorithm (§3.4). We test our algorithm on toy 2D examples and high-dimensional unpaired image translation (§5).

**Notation.** We consider *compact* Polish spaces $(\mathcal{X}, \|\cdot\|_{\mathcal{X}})$, $(\mathcal{Y}, \|\cdot\|_{\mathcal{Y}})$ and use $\mathcal{P}(\mathcal{X}), \mathcal{P}(\mathcal{Y})$ to denote the sets of Radon probability measures on them. We use $\mathcal{M}_+(\mathcal{X}) \subset \mathcal{M}(\mathcal{X})$ to denote the sets of finite non-negative and finite signed (Radon) measures on $\mathcal{X}$, respectively. They both contain $\mathcal{P}(\mathcal{X})$ as a subset. For a non-negative $\mu \in \mathcal{M}_+(\mathcal{X})$, its support is denoted by $\text{Supp}(\mu) \subset \mathcal{X}$. It is a closed set consisting of all points $x \in \mathcal{X}$ for which every open neighbourhood $A \ni x$ satisfies $\mu(A) > 0$. We use $\mathcal{C}(\mathcal{X})$ to denote the set of continuous functions $\mathcal{X} \to \mathbb{R}$ equipped with $\|\cdot\|_\infty$ norm. Its dual space is $\mathcal{M}(\mathcal{X})$ equipped with the $\|\cdot\|_1$ norm. A sequence $\mu_1, \mu_2, \cdots \in \mathcal{M}(\mathcal{X})$ is said to be weakly-* converging to $\mu^* \in \mathcal{M}(\mathcal{X})$ if for every $f \in \mathcal{C}(\mathcal{X})$ it holds that $\lim_{n\to\infty} \int_{\mathcal{X}} f(x) d\mu_n(x) = \int_{\mathcal{X}} f(x) d\mu^*(x)$. For a probability measure $\pi \in \mathcal{P}(\mathcal{X} \times \mathcal{Y})$, we use $\pi_x \in \mathcal{P}(\mathcal{X})$ and $\pi_y \in \mathcal{P}(\mathcal{Y})$ to denote its projections onto $\mathcal{X}, \mathcal{Y}$, respectively. Disintegration of $\pi$ yields $d\pi(x, y) = d\pi_x(x) d\pi(y|x)$, where $\pi(y|x)$ denotes the conditional distribution of $y \in \mathcal{Y}$ for a given $x \in \mathcal{X}$. For $\mu, \nu \in \mathcal{M}(\mathcal{Y})$, we write $\mu \leq \nu$ if for all measurable $A \subset \mathcal{Y}$ it holds that $\mu(A) \leq \nu(A)$. For a measurable map $T : \mathcal{X} \to \mathcal{Y}$, we use $T\sharp$ to denote the associated pushforward operator $\mathcal{P}(\mathcal{X}) \to \mathcal{P}(\mathcal{Y})$.

## 2 Background on Optimal Transport

In this section, we give an overview of the OT theory concepts related to our paper. For details on OT, we refer to [62, 65, 56], partial OT - [22, 10].

**Standard OT formulation.** Let $c : \mathcal{X} \times \mathcal{Y} \to \mathbb{R}$ be a continuous cost function. For $\mathbb{P} \in \mathcal{P}(\mathcal{X})$, $\mathbb{Q} \in \mathcal{P}(\mathcal{Y})$, the OT cost between them is given by

$$\text{Cost}(\mathbb{P}, \mathbb{Q}) \stackrel{def}{=} \inf_{T\sharp\mathbb{P}=\mathbb{Q}} \int_{\mathcal{X}} c(x, T(x)) d\mathbb{P}(x), \tag{1}$$

where $\inf$ is taken over measurable $T : \mathcal{X} \to \mathcal{Y}$ pushing $\mathbb{P}$ to $\mathbb{Q}$ (transport maps), see Fig. 3a. Problem (1) is called the *Monge's OT* problem, and its minimizer $T^*$ is called an *OT map*.

In some cases, there may be no minimizer $T^*$ of (1). Therefore, it is common to consider *Kantorovich's* relaxation:

$$\text{Cost}(\mathbb{P}, \mathbb{Q}) \stackrel{def}{=} \inf_{\pi\in\Pi(\mathbb{P},\mathbb{Q})} \int_{\mathcal{X}\times\mathcal{Y}} c(x, y) d\pi(x, y), \tag{2}$$

where $\inf$ is taken over $\pi \in \mathcal{P}(\mathcal{X} \times \mathcal{Y})$ satisfying $\pi_x = \mathbb{P}$ and $\pi_y = \mathbb{Q}$, respectively. A minimizer $\pi^* \in \Pi(\mathbb{P}, \mathbb{Q})$ in (2) always exists and is called an *OT plan*. A widely used example of OT cost for $\mathcal{X} = \mathcal{Y} = \mathbb{R}^d$ is the Wasserstein-1 distance ($\mathbb{W}_1$), i.e., OT cost (2) for $c(x, y) = \|x - y\|$.

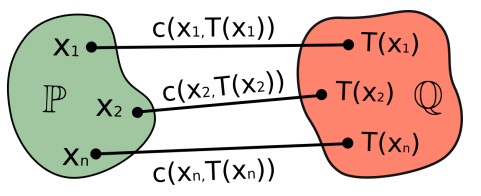

(a) Monge's OT formulation.

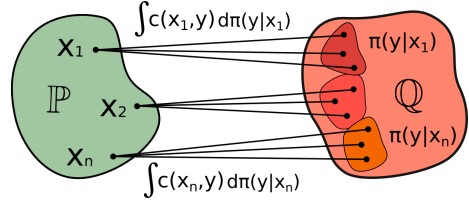

(b) Kantorovich's OT formulation.

Figure 3: Classic optimal transport (OT) formulations.

To provide an intuition behind (2), we disintegrate $d\pi(x,y) = d\pi_x(x)d\pi(y|x)$:

$$\inf_{\pi \in \Pi(\mathbb{P},\mathbb{Q})} \int_{\mathcal{X}} \left\{ \int_{\mathcal{Y}} c(x,y)d\pi(y|x) \right\} \underbrace{d\mathbb{P}(x)}_{=d\pi_x(x)}, \tag{3}$$

i.e., (2) can be viewed as an extension of (1) allowing to *split* the mass of input points $x \sim \mathbb{P}$ (Fig. 3b). With mild assumptions on $\mathbb{P}, \mathbb{Q}$, the OT cost value (2) coincides with (1), see [62, Theorem 1.33].

**Partial OT formulation.** Let $w_0, w_1 \geq m \geq 0$. We consider

$$\inf_{\substack{m\pi_x \leq w_0\mathbb{P} \\ m\pi_y \leq w_1\mathbb{Q}}} \int_{\mathcal{X} \times \mathcal{Y}} c(x,y)d\big[m\pi\big](x,y), \tag{4}$$

where inf is taken over $\pi \in \mathcal{P}(\mathcal{X} \times \mathcal{Y})$ satisfying the inequality constraints $m\pi_x \leq w_0\mathbb{P}$ and $m\pi_y \leq w_1\mathbb{Q}$. Minimizers $\pi^*$ of (4) are called *partial OT plans* (Fig. 4).

Here the inputs are two measures $w_0\mathbb{P}$ and $w_1\mathbb{Q}$ with masses $w_0$ and $w_1$. Intuitively, we need to match a $\frac{m}{w_0}$-th fraction $m\pi_x$ of the first measure $w_0\mathbb{P}$ with a $\frac{m}{w_1}$-th fraction $m\pi_y$ of the second measure $w_1\mathbb{Q}$ (Fig. 4); choosing $\pi_x, \pi_y$ is also a part of this problem. The key difference from problem (4) is that the constraints are *inequalities*. In the particular case $m = w_0 = w_1$, problem (4) reduces to (2) as the inequality constraints can be replaced by equalities.

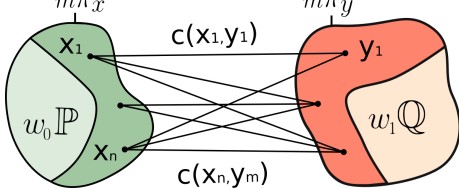

Figure 4: Partial optimal transport formulation.

## 3 Main Results

First, we formulate the extremal transport (ET) problem (§3.1). Next, we prove that ET maps can be recovered as a limit of incomplete transport (IT) maps (§3.2, 3.3). Then we propose an algorithm to solve the IT problem (§3.4). We provide the proofs for all the theorems in Appendix F.

### 3.1 Extremal Transport Problem

Popular unpaired translation methods, e.g., [72, §3.1] and [33, §3], de-facto assume that available samples $x, y$ from the input and output domains come from the data distributions $\mathbb{P}, \mathbb{Q} \in \mathcal{P}(\mathcal{X}), \mathcal{P}(\mathcal{Y})$. As a result, in their optimization objectives, the *domain loss* compares the translated $T(x) \sim T\sharp\mathbb{P}$ and target samples $y \sim \mathbb{Q}$ by using a metric for comparing *probability measures*, e.g., GAN loss [27]. Thus, the target *domain* is identified with the probability measure $\mathbb{Q}$.

We pick a different approach to define what the *domain* is. We still assume that the available data comes from data distributions, i.e., $x \sim \mathbb{P}, y \sim \mathbb{Q}$. However, we say that the target domain is the part of $\mathcal{Y}$ where the probability mass of $\mathbb{Q}$ lives.[2] Namely, it is $\text{Supp}(\mathbb{Q}) \subset \mathcal{Y}$. We say that a map $T$ translates the domains if $\text{Supp}(T\sharp\mathbb{P}) \subset \text{Supp}(\mathbb{Q})$. This requirement is weaker than the usual $T\sharp\mathbb{P} = \mathbb{Q}$. Assume that $c(x,y)$ is a function estimating the dissimilarity between $x, y$. We would like to pick $T(x) \in \text{Supp}(\mathbb{Q})$ which is *maximally* similar to $x$ in terms of $c(x,y)$. This preference of $T$ can be formalized as follows:

$$\text{Cost}_\infty(\mathbb{P}, \mathbb{Q}) \stackrel{def}{=} \inf_{\text{Supp}(T\sharp\mathbb{P}) \subset \text{Supp}(\mathbb{Q})} \int_{\mathcal{X}} c\big(x, T(x)\big)d\mathbb{P}(x), \tag{5}$$

where the inf is taken over measurable $T : \mathcal{X} \to \mathcal{Y}$ which map the probability mass of $\mathbb{P}$ to $\text{Supp}(\mathbb{Q})$. We say that (5) is the (Monge's) *extremal transport* (ET) problem.

---

[2]Following the standard manifold hypothesis [21], real data distribution $\mathbb{Q}$ is usually supported on a small-dimensional manifold $M = \text{Supp}(\mathbb{Q}) \subset [-1, 1]^D$ occupying a **tiny** part of the ambient space $[-1, 1]^D$.

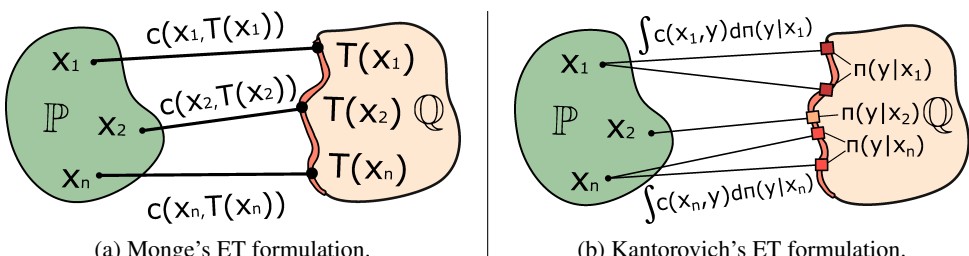

(a) Monge's ET formulation.  (b) Kantorovich's ET formulation.

Figure 5: Extremal transport (ET) formulations.

Problem (5) is **atypical** for the common OT framework. For example, the usual measure-preserving constraint $T\sharp\mathbb{P} = \mathbb{Q}$ in (1) is replaced with $\text{Supp}(T\sharp\mathbb{P}) \subset \text{Supp}(\mathbb{Q})$ which is more tricky. Importantly, measure $\mathbb{Q}$ can be replaced with any other $\mathbb{Q}' \in \mathcal{P}(\mathcal{Y})$ with the same support yielding the same inf.

Below we analyse the minimizers $T^*$ of (5). We define $c^*(x) \stackrel{def}{=} \min_{y \in \text{Supp}(\mathbb{Q})} c(x, y)$. Here the $\min$ is indeed attained (for all $x \in \mathcal{X}$) because $c(x, y)$ is continuous and $\text{Supp}(\mathbb{Q}) \subset \mathcal{Y}$ is a compact set. The value $c^*(x)$ can be understood as the *lowest* possible transport cost when mapping the mass of point $x$ to the support of $\mathbb{Q}$. For any admissible $T$ in (5), it holds ($\mathbb{P}$-almost surely):

$$c^*(x) = \min_{y \in \text{Supp}(\mathbb{Q})} c(x, y) \leq c\big(x, T(x)\big). \tag{6}$$

**Proposition 1** (Continuity of $c^*$). *It holds that $c^* \in \mathcal{C}(\mathcal{X})$.*

As a consequence of Proposition 1, we see that $c^*$ is measurable. We integrate (6) w.r.t. $x \sim \mathbb{P}$ and take inf over all feasible $T$. This yields a lower bound on $\text{Cost}_\infty(\mathbb{P}, \mathbb{Q})$:

$$\int_{\mathcal{X}} c^*(x)d\mathbb{P}(x) \leq \overbrace{\inf_{\substack{\text{Supp}(T\sharp\mathbb{P}) \\ \subset \text{Supp}(\mathbb{Q})}} \int_{\mathcal{X}} c\big(x, T(x)\big)d\mathbb{P}(x)}^{\text{Cost}_\infty(\mathbb{P}, \mathbb{Q})}. \tag{7}$$

There exists admissible $T$ making (7) the equality. Indeed, let $\text{NN}(x) \stackrel{def}{=} \{y \in \text{Supp}(\mathbb{Q}) \text{ s.t. } c(x, y) = c^*(x)\}$ be the set of points $y$ which attain $\min$ in the definition of $c^*$. These points are the closest to $x$ points in $\mathbb{Q}$ w.r.t. the cost $c(x, y)$. We call them the *nearest neighbors* of $x$. From this perspective, we see that (7) turns to equality if and only if $T(x) \in \text{NN}(x)$ holds for $\mathbb{P}$-almost all $x \in \mathcal{X}$, i.e., $T$ maps points $x \sim \mathbb{P}$ to their nearest neighbors in $\text{Supp}(\mathbb{Q})$. We need to make sure that such *measurable* $T$ exists (Fig. 5a).

**Theorem 1** (Existence of ET maps). *There exists at least one measurable map $T^* : \mathcal{X} \to \mathcal{Y}$ minimizing (5). For $\mathbb{P}$-almost all $x \in \mathcal{X}$ it holds that $T^*(x) \in \text{NN}(x)$. Besides,*

$$\text{Cost}_\infty(\mathbb{P}, \mathbb{Q}) = \int_{\mathcal{X}} c^*(x)d\mathbb{P}(x).$$

We say that $\text{Cost}_\infty(\mathbb{P}, \mathbb{Q})$ is the *extremal cost* because one can not obtain smaller cost when moving the mass of $\mathbb{P}$ to $\text{Supp}(\mathbb{Q})$. In turn, we say that minimizers $T^*$ are *ET maps*.

One may extend the ET problem (8) in the Kantorovich's manner by allowing the mass splitting and stochastic plans:

$$\text{Cost}_\infty(\mathbb{P}, \mathbb{Q}) \stackrel{def}{=} \inf_{\pi \in \Pi^\infty(\mathbb{P}, \mathbb{Q})} \int_{\mathcal{X} \times \mathcal{Y}} c(x, y)d\pi(x, y), \tag{8}$$

where $\Pi^\infty(\mathbb{P}, \mathbb{Q})$ are probability measures $\pi \in \mathcal{P}(\mathcal{X} \times \mathcal{Y})$ s.t. $\pi_x = \mathbb{P}$ and $\text{Supp}(\pi_y) \subset \text{Supp}(\mathbb{Q})$. To understand the structure of minimizers in (8), it is more convenient to disintegrate $d\pi(x, y) = d\pi(y|x)d\pi_x(x)$:

$$\text{Cost}_\infty(\mathbb{P}, \mathbb{Q}) = \inf_{\pi \in \Pi^\infty(\mathbb{P}, \mathbb{Q})} \int_{\mathcal{X}} \int_{\mathcal{Y}} c(x, y)d\pi(y|x) \underbrace{d\mathbb{P}(x)}_{=d\pi_x(x)}. \tag{9}$$

Thus, computing (8) boils down to computing a family of conditional measures $\pi(\cdot|x)$ minimizing (8). As in (7), for any $\pi \in \Pi^\infty(\mathbb{P}, \mathbb{Q})$, it holds (for $\mathbb{P}$-almost all $x \in \mathcal{X}$) that

$$c^*(x) = \min_{y \in \text{Supp}(\mathbb{Q})} c(x, y) \leq \int_{\mathcal{Y}} c(x, y)d\pi(y|x) \tag{10}$$

because $\pi$ redistributes the mass of $\mathbb{P}$ to $\text{Supp}(\mathbb{Q})$. By integrating (10) w.r.t. $x \sim \mathbb{P} = \pi_x$ and taking inf over all admissible plans $\pi$, we derive that $\int_{\mathcal{X}} c^*(x)d\mathbb{P}(x)$ is a lower bound for (8). In particular,

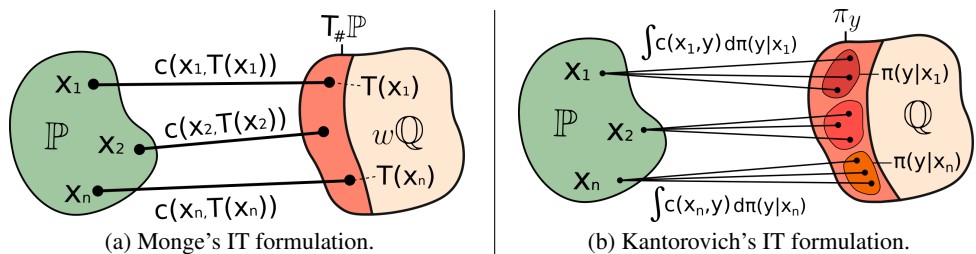

Figure 6: Incomplete transport (IT) formulations.

(a) Monge's IT formulation.

(b) Kantorovich's IT formulation.

the bound is tight for $\pi^*(y|x) = \delta_{T^*(x)}$, where $T^*$ is the ET map from our Theorem 1. Therefore, the value (8) is the same as (5) but possibly admits more minimizers. We call the minimizers $\pi^*$ of (8) the *ET plans* (Fig. 5b).

From (10) and the definition of $\text{NN}(x)$, we see that minimizers $\pi^*$ are the plans for which $\pi^*(y|x)$ redistributes the mass of $x$ among the nearest neighbors $y \in \text{NN}(x)$ of $x$ (for $\mathbb{P}$-almost all $x \in \mathcal{X}$). As a result, $\pi^*(y|x)$ can be viewed as a *stochastic nearest neighbor assignment* between the probability mass of $\mathbb{P}$ and the support of $\mathbb{Q}$.

## 3.2    Incomplete Transport Problem

In practice, solving extremal problem (5) is challenging because it is hard to enforce $\text{Supp}(T\sharp\mathbb{P}) \subset \text{Supp}(\mathbb{Q})$. To avoid enforcing this constraint, we replace it and consider the following problem with finite parameter $w \geq 1$:

$$\text{Cost}_w(\mathbb{P}, \mathbb{Q}) \overset{def}{=} \inf_{T\sharp\mathbb{P} \leq w\mathbb{Q}} \int_{\mathcal{X}} c(x, T(x))d\mathbb{P}(x). \tag{11}$$

We call (11) Monge's *incomplete transport* (IT) problem (Fig. 6a). With the increase of $w$, admissible maps $T$ obtain more ways to redistribute the mass of $\mathbb{P}$ among $\text{Supp}(\mathbb{Q})$. Informally, when $w \to \infty$, the constraint $T\sharp\mathbb{P} \leq w\mathbb{Q}$ in (11) tends to the constraint $\text{Supp}(T\sharp\mathbb{P}) \subset \text{Supp}(\mathbb{Q})$ in (5), i.e., (11) itself tends to ET problem (5). We will formalize this statement a few paragraphs later (in §3.3).

As in (1), problem (11) may have no minimizer $T^*$ or even may have the empty feasible set. Therefore, it is natural to relax problem (11) in the Kantorovich's manner:

$$\text{Cost}_w(\mathbb{P}, \mathbb{Q}) \overset{def}{=} \inf_{\pi \in \Pi^w(\mathbb{P}, \mathbb{Q})} \int_{\mathcal{X} \times \mathcal{Y}} c(x, y)d\pi(x, y), \tag{12}$$

where the inf is taken over the set $\Pi^w(\mathbb{P}, \mathbb{Q})$ of probability measures $\pi \in \mathcal{P}(\mathcal{X} \times \mathcal{Y})$ whose first marginal is $\pi_x = \mathbb{P}$, and the second marginal satisfies $\pi_y \leq w\mathbb{Q}$ (Fig. 6b).

We note that IT problem (12) is a special case of partial OT (4) with $w_0 = m = 1$ and $w_1 = w$. In (12), one may actually replace inf with min, see our proposition below.

**Proposition 2** (Existence of IT plans). *Problem* (12) *admits at least one minimizer* $\pi^* \in \Pi^w(\mathbb{P}, \mathbb{Q})$.

We say that minimizers of (12) are *IT plans*. In the general case, Kantorovich's *IT cost* (12) always lower bounds Monge's counterpart (11). Below we show that they coincide in the practically most interesting Euclidean case.

**Proposition 3** (Equivalence of Monge's, Kantorovich's IT costs). *Let* $\mathcal{X}, \mathcal{Y} \subset \mathbb{R}^D$ *be two compact sets,* $\mathbb{P} \in \mathcal{P}(\mathcal{X})$ *be atomless,* $\mathbb{Q} \in \mathcal{P}(\mathcal{Y})$. *Then Monge's* (11) *and Kantorovich's* (12) *IT costs coincide.*

However, it is not guaranteed that inf in Monge's problem (11) is attained even in the Euclidean case. Still for general Polish spaces $\mathcal{X}, \mathcal{Y}$ it is clear that if there exists a deterministic IT plan in Kantorovich's problem (12) of the form $\pi^* = [\text{id}_{\mathcal{X}}, T^*]$, then $T^*$ is an IT map in (11), and the IT Monge's (11) and Kantorovich's (12) costs coincide. Henceforth, for simplicity, we assume that $\mathcal{X}, \mathcal{Y}, c, \mathbb{P}, \mathbb{Q}$ are such that (11) and (12) coincide, e.g., those from Prop. 3.

IT problem (12) can be viewed as an interpolation between OT (2) and ET problems (8). Indeed, when $w = 1$, the constraint $\pi_y \leq \mathbb{Q}$ is equivalent to $\pi_y = \mathbb{Q}$ as there is only one *probability* measure which is $\leq \mathbb{Q}$, and it is $\mathbb{Q}$ itself. Thus, IT (12) with $w = 1$ coincides with OT (2). In the next section, we show that for $w \to \infty$ one recovers ET from IT.

## 3.3    Link between Incomplete and Extremal Transport

Now we connect incomplete (12) and extremal (8) transport tasks.

**Theorem 2** (IT costs converge to the ET cost when $w \to \infty$). *Function $w \mapsto \text{Cost}_w(\mathbb{P}, \mathbb{Q})$ is convex, non-increasing in $w \in [1, +\infty)$ and*

$$\lim_{w \to \infty} \text{Cost}_w(\mathbb{P}, \mathbb{Q}) = \text{Cost}_\infty(\mathbb{P}, \mathbb{Q}).$$

A natural subsequent question here is whether IT plans in (12) converge to ET plans (8) when $w \to \infty$. Our following result sheds the light on this question.

**Theorem 3** (IT plans converge to ET plans when $w \to \infty$). *Consider $w_1, w_2, w_3, \cdots \geq 1$ satisfying $\lim_{n \to \infty} w_n = \infty$. Let $\pi^{w_n} \in \Pi^{w_n}(\mathbb{P}, \mathbb{Q})$ be a sequence of IT plans solving (12) with $w = w_n$, respectively. Then it has a (weakly-\*) converging sub-sequence. Every such sub-sequence of IT plans converges to an ET plan $\pi^* \in \Pi^\infty(\mathbb{P}, \mathbb{Q})$.*

In general, there may be sub-sequences of IT plans converging to different ET plans $\pi^* \in \Pi^\infty(\mathbb{P}, \mathbb{Q})$. However, our following corollary shows that with the increase of weight $w$, elements of **any** sub-sequence become closer to the **set** of ET plans.

**Corollary 1** (IT plans become closer to the set of ET plans when $w \to \infty$). *For all $\varepsilon > 0 \; \exists w(\varepsilon) \in [1, \infty)$ such that $\forall w \geq w(\varepsilon)$ and $\forall$ IT plan $\pi^w \in \Pi_w(\mathbb{P}, \mathbb{Q})$ solving Kantorovich's IT problem (12), there exists an ET plan $\pi^*$ which is $\varepsilon$-close to $\pi^w$ in $\mathbb{W}_1$, i.e., $\mathbb{W}_1(\pi^*, \pi^w) \leq \varepsilon$.*

Providing a stronger convergence result here is challenging, and we leave this theoretical question open for future studies. Our Theorems 2, 3 and Corollary 1 suggest that to obtain a fine approximation of an ET plan ($w = \infty$), one may use an IT plan for sufficiently large finite $w$. In Appendix G.1, we *empirically demonstrate* this observation through an experiment where the ground-truth deterministic ET plan is analytically known. Below we develop a neural algorithm to compute IT plans.

### 3.4 Computational Algorithm for Incomplete Transport

To begin with, for IT (12), we derive the dual problem.

**Theorem 4** (Dual problem for IT). *It holds*

$$\text{Cost}_w(\mathbb{P}, \mathbb{Q}) = \max_{f \leq 0} \int_{\mathcal{X}} f^c(x) d\mathbb{P}(x) + w \int_{\mathcal{Y}} f(y) d\mathbb{Q}(y), \tag{13}$$

*where the $\max$ is taken over non-positive $f \in \mathcal{C}(\mathcal{Y})$ and $f^c(x) \stackrel{def}{=} \min_{y \in \mathcal{Y}} \{c(x, y) - f(y)\}$.*

We call the function $f$ *potential*. In the definition of $f^c$, $\min$ is attained because $c, f$ are continuous and $\mathcal{Y}$ is compact. The function $f^c$ is called the $c$-transform of $f$.

The difference of formula (13) from usual $c$-transform-based duality formulas for OT (2), see [62, §1.2], [65, §5], is that $f$ is required to be non-positive and the second term is multiplied by $w \geq 1$. We rewrite the term $\int_{\mathcal{X}} f^c(x) d\mathbb{P}(x)$ in (13):

$$\int_{\mathcal{X}} f^c(x) d\mathbb{P}(x) = \int_{\mathcal{X}} \min_{y \in \mathcal{Y}} \{c(x, y) - f(y)\} d\mathbb{P}(x) = \inf_{T: \mathcal{X} \to \mathcal{Y}} \int_{\mathcal{X}} \{c(x, T(x)) - f(y)\} d\mathbb{P}(x). \tag{14}$$

Here we use the interchange between the integral and $\inf$ [59, Theorem 3A]; in (14) the $\inf$ is taken over measurable maps. Since $(x, y) \mapsto c(x, y) - f(y)$ is a continuous function on a compact set, it admits a measurable selection $T(x) \in \arg\min_{y \in \mathcal{Y}} \{c(x, y) - f(y)\}$ minimizing (14), see [2, Theorem 18.19]. Thus, $\inf$ can be replaced by $\min$. We combine (14) and (13) and obtain an equivalent saddle point problem:

$$\text{Cost}_w(\mathbb{P}, \mathbb{Q}) = \max_{f \leq 0} \min_{T: \mathcal{X} \to \mathcal{Y}} \mathcal{L}(f, T), \tag{15}$$

where the functional $\mathcal{L}(f, T)$ is defined by

$$\mathcal{L}(f, T) \stackrel{def}{=} \int_{\mathcal{X}} c(x, T(x)) d\mathbb{P}(x) - \int_{\mathcal{X}} f(T(x)) d\mathbb{P}(x) + w \int_{\mathcal{Y}} f(y) d\mathbb{Q}(y). \tag{16}$$

Functional $\mathcal{L}(f, T)$ can be viewed as a Lagrangian with $f \leq 0$ being a multiplier for the constraint $T\sharp\mathbb{P} - w\mathbb{Q} \leq 0$. By solving (15), one may obtain IT maps.

**Theorem 5** (IT maps are contained in optimal saddle points). *Let $f^*$ be any maximizer in (13). If $\pi^* \in \Pi^w(\mathbb{P}, \mathbb{Q})$ is a deterministic IT plan, i.e., it solves (12) and has the form $\pi^* = [\text{id}_{\mathcal{X}}, T^*]\sharp\mathbb{P}$ for some measurable $T^* : \mathcal{X} \to \mathcal{Y}$, then*

$$T^* \in \arg\min_{T: \mathcal{X} \to \mathcal{Y}} \mathcal{L}(f^*, T).$$

---

**Algorithm 1:** Procedure to compute the IT map between $\mathbb{P}$ and $\mathbb{Q}$ for transport cost $c(x, y)$ and weight $w$.

---

**Input** : distributions $\mathbb{P}, \mathbb{Q}$ accessible by samples; mapper $T_\theta : \mathcal{X} \to \mathcal{Y}$; potential $f_\psi : \mathcal{X} \to \mathbb{R}_-$; transport cost $c : \mathcal{X} \times \mathcal{Y} \to \mathbb{R}$; weight $w \geq 1$; number $K_T$ of inner iterations;
**Output** : approximate IT map $(T_\theta)_{\#}\mathbb{P} \leq w\mathbb{Q}$;
**repeat**
   Sample batches $X \sim \mathbb{P}, Y \sim \mathbb{Q}$;
   $\mathcal{L}_f \leftarrow w \cdot \frac{1}{|Y|} \sum_{y \in Y} f_\psi(y) - \frac{1}{|X|} \sum_{x \in X} f_\psi(T_\theta(x))$;
   Update $\psi$ by using $\frac{\partial \mathcal{L}_f}{\partial \psi}$ to maximize $\mathcal{L}_f$;
   **for** $k_T = 1, 2, \ldots, K_T$ **do**
      Sample batch $X \sim \mathbb{P}$;
      $\mathcal{L}_T \leftarrow \frac{1}{|X|} \sum_{x \in X} \big[ c(x, T_\theta(x)) - f_\psi(T_\theta(x)) \big]$;
      Update $\theta$ by using $\frac{\partial \mathcal{L}_T}{\partial \theta}$ to minimize $\mathcal{L}_T$;
**until** *not converged*;

---

Our Theorem 5 states that in *some* optimal saddle points $(f^*, T^*)$ of (15) it holds that $T^*$ is the IT map between $\mathbb{P}, \mathbb{Q}$. In general, the $\arg\inf_T$ set for an optimal $f^*$ might contain not only IT maps $T^*$, but other functions as well (*fake solutions*), see limitations in Appendix A.

We solve the optimization problem (15) by approximating the map $T$ and potential $f$ with neural networks $T_\theta$ and $f_\psi$, respectively. To make $f_\psi$ non-positive, we use $x \mapsto -|x|$ as the last layer. The nets are trained using random batches from $\mathbb{P}, \mathbb{Q}$ and stochastic gradient ascent-descent. We detail the optimization procedure in Algorithm 1.

## 4 Related work

**OT in generative models.** A popular way to apply OT in generative models is to use the **OT cost** as the loss function to update the generator [4, 28, 26], see [41] for a survey. These methods are *not relevant* to our study as they do not learn an OT map but only compute the OT cost.

Recent works [43, 42, 61, 20, 5, 24, 29] are the most related to our study. These papers show the possibility to learn the **OT maps** (or plans) via solving saddle point optimization problems derived from the standard $c$-transform-based duality formulas for OT. The underlying principle of our objective (15) is analogous to theirs. The key **difference** is that they consider OT problems (1), (2) and enforce the *equality* costraints, e.g., $T\sharp\mathbb{P} = \mathbb{Q}$, while our approach enforces the *inequality* constraint $T\sharp\mathbb{P} \leq w\mathbb{Q}$ allowing to *partially* align the measures. *We provide a detailed discussion of relation with these works as well as with the fundamental OT* (2) *and partial OT* (4) *literature* [22, 10, 62] in Appendix F, see bibliographic remarks after the proofs of Theorems 2, 4 and 5.

For completeness, we also mention other existing neural OT methods [25, 63, 49, 17, 39]. These works are less related to our work because they either underperform compared to the above-mentioned saddle point methods, see [40] for evaluation, or they solve specific OT formulations, e.g., entropic OT [56, §4], which are not relevant to our study.

The papers [66, 47, 19] are slightly more related to our work. They propose neural methods for unbalanced OT [14] which can also be used to *partially* align measures. As we will see in Appendix B, UOT is hardly suitable for ET (8) as it is not easy to control how it spreads the probability mass. Besides, these methods consider OT between small-dimensional datasets or in latent spaces. It is not clear whether they scale to high-dimensions, e.g., images.

**Discrete OT methods**, including partial OT [11], are not relevant to us, see Appendix D for details.

**Unpaired domain translation** [3] is a generic problem which includes image super-resolution [12], inpainting [71], style translation [34] tasks, etc. Hence, we do not mention all of the existing approaches but focus on their common main features instead. In many applications, it is important to preserve semantic information during the translation, e.g., the image content. In most cases, to do this it is sufficient to use convolutional neural networks. They preserve the image content thanks to their design which is targeted to only locally change the image [18]. However, in some of the tasks

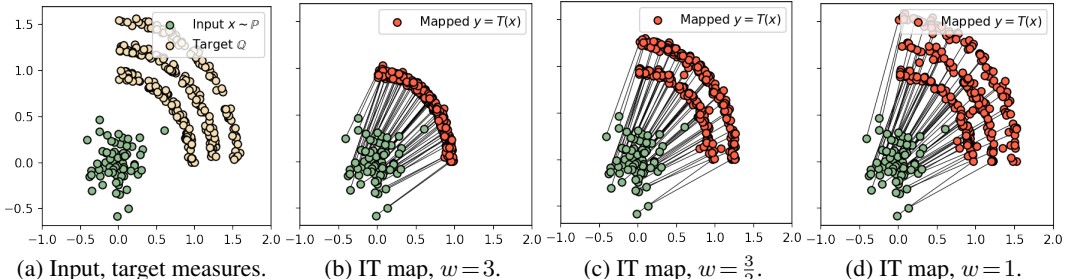

Figure 7: Incomplete transport (IT) maps learned by our Algorithm 1 in *'Wi-Fi'* experiment.

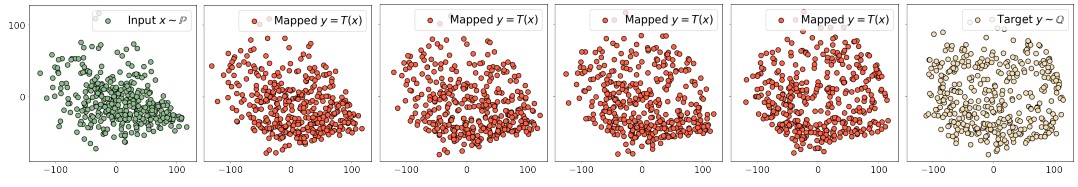

(a) Input measure.    (b) IT map, $w=8$.    (c) IT map, $w=4$.    (d) IT map, $w=2$.    (e) IT map, $w=1$.    (f) Target measure.

Figure 8: PCA projections of input $x \sim \mathbb{P}$, mapped $T(x) \sim T \sharp \mathbb{P}$ and target $y \sim \mathbb{Q}$ test samples (*handbag→shoes* experiment).

additional image properties must be kept, e.g., image colors in super-resolution. Typical approaches to such problems are based on GANs and use additional *similarity* losses, e.g., the basic CycleGAN [72] enforces $\ell^1$ similarity. Such methods are mostly related to our work. However, without additional modifications most of these approaches only partially maintain the image properties, see [44, Figure 5]. As we show in experiments (Appendix C), IT achieves better similarity than popular CycleGAN, StarGAN-v2 [15] methods, both default and modified to better preserve the image content.

## 5 Evaluation

In §5.1, we provide illustrative toy 2D examples. In §5.2, we evaluate our method on the unpaired image-to-image translation task. Technical *training details* (architectures, learning rates, etc.) are given in Appendix E. The code is written using `PyTorch` framework and is publicly available at

$$\texttt{https://github.com/milenagazdieva/ExtremalNeuralOptimalTransport}.$$

**Transport costs.** We experiment with the quadratic cost $c(x,y) = \ell^2(x,y)$ as this cost already provides reasonable performance. We slightly abuse the notation and use $\ell^2$ to denote the squared error *normalized* by the dimension. Experiments with the *perceptual cost* are given in Appendix G.3.

### 5.1 Toy 2D experiments

In this section, we provide *'Wi-Fi'* and *'Accept'* examples in 2D to show how the choice of $w$ affects the fraction of the target measure $\mathbb{Q}$ to which the probability mass of the input $\mathbb{P}$ is mapped. In both cases, measure $\mathbb{P}$ is *Gaussian*. In Appendix B, we demonstrate how *other methods* perform on these *'Wi-Fi'* and *'Accept'* toy examples.

In *'Wi-Fi'* experiment (Fig. 7), target $\mathbb{Q}$ contains 3 arcs. We provide the learned IT maps for $w \in [1, \frac{3}{2}, 3]$. The results show that by varying $w$ it is possible to control the fraction of $\mathbb{Q}$ to which the mass of $\mathbb{P}$ will be mapped. In Fig. 7, we see that for $w = 1$ our IT method learns all 3 arcs. For $w = \frac{3}{2}$, it captures 2 arcs, i.e., $\approx \frac{2}{3}$ of $\mathbb{Q}$. For $w = 3$, it learns 1 arc which corresponds to $\approx \frac{1}{3}$ of $\mathbb{Q}$.

In *'Accept'* experiment (Fig. 2), target $\mathbb{Q}$ is a two-line text. Here we put $w = 2$ and, as expected, our method captures only one line of the text which is the closest to $\mathbb{P}$ in $\ell^2$.

### 5.2 Unpaired Image-to-image Translation

Here we learn IT maps between various pairs of datasets. We test $w \in \{1, 2, 4, 8\}$ in all experiments. For completeness, we consider *bigger weights* $w \in \{16, 32\}$ in Appendix G.4.

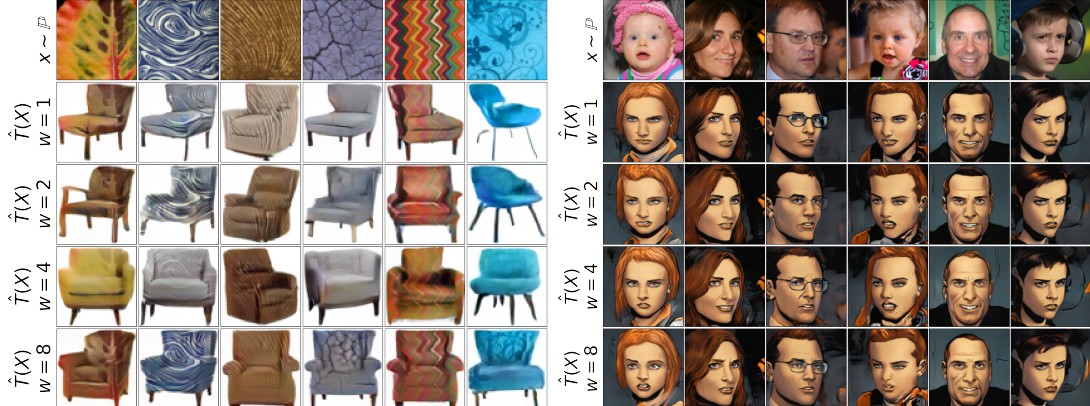

(a) IT results for *textures* → *chairs* (64×64).   (b) IT results for *ffhq* → *comics* (128×128).

Figure 9: Unpaired Translation with our Incomplete Transport.

| Experiment | $w = 1$ | $w = 2$ | $w = 4$ | $w = 8$ |
|---|---|---|---|---|
| *celeba → anime* | 0.297 | 0.154 | 0.133 | 0.094 |
| *handbag → shoes* | 0.368 | 0.320 | 0.259 | 0.252 |
| *textures → chairs* | 0.603 | 0.516 | 0.474 | 0.408 |
| *ffhq → comics* | 0.224 | 0.220 | 0.200 | 0.196 |

(a) Test $\ell^2$ transport cost of our learned IT maps.

| Experiment | $w = 1$ | $w = 2$ | $w = 4$ | $w = 8$ |
|---|---|---|---|---|
| *celeba → anime* | 14.65 | 20.79 | 22.18 | 22.84 |
| *handbag → shoes* | 27.10 | 29.70 | 42.90 | 53.80 |
| *textures → chairs* | N/A | N/A | N/A | N/A |
| *ffhq → comics* | 20.95 | 22.38 | 22.77 | 23.67 |

(b) Test FID of our learned IT maps.

Table 1: Test $\ell^2$ cost and FID of our learned IT maps.

**Image datasets.** We utilize the following publicly available datasets as $\mathbb{P}, \mathbb{Q}$: celebrity faces [46], aligned anime faces[3], flickr-faces-HQ [36], comic faces[4], Amazon handbags from LSUN dataset [68], shoes [67], textures [16] and chairs from Bonn furniture styles dataset [1]. The sizes of datasets are from 5K to 500K samples. We work with $64 \times 64$ and $128 \times 128$ images.

**Train-test split.** We use 90% of each dataset for training. The remaining 10% are held for test. All the presented qualitative and quantitative results are obtained for test images.

**Experimental results.** Our evaluation shows that with the increase of $w$ the images $\hat{T}(x)$ translated by our IT method become more similar to the respective inputs $x$ w.r.t. $\ell^2$. In Table 1a, we quantify this effect. Namely, we show that the test transport cost $\frac{1}{N_{\text{test}}} \sum_{n=1}^{N_{\text{test}}} c(x, \hat{T}(x))$ decreases with the increase of $w$ which empirically verifies our Theorem 2.

We qualitatively demonstrate this effect in Fig. 1b, 1a, 9b, 9a and 8. In *celeba* (female) → *anime* (Fig. 1b), the hair and background colors of the learned anime images become closer to celebrity faces' colors with the increase of $w$. For example, in the 4th column of Fig. 1b, the anime hair color changes from green to brown, which is close to that of the respective celebrity. In the 6th column, the background is getting darker. In *handbag→shoes* (Fig. 1a), the color, texture and size of the shoes become closer to that of handbag. Additionally, for this experiment we plot the projections of the learned IT maps to the first 2 principal components of $\mathbb{Q}$ (Fig. 8). We see that projections are close to $\mathbb{Q}$ for $w = 1$ and become closer to $\mathbb{P}$ with the increase of $w$. In *ffhq→comics*, the changes mostly affect facial expressions and individual characteristics. In *textures→chairs*, the changes are mostly related to chairs' size which is expected since we use pixel-wise $\ell^2$ as the cost function. *Additional qualitative results* are given in Appendix G.5 (Fig. 24, 25, 9a).

For completeness, we measure test FID [32] of the translated samples, see Table 1b. We do not calculate FID in the *handbag→chairs* experiment because of too small sizes of the test parts of the datasets (500 textures, 2K chairs). However, we emphasize that FID is not representative when $w > 1$. In this case, IT maps learn by construction only a part of the target measure $\mathbb{Q}$. At the same time, FID tests how well the transported samples represent the entire target distribution and is very sensitive to mode dropping [48, Fig. 1b]. Therefore, while the cost decreases with the growth of $w$, FID, on the contrary, increases. This is expected since IT maps to smaller part of $\mathbb{Q}$. Importantly, the visual quality of the translated images $\hat{T}(x)$ is not decreasing.

---

[3]`kaggle.com/reitanaka/alignedanimefaces`
[4]`kaggle.com/datasets/defileroff/comic-faces-paired-synthetic-v2`

In Appendix C, we *compare* our IT method with other image-to-image translation methods and show that IT better preserves the input-output similarity.

## 6 Potential Impact

**Inequality constraints for generative models.** The majority of optimization objectives in generative models (GANs, diffusion models, normalizing flows, etc.) enforce the *equality* constraints, e.g., $T\sharp\mathbb{P} = \mathbb{Q}$, where $T\sharp\mathbb{P}$ is the generated measure and $\mathbb{Q}$ is the data measure. Our work demonstrates that it is possible to enforce *inequality* constraints, e.g., $T\sharp\mathbb{P} \leq w\mathbb{Q}$, and apply them to a large-scale problem. While in this work we primarily focus on the image-to-image translation task, we believe that the ideas presented in our paper have several visible prospects for further improvement and applications. We list them below.

**(1) Partial OT.** Learning alignments between measures of *unequal* mass is a pervasive topic which has already perspective applications in biology to single-cell data [66, 47, 19]. The mentioned works use unbalanced OT [14]. This is an unconstrained problem where the mass spread is softly controlled by the regularization. Therefore, may be hard to control how the mass is actually distributed. Using partial OT which enforces hard inequality constraints might potentially soften this issue. Our IT problem (12) is a particular case of partial OT (4). We believe that our study is useful for future development of partial OT methods.

**(2) Generative nearest neighbors.** NNs play an important role in machine learning applications such as, e.g., image retrieval [6]. These methods typically rely on fast *discrete* approximate NN [50, 35] and perform *matching* with the latent codes of the train samples. In contrast, nowadays, with the rapid growth of large generative models such as DALL-E [58], CLIP [57], GPT-3 [9], it becomes relevant to perform *out-of-sample* estimation, e.g., map the latent vectors to *new* vectors which are not present in the train set to generate *new* data. Our IT approach (for $w \rightarrow \infty$) is a theoretically justified way to learn approximate NN maps exclusively from samples. We think our approach might acquire applications here as well, especially since there already exist ways to apply OT in latent spaces of such models [20].

**(3) Robustness and outlier detection.** Our IT aligns the input measure $\mathbb{P}$ only with a part of the target measure $\mathbb{Q}$. This property might be potentially used to make the learning robust, e.g., ignore outliers in the target dataset. Importantly, the choice of outliers and contamination level are tunable via $c(x,y)$ and $w$, but their selection may be not obvious. At the same time, the potential $f^*$ vanishes on the outliers, i.e., samples in Supp($\mathbb{Q}$) to which the mass is not transported.

**Proposition 4** (The potential vanishes at outliers)**.** *Under the assuptions of Theorem 5, the equality $f^*(y) = 0$ holds for all $y \in Supp(\mathbb{Q}) \setminus Supp(T^*\sharp\mathbb{P})$.*

We empirically illustrate this statement in Appendix A. As a result of this proposition, a possible byproduct of our method is an outlier-score $f^*(y)$ for the target data. Such applications of OT are promising and there already exist works [52, 7, 54] developing approaches to make OT more robust.

**Limitations, societal impact**. We discuss *limitations*, *societal impact* of our study in Appendix A.

ACKNOWLEDGEMENTS. This work was partially supported by Skoltech NGP Program (Skoltech-MIT joint project).

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

# A    Limitations

**Transport costs**. Our theoretical results hold true for any continuous cost function $c(x, y)$, but our experimental study uses $\ell^2$ as it already yields a reasonable performance in many cases. Considering more semantically meaningful costs for image translation, e.g., perceptual [70], is a promising future research direction.

**Intersecting supports**. ET is the nearest neighbor assignment (§3.1). Using ET may be unreasonable when $\mathcal{X} = \mathcal{Y}$ and $\text{Supp}(\mathbb{P})$ intersects with $\text{Supp}(\mathbb{Q})$. For example, if $c(x, y)$ attains minimum over $y \in \mathcal{Y}$ for a given $x \in \mathcal{X}$ at $x = y$, e.g., $c = \ell^2$, then there exists a ET plan satisfying $\pi^*(y|x) = \delta_x$ for all $x \in \text{Supp}(\mathbb{P}) \cap \text{Supp}(\mathbb{Q})$. It does not move the mass of points $x$ in this intersection. We provide an illustrative toy 2D example in Fig. 10b.

**Limited diversity.** It is theoretically impossible to preserve input-output similarity better than ET maps. Still one should understand that in some cases these maps may yield degenerate solutions. In Fig. 10a, we provide a toy 2D example of an IT map ($w = 8$) which maps all inputs to nearly the same point. In Fig. 9a (*texture → chair* translation), we see that with the increase of $w$ the IT map produces less small chairs but more large armchairs. In particular, when $w = 8$, only armchairs appear, see Fig. 9a. This is because they are closer (in $\ell^2$) to textures due to having smaller white background area.

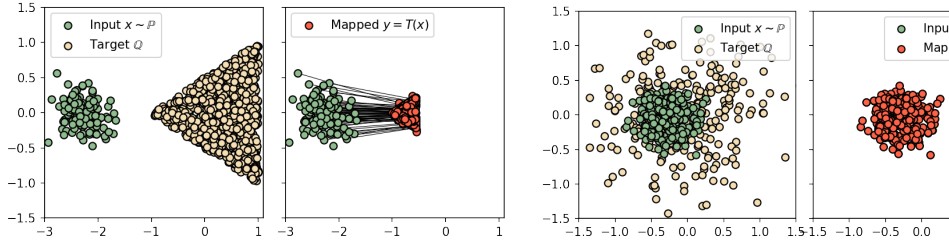

(a) **Limited diversity**. The true ET map is degenerate: it maps the entire $\mathbb{P}$ to a single vertex of the triangle $\mathbb{Q}$. The example shows the learned IT map with high $w = 20$ approximating the ET map.

(b) **Intersecting supports**. The true ET map is the identity map: it does not move the probability mass of $\mathbb{P}$. The example shows the learned IT map with high $w = 8$ approximating the ET map.

Figure 10: Toy 2D examples showing two (potential) limitations of IT maps.

**Unused samples.** Doing experiments, we noticed that the model training slows down with the increase of $w$. A possible cause of this is that some samples from $\mathbb{Q}$ become non-informative for training (this follows from our Proposition 4). Intuitively, the part of $\text{Supp}(\mathbb{Q})$ to which the samples of $\mathbb{P}$ will *not* be mapped to is not informative for training. We illustrate this effect on toy *'Wi-Fi'* example and plot the histogram of values of $f^*$ in Fig. 11. One possible negative of this observation is that the training of IT maps or, more generally, partial OT maps, may naturally require larger training datasets.

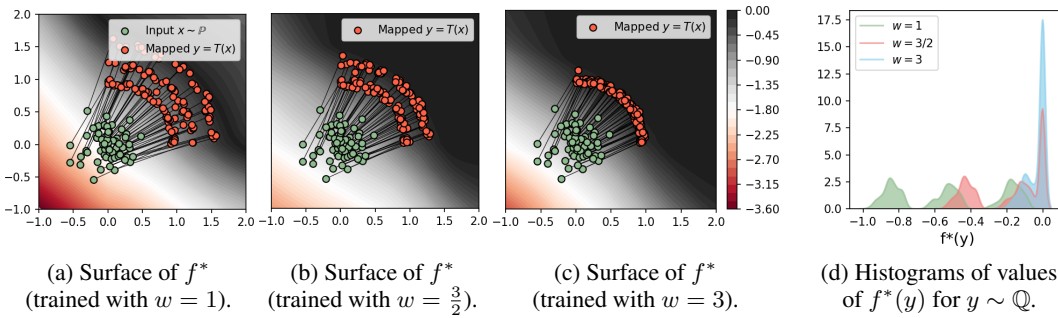

(a) Surface of $f^*$ (trained with $w = 1$).

(b) Surface of $f^*$ (trained with $w = \frac{3}{2}$).

(c) Surface of $f^*$ (trained with $w = 3$).

(d) Histograms of values of $f^*(y)$ for $y \sim \mathbb{Q}$.

Figure 11: Illustration the **unused samples**. In Figures 11a, 11b, 11c, we visualize the surface of the learned potential $f^*$ on the *'Wi-Fi'* example and $w \in \{1, \frac{3}{2}, 3\}$. In Figures 11b, 11c, the potential vanishes on the arcs of $\mathbb{Q}$ to which the mass of $\mathbb{P}$ is not mapped, i.e., $f^*(y) = 0$. In Figure 11d, we plot the distribution of values of $f^*(y)$ for $y \sim \mathbb{Q}$. For $w \in \{\frac{3}{2}, 3\}$, we see large pikes around $f^*(y) = 0$ further demonstrating that the potential equals to zero on a certain part of $\mathbb{Q}$.

**Limited quantitative metrics.** In experiments (§5), we use a limited amount of quantitative metrics. This is because existing unpaired metrics, e.g., FID [32], KID [8], etc., are **not suitable for our setup**. They aim to test equalities, such as $T \sharp \mathbb{P} = \mathbb{Q}$, while our learned maps disobey it *by the construction* (they capture only a part of $\mathbb{Q}$). Developing quality metrics for *partial* generative models is an important future research direction. Meanwhile, for our method, we have provided a toy 2D analysis (§5.1) and explanatory metrics (§5.2), such as the transport cost (Table 1a).

**Inexistent IT maps.** The actual IT plans between $\mathbb{P}, \mathbb{Q}$ may be non-deterministic, while our approach only learns a deterministic map $T$. Nevertheless, thanks to our Proposition 3, for every $\epsilon > 0$ there always exists a 1-to-1 map $T_\epsilon \sharp \mathbb{P} \leq w\mathbb{Q}$ which provides $\epsilon$-sub-optimal cost $\int_\mathcal{X} c(x, T_\epsilon(x)) d\mathbb{P}(x) \leq \mathrm{Cost}_w(\mathbb{P}, \mathbb{Q}) + \epsilon$. Thus, IT cost (12) can be approached arbitrary well with deterministic transport maps. A potential way to modify our algorithm is to learn stochastic plans is to add random noise $z$ to generator $T(x, z)$ as input, although this approach may suffer from ignoring $z$, see [43, §5.1].

**Fake solutions and instabilities.** Lagrangian objectives such as (15) may potentially have optimal saddle points $(f^*, T^*)$ in which $T^*$ is not an OT map. Such $T^*$ are called *fake solutions* [42] and may be one of the causes of training instabilities. Fake solutions can be removed by considering OT with strictly convex *weak* costs functions [43, Appendix H], see Appendix G.2 for examples.

**Potential societal impact.** Neural OT methods and, more generally, generative models are a developing research direction. They find applications such as style translation and realistic content generation. We expect that our method may improve existing applications of generative models and add new directions of neural OT usage like outlier detection. However, it should be taken into account that generative models can also be used for negative purposes such as creating fake faces.

# B    Toy 2D Illustrations of Other Methods

In this section, we demonstrate how the other methods perform in *'Wi-Fi'* and *'Accept'* experiments. We start with *'Wi-Fi'*. Assume that we would like to map $\mathbb{P}$ to the closest $\frac{2}{3}$-rd fraction of $\mathbb{Q}$, i.e., we aim to learn 2 of 3 arcs in $\mathbb{Q}$, (as in Fig. 7c).

In Fig. 12d, we show the discrete partial OT (4) [11] with parameters $w_0 = m = 1$, $w_1 = \frac{3}{2}$, corresponding to IT (12) with $w = \frac{3}{2}$. To obtain the discrete matching, we run `ot.partial.partial_wasserstein2` from POT[5]. As expected, it matches the input $\mathbb{P}$ with $\frac{2}{3}$ of $\mathbb{Q}$ and can be viewed as the ground truth (coinciding with our Fig. 7c).

First, we show the GAN [27] endowed with additional $\ell^2$ loss with weight $\lambda = 0.5$ (in Fig. 12a). Next, we consider discrete unbalanced OT [14] with the quadratic cost $c = \ell^2$. In Fig. 12b, we show the results of the matching obtained by `ot.unbalanced` with parameters $m = 1, reg = 0.1, reg_m = 1$, $numItermax = 200000$. Additionally, in Fig. 12c we show the result of neural unbalanced OT method [66].[6] To make their unbalanced setup maximally similar to our IT, we set to zero their regularization parameters. The rest parameters are default except for $\lambda_0 = 0.02$ ($\ell^2$ loss parameter), $\lambda_2 = 5$ (input and target measures' variation parameter).

We see that GAN+$\ell^2$ (Fig. 12a) and unbalanced OT (Fig. 12b and 12c) indeed match $\mathbb{P}$ with only a *part* of the target measure $\mathbb{Q}$. The transported mass is mostly concentrated in the two small arcs of $\mathbb{Q}$ which are closest to $\mathbb{P}$ w.r.t. $\ell^2$ cost. The issue here is that some mass of $\mathbb{P}$ spreads over the third (biggest) arc of $\mathbb{Q}$ yielding **outliers**. This happens because unbalanced OT (GAN can be viewed as its particular case) is an *unconstrained* problem: the mass spreading is controlled via soft penalization ($f$-divergence loss term). The lack of hard constraints, such as those in partial OT (4) or IT (12), makes it challenging to strictly control how the mass in unbalanced OT actually spreads.

For completeness, we also show the results of these methods applied to *'Accept'* experiment, see Fig. 13. Here we tested various hyperparameters for these methods but did not achieve the desired behaviour, i.e., learning only the text *'Accept'*. Moreover, we noted that GAN+$\ell^2$ for large $\lambda$ yields undesirable artifacts (Fig. 13a). This is because GAN and $\ell^2$ losses contradict to each other and still the models tried to minimize them both. We further discuss in Appendix C below.

---

[5]`pythonot.github.io`
[6]`github.com/uhlerlab/unbalanced_ot`

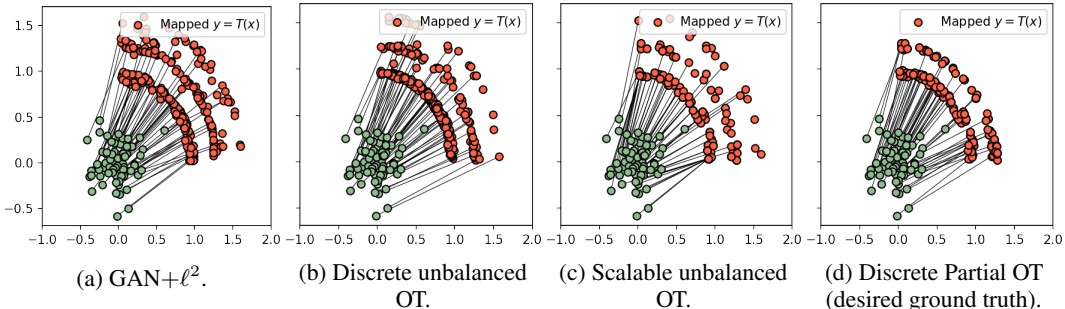

(a) GAN+$\ell^2$.
(b) Discrete unbalanced OT.
(c) Scalable unbalanced OT.
(d) Discrete Partial OT (desired ground truth).

Figure 12: Transport maps learned by various methods in *'Wi-Fi'* experiment (Fig. 7a).

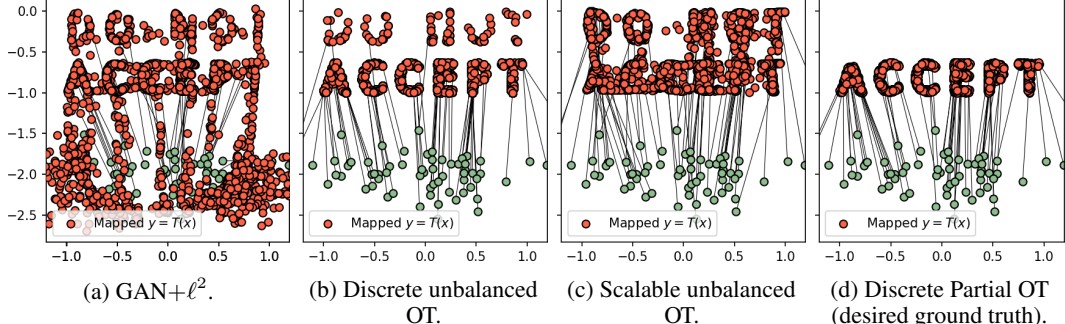

(a) GAN+$\ell^2$.
(b) Discrete unbalanced OT.
(c) Scalable unbalanced OT.
(d) Discrete Partial OT (desired ground truth).

Figure 13: Transport maps learned by various methods in *'Accept'* experiment.

## C   Comparison with Other Image-to-Image Translation Methods

Recall that ET by design is the best translation between a pair of domains w.r.t. the **given** dissimilarity function $c(x, y)$. Our IT maps with the increase of $w$ provide better input-output similarity and recover ET when $w \to \infty$ (§3.3). A reader may naturally ask: **(a)** How else can we recover ET maps? **(b)** To which extent one can control the input-output similarity in existing translation methods? **(c)** Can these methods be used to approximate ET? We discuss these aspects below.

Many translation methods are based on GANs, see [55, 3, 13] for a survey. Their learning objectives are usually combined of several loss terms:

$$\mathcal{L}_{\text{Total}}(T) \overset{def}{=} \mathcal{L}_{Dom}(T) + \lambda \cdot \mathcal{L}_{Sim}(T) + [\text{other terms}]. \tag{17}$$

In (17), the **domain loss** $\mathcal{L}_{Dom}$ is usually the vanilla GAN loss involving a discriminator [27] ensuring that the learned map $x \mapsto T(x)$ transforms inputs $x \sim \mathbb{P}$ to the samples from the target data distribution $\mathbb{Q}$. The **similarity loss** $\mathcal{L}_{Sim}$ (with $\lambda \geq 0$) is usually the *identity* loss $\int_{\mathcal{X}} \|x - T(x)\|_1 d\mathbb{P}(x)$. More generally, it can be an arbitrary *unsupervised* loss of the form $\int_{\mathcal{X}} c(x, T(x)) d\mathbb{P}(x)$ stimulating the output sample $T(x)$ to look like the input samples $x$ w.r.t. given dissimilarity function $c$, e.g., $\ell^1, \ell^2, \ell^p$, perceptual, etc. The other terms in (17) involve model-specific terms (e.g., cycle consistency loss in CycleGAN) which are not related to our study.

When learning a model via optimizing (17), a natural way to get better similarity of $x$ and $T(x)$ in (17) is to simply increase weight $\lambda$ of the corresponding loss term. This is a straightforward approach but it has a visible **limitation**. When $\lambda$ is high, the term $\lambda \cdot \mathcal{L}_{\text{Sim}}$ dominates over the other terms such as $\mathcal{L}_{\text{Dom}}$, and the model $T$ simply learns to minimize this loss ignoring the fact that the output sample should be from the target data distribution $\mathbb{Q}$. In other words, in (17) there is a nasty **trade-off** between $T(x)$ belonging to the target data distribution $\mathbb{Q}$ and input-output similarity of $x$ and $T(x)$. The parameter $\lambda \geq 0$ controls this *realism-similarity* trade-off, and we study how it affects the learned map $T$ below.

We pick **CycleGAN** [72] as a base model for evaluation since it is known as one of the principal models to solve the unpaired translation problem. We use $c = \ell^1$ as the similarity loss as the CycleGAN's authors **originally used in their paper**.[7] We consider parameter $\lambda \in [0, 50, 100, 200, 250, 300, 350, 500]$.

---

[7]We also conducted a separate experiment to train CycleGAN with $\ell^2$ identity loss. However, in this case CycleGAN's training turned to be less stable and, importantly, yielded (mostly) worse FID. Surprisingly, we

Additionally, we perform comparison with a more recent StarGAN-v2 [15] model. By default, StarGAN-v2 does not use any similarity loss and does not enforce the output to be similar to the input. Therefore, analogously to CycleGAN, we endow the model with an additional $\ell^1$ similarity loss and consider $\lambda \in [0, 1, 10, 50, 100, 200, 500]$.

We train both models on *celeba → anime* ($64 \times 64$) and *handbags → shoes* ($128 \times 128$) translation with various $\lambda$ and report the qualitative and quantitative results below. In all the cases, we report both $\ell^2$ and $\ell^1$ transport costs and FID on the **test** samples. Tables 2, 3, 4, 5 show transport costs and FID of GANs. FID and $\ell^2$ metrics for our method are given in the main text (Tables 1b, 1a) and $\ell^1$ cost is given in Table 6 below. For convenience, we visualize (FID, Cost) pairs for our method and GANs in Fig. 14.

**Results and discussion (CycleGAN).** Interestingly, we see that for CycleGAN adding small identity loss $\lambda = 50$ yields not only decrease of the transport cost (compared to $\lambda = 0$), but some improvement of FID as well. Still we see that the transport cost in CycleGAN naturally decreases with the increase of weight $\lambda$. Unfortunately, this decrease is accompanied by the **decrease** of the visual image quality, see Fig. 15. While for large $\lambda$ the cost for CycleGAN is really small, the model is **practically useless** since it poors image quality. For very large $\lambda$, CycleGAN simply learn the identity map, as expected.

For $\lambda$ *providing acceptable visual quality*, CycleGAN yields a transport cost which is bigger than that of **standard** OT ($w = 1$). Our result for $w = 8$ is unachievable for it. Note that in most cases FID of our IT method is smaller than that of CycleGAN.

**Results and discussion (StarGAN-v2).** In the *celeba→anime* experiment, StarGAN-v2 results are similar to CycleGAN ones. Our IT with $w = 2$ easily provides smaller transport cost (better similarity) than StarGAN-v2. In the *handbags → shoes* translation, we have encountered surprising observations. We see that starting from $\lambda = 10$ the model **fails** to translate some of the handbags to shoes, i.e., these handbags remain nearly unchanged. We notice the similar behaviour for vanilla GAN in *'Accept'* experiment, see Fig. 13a. This explains the low cost for StarGAN-v2 model ($\lambda \geq 10$). Surprisingly to us, for $\lambda = 10$, FID metric is also low despite the fact that model frequently produces **failures**, see the highlighted results in Fig. 16. Note that while FID is a widely used metric, it still could produce misleading estimations which we observe in the latter case.

Additionally, we provide a large set of randomly generated images for $\lambda = 10$ to qualitatively show that the stated issue is indeed notable, see Fig. 17. These failures demonstrate the *limited practical usage* of the model. Our IT method does not suffer from this issue which we qualitatively demonstrate on the same set of images for different weights $w$, see Fig. 25.

| Experiment | Cost | $\lambda = 0$ | $\lambda = 50$ | $\lambda = 100$ | $\lambda = 200$ | $\lambda = 250$ | $\lambda = 300$ | $\lambda = 350$ | $\lambda = 500$ |
|---|---|---|---|---|---|---|---|---|---|
| *celeba → anime* | $\ell^1$ | 0.48 | 0.48 | 0.39 | 0.26 | 0.09 | 0.09 | 0.09 | 0.09 |
| *handbag → shoes* | | 0.42 | 0.36 | 0.34 | 0.31 | 0.32 | 0.22 | 0.16 | 0.07 |
| *celeba → anime* | $\ell^2$ | 0.33 | 0.32 | 0.24 | 0.11 | 0.01 | 0.02 | 0.02 | 0.01 |
| *handbag → shoes* | | 0.51 | 0.43 | 0.41 | 0.35 | 0.37 | 0.22 | 0.14 | 0.02 |

Table 2: Test $\ell^1$ and $\ell^2$ transport cost of CycleGAN.

| Experiment | $\lambda = 0$ | $\lambda = 50$ | $\lambda = 100$ | $\lambda = 200$ | $\lambda = 250$ | $\lambda = 300$ | $\lambda = 350$ | $\lambda = 500$ |
|---|---|---|---|---|---|---|---|---|
| *celeba → anime* | 22.9 | 20.8 | 35.2 | 88.8 | 122.2 | 123.5 | 120.0 | 122.8 |
| *handbag → shoes* | 27.8 | 23.4 | 23.6 | 37.4 | 38.5 | 105.6 | 144.9 | 152.9 |

Table 3: Test FID of CycleGAN.

| Experiment | Cost | $\lambda = 0$ | $\lambda = 1$ | $\lambda = 10$ | $\lambda = 50$ | $\lambda = 100$ | $\lambda = 200$ | $\lambda = 500$ |
|---|---|---|---|---|---|---|---|---|
| *celeba → anime* | $\ell^1$ | 0.672 | 0.355 | 0.210 | 0.076 | 0.050 | 0.030 | 0.029 |
| *handbag → shoes* | | 0.562 | 0.465 | 0.244 | 0.087 | 0.068 | 0.054 | 0.048 |
| *celeba → anime* | $\ell^2$ | 0.686 | 0.216 | 0.094 | 0.017 | 0.006 | 0.002 | 0.002 |
| *handbag → shoes* | | 0.739 | 0.584 | 0.244 | 0.040 | 0.023 | 0.015 | 0.012 |

Table 4: Test $\ell^1$ and $\ell^2$ transport costs of StarGAN-v2.

| Experiment | $\lambda = 0$ | $\lambda = 1$ | $\lambda = 10$ | $\lambda = 50$ | $\lambda = 100$ | $\lambda = 200$ | $\lambda = 500$ |
|---|---|---|---|---|---|---|---|
| *celeba → anime* | 19.55 | 22.40 | 42.30 | 99.68 | 123.76 | 137.8 | 139.11 |
| *handbag → shoes* | 25.45 | 45.13 | 22.36 | 131.8 | 149.8 | 155.8 | 158.8 |

Table 5: Test FID of StarGAN-v2.

---

observed higher test transport costs (both $\ell^2$ and $\ell^1$). Therefore, not to overload the exposition, we decided to keep only the experiment with CycleGAN trained with $\ell^1$ identity loss.

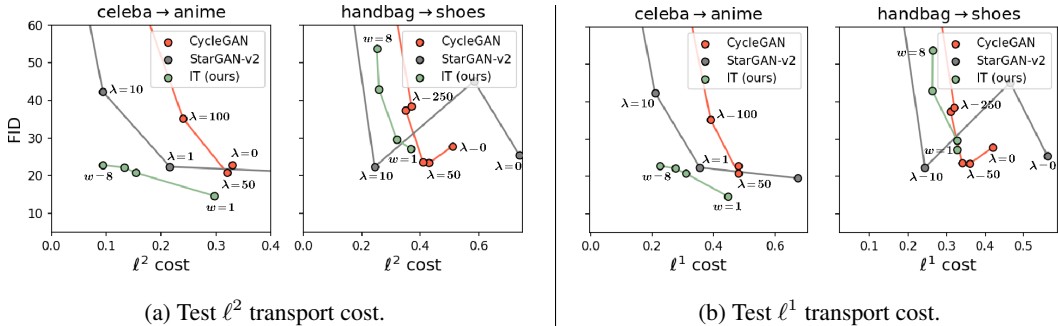

(a) Test $\ell^2$ transport cost.       (b) Test $\ell^1$ transport cost.

Figure 14: Comparison of test FID and transport costs ($\ell^2$ and $\ell^1$) of our IT method and CycleGAN.

| Experiment | $w = 1$ | $w = 2$ | $w = 4$ | $w = 8$ |
|---|---|---|---|---|
| *celeba → anime* | 0.447 | 0.309 | 0.275 | 0.225 |
| *handbag → shoes* | 0.327 | 0.328 | 0.263 | 0.264 |

Table 6: Test $\ell^1$ transport cost of our IT maps (learned with $\ell^2$ transport cost).

**Concluding remarks.** Our algorithm with $w \to \infty$ allows us to achieve better similarity without the decrease of the image quality. At the same time, GANs fail to do this when $\lambda \to \infty$. Why?

We again emphasize that typical GAN objective (17) consists of several loss terms. Each term stimulates the model to attain certain properties (realism, similarity to the input, etc.). These terms, in general, *contradict* each other as they have different minima $T$. This yields the nasty **trade-off** between the loss terms. Our analysis shows that conceptually there is no significant difference between CycleGAN and a more recent StarGAN-v2 model. More generally, any GAN-based method inherits *realism-similarity* tradeoff issue. GANs' differences are mostly related to the use of other architectures or additional losses. Therefore, we think that additional comparisons are excessive since they may not provide any new insights.

In contrast, **our method is not a sum of losses**. Our objective (15) may look like a direct sum of a transport cost with an adversarial loss; our method does have a generator (transport maps $T$) and a discriminator (potential $f$) which are trained via the saddle-point optimization objective $\max_f \min_T$. Yet, this visible **similarity to GAN-based methods is deceptive**. Our objective $\max_f \min_T$ can be viewed as a Lagrangian and is **atypical** for GANs: the generator is in the inner optimization problem $\min_T$ while in GANs the objective is $\min_T \max_f$. In our case, similar to other neural dual OT methods, the generator is adversarial to the discriminator but not vice versa, as in GANs. Please consult [43, §4.3], [24] or [20] for further discussion about OT methods.

GANs aim to balance loss terms $\mathcal{L}_{\text{Dom}}$ and $\mathcal{L}_{\text{Sim}}$. Our optimization objective enforces the constraint $T \sharp \mathbb{P} \le w \mathbb{Q}$ via learning the potential $f$ (a.k.a. Lagrange multiplier) and among admissible maps $T$ searches for the one providing the smallest transport cost. There is no realism-similarity trade-off. For completeness, we emphasize that when $w \to \infty$, FID in Table 1b does not drop because of the decrease of the image quality, but because our method covers the less part of $\mathbb{Q}$. FID negatively reacts to this [48, Fig. 1b].

## D    Relation and Comparison with Discrete Partial OT Methods

The goal of domain translation is to recover the map $x \mapsto T(x)$ between two domains $\mathbb{P}$, $\mathbb{Q}$. We approach this problem by approximating $T$ with a neural network trained on the empirical samples $X = \{x_1, \dots, x_N\}$, $Y = \{y_1, \dots, y_M\}$, i.e., train datasets. Our method *generalizes* to new (previously unseen, test) input samples $x_{new} \sim \mathbb{P}$, i.e, our learned map $\widehat{T}$ can be applied to new input samples to *generate* new target samples $\widehat{T}(x_{new})$.

In contrast, discrete OT methods (including discrete partial OT) *perform a matching* between the empirical distributions $\widehat{\mathbb{P}}_N = \sum_{n=1}^{N} \delta_{x_n}$, $\widehat{\mathbb{Q}}_M = \sum_{m=1}^{M} \delta_{y_m}$. Thus, they do not provide out-of-sample estimation of the transport plan $\pi(y|x_{new})$ or map $T(x_{new})$. The reader may naturally wonder: why not to **interpolate** the solutions of discrete OT? For example, a common strategy is to derive barycentric projections $\overline{T}(x) = \int_{\mathcal{Y}} y d\pi(y|x)$ of the discrete OT plan $\pi$ and then learn a network $T_\theta(x) \approx \overline{T}(x)$ to approximate them [63]. Below we study this approach and show that it does not provide reasonable performance.

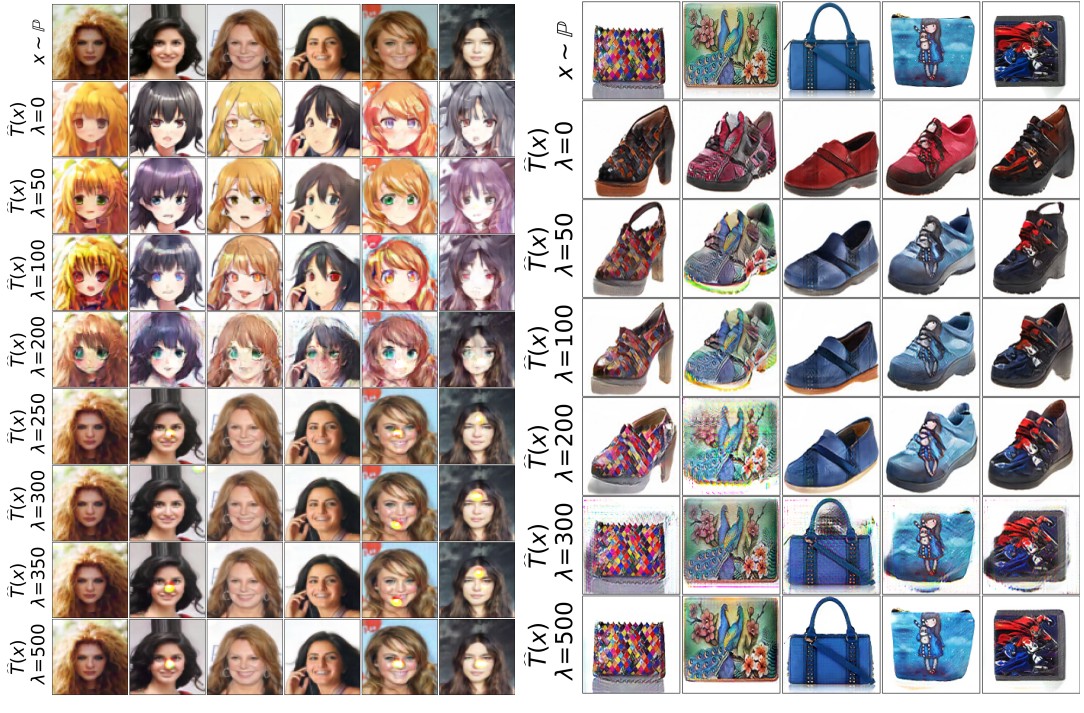

(a) *Celeba* (female) → *anime* (64×64).

(b) *Handbags* → *shoes* (128×128).

Figure 15: Unpaired translation via CycleGAN endowed with $\ell^1$ identity loss with various weights $\lambda$.

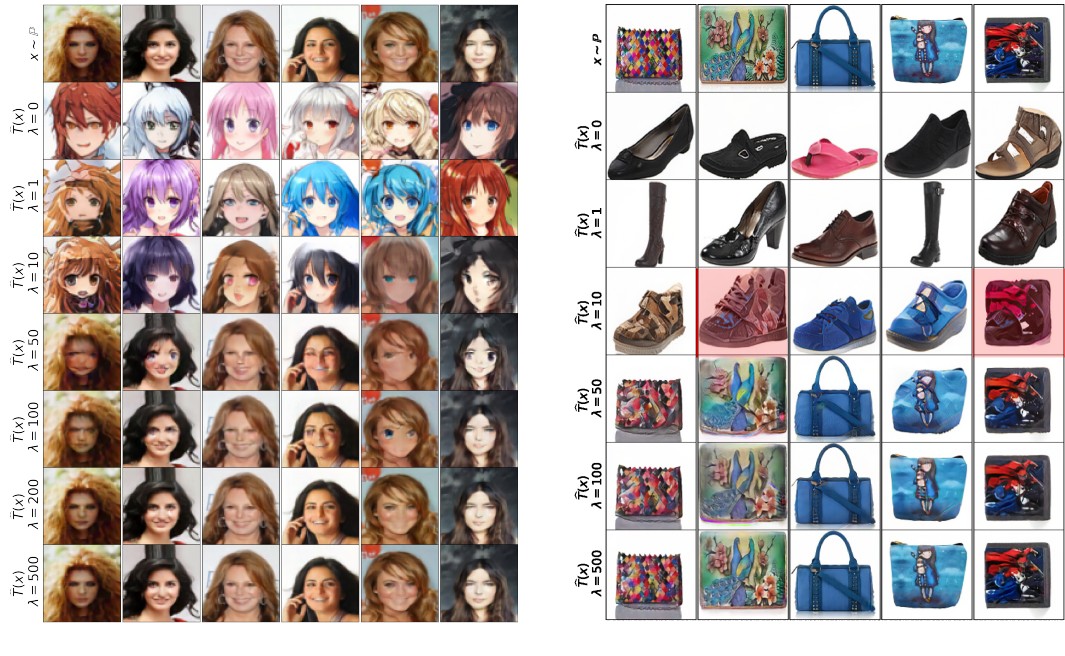

(a) *Celeba* (female) → *anime* (64×64).

(b) *Handbags*→ *shoes* (128×128).

Figure 16: Unpaired translation via StarGAN-v2 endowed with $\ell^1$ identity loss with various weights $\lambda$.

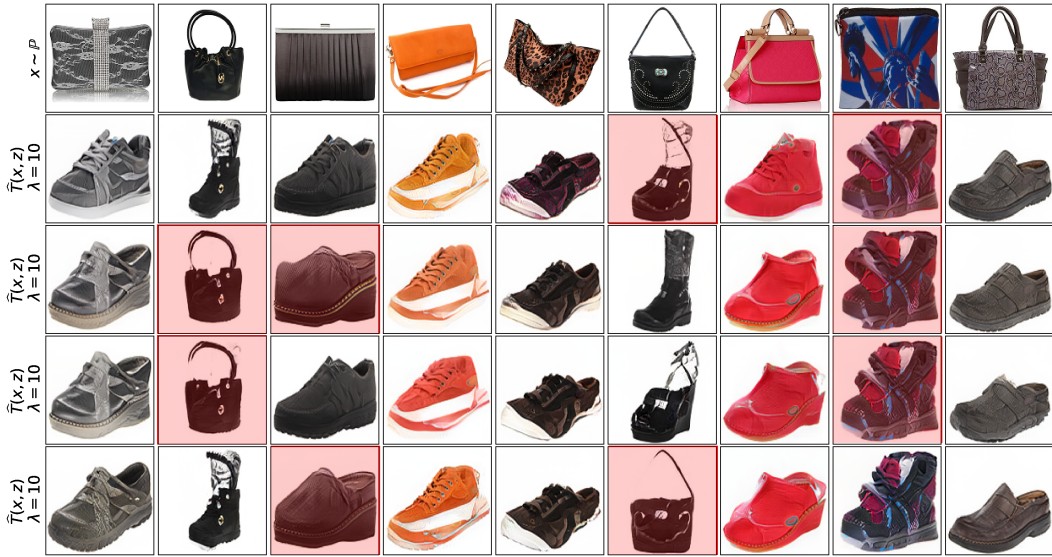

Figure 17: Unpaired translation of *handbags* to *shoes* (128×128) via StarGAN-v2 endowed with $\ell^1$ identity loss ($\lambda = 10$). Since StarGAN-v2 is a stochastic (one-to-many) approach, we visualize several generated samples for different noise vectors $z$. Failures (poorly translated handbags which remain nearly unchanged) are highlighted with **red**.

We perform evaluation of the BP approach on *celeba→anime* translation. We solve the discrete partial OT (DOT) between the parts of the train data $X$ (11K samples), $Y$ (50K samples).[8] We use `ot.partial.partial_wasserstein` algorithm with parameters $w_0 = m = 1$ and vary $w_1 \in \{1, 2, 4, 8\}$, see the notation in (4). This corresponds to our IT (12) with $w = w_1$. Then we regress a UNet $T_\theta$ to recover the barycentric projection $\overline{T}$. We also simulate the case of $w = \infty$ by learning the discrete NNs in the train dataset using `NearestNeighbors` algorithm from `sklearn.neighbors`. We experiment with using MSE or VGG-based perceptual error[9] as the loss function for regression. We visualize the obtained results in Fig. 18 and report average $\ell^2$ **test** transport cost, FID in Table 7.

BP methods are known **not to work well** in large scale problems such as unpaired translation because of the *averaging effect*, see Figure 3 and large FID values of BP in Table 1 of [17]. Indeed, one may learn a barycentric projection network $T_\theta(x) \approx \int_\mathcal{Y} y\, d\pi(y|x)$ over the discrete OT plan $\pi$,

| Metrics | $w = 1$ | $w = 2$ | $w = 4$ | $w = 8$ | $w = \infty$ | $w = \infty$ (perc.) |
|---|---|---|---|---|---|---|
| $\ell^2$ cost | 0.298 | 0.199 | 0.184 | 0.167 | 0.158 | 0.165 |
| FID | 185.35 | 137.53 | 82.67 | 82.19 | 82.24 | 82.45 |

Table 7: Test $\ell^2$ cost, FID of the DOT+BP method trained with $\ell^2$ or perceptual loss function.

yet it is clear that $\int_\mathcal{Y} y\, d\pi(y|x)$ is a direct weighted *average* of several images $y$, i.e., some blurry *average* image of poor quality. It is not satisfactory for practical applications.

Our qualitative results indeed show that BP approach for small $w$ leads to the *averaging effect*. We see that on the train dataset this effect disappears with the increase of $w$, see Fig. 18b. It is expected, since with the increase of weight the conditional distribution of a plan $\pi(y|x)$ tends to a degenerate distribution concentrated at the nearest neighbor $NN(x)$ of $x$ in the train dataset, i.e., $\pi(y|x) \approx \delta_{NN(x)}$ when $w \to \infty$. This means, that in the limit the barycentric projection in point $x$ is its nearest neighbor $NN(x)$.

However, the learned network ($w = \infty$) does not generalize well to unseen test samples and produces images of insufficient quality which is much worse than for the train samples, see Fig. 18a. Despite the fact that test $\ell^2$ cost and FID decrease with the increase of $w$, see

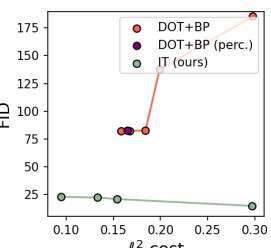

Figure 19: Test FID, $\ell^2$ cost of our IT vs. DOT+BP.

---

[8] We do not use the whole datasets since computing discrete OT between them is computationally infeasible.

[9] `github.com/iamalexkorotin/WassersteinIterativeNetworks/src/losses.py`

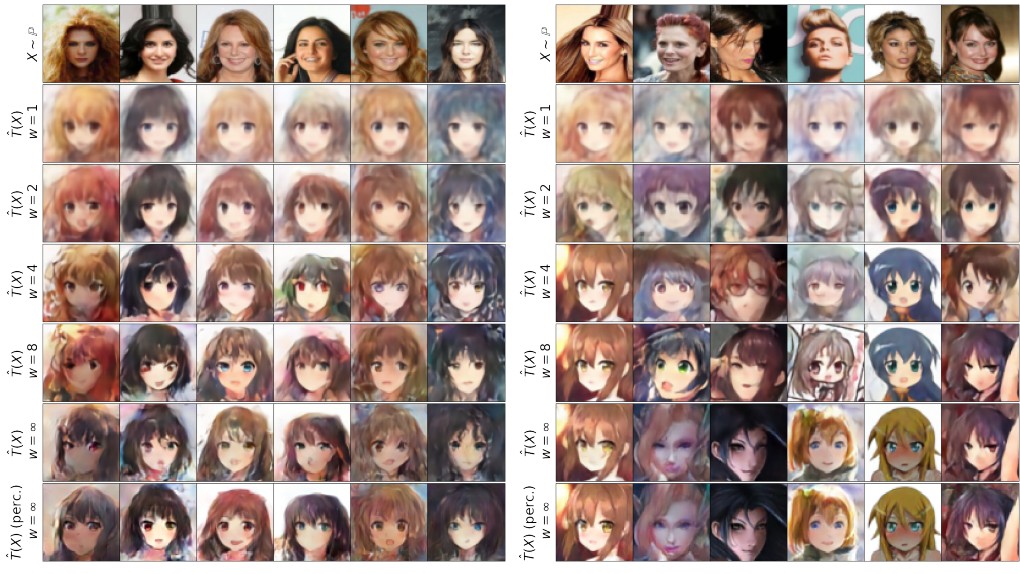

(a) Results on *test* set.  (b) Results on *train* set.

Figure 18: Unpaired translation with DOT+BP in *celeba→ anime* experiment visualized on test, train partitions.

Table 7, Fig. 19, generated test images have poor quality for all weights $w$. Yet, the FID scores as well as the $\ell^2$ costs are much bigger than in our method in all the cases.

As the MSE error for regression is known not to work well in image tasks, we also conduct an experiment using the perceptual error function to learn $T_\theta$. Unfortunately, training the neural network with perceptual error does not lead to meaningful improvements, see the bottom lines of Fig. 18b, 18a, Table 7. This confirms that the origins of DOT+BP method's poor performance lie not in the regression part, but in the entire methodology based on barycentric projections.

To conclude, discrete OT methods are not competitors to our work as there is no straightforward and well-performing way to make out-of-sample DOT estimation in large scale image processing tasks.

## E  Experimental Details

**Pre-processing.** For all datasets, we rescale images' RGB channels to [-1, 1]. As in [40], we rescale aligned anime face images to 512×512. Then we do 256×256 crop with the center located 14 pixels above the image center to get the face. Finally, for all datasets except the textures dataset, we resize the images to the required size (64×64 or 128×128). We apply random horizontal flipping augmentation to the comic faces and chairs datasets. Analogously to [42], we rescale describable textures to minimal border size of 300, do the random resized crop (from 128 to 300 pixels) and random horizontal, vertical flips. Next, we resize images to the required size (64 × 64 or 128 × 128).

**Neural networks.** In §5.1, we use fully connected networks both for the mapping $T_\theta$ and potential $f_\psi$. In §5.2, we use UNet [60] architecture for the transport map $T_\theta$. We use WGAN-QC's [45] ResNet [30] discriminator as a potential $f_\psi$. We add an additional final layer $x \mapsto -|x|$ to $f_\psi$ to make its outputs non-positive.

**Optimization.** We employ Adam [38] optimizer with the default betas both for $T_\theta$ and $f_\psi$. The learning rate is $lr = 10^{-4}$. We use the MultiStepLR scheduler which decreases $lr$ by 2 after $[(5+5\cdot w)K, (20+5\cdot w)K, (40+5\cdot w)K, (70+5\cdot w)K]$ iterations of $f_\psi$ where $w \in \{1, 2, 4, 8\}$ is a weight parameter. The batch size is $|X| = 256$ for toy *'Wi-Fi'*, $|X| = 4096$ for toy Accept, and $|X| = 64$ for image-to-image translation experiments. The number of inner $T_\theta$ iterations is $k_T = 10$. In toy experiments, we observe convergence in $\approx$ 30K total iterations of $f_\psi$ for *'Wi-Fi'*, in $\approx$ 70K for Accept. In the image-to-image translation, we do $\approx$ 70K iterations for 128 ×128 datasets, $\approx$ 40K iterations for 64×64 datasets. In the experiments with image-to-image translation experiments, we gradually increase $w$ for 20K first iterations of $f_\psi$. We start from $w = 1$ and linearly increase it to the desired $w$ (2, 4 or 8).

**Computational complexity**. The complexity of training IT maps depends on the dataset, size of images and weight $w$. Convergence time increases with the increase of $w$: possible reasons for this are discussed in the *unused samples* limitation (Appendix A). In general, it takes from 2 (for $w = 1$) up to 6 (for $w = 8$) days on a single Tesla V100 GPU.

**Reproducibility.** We provide the code for the experiments and will provide the trained models, see `README.md`.

## F  Proofs

*Proof of Proposition 1.* Since continuous $c$ is defined on a compact set $\mathcal{X} \times \mathcal{Y}$, it is uniformly continuous on $\mathcal{X} \times \mathcal{Y}$. This means that there exists a modulus of continuity $\omega : [0, +\infty) \to [0, \infty)$ such that for all $(x, y), (x', y') \in \mathcal{X} \times \mathcal{Y}$ it holds

$$|c(x, y) - c(x', y')| \le \omega(||x - x'||_{\mathcal{X}} + ||y - y'||_{\mathcal{Y}}),$$

and function $\omega$ is monotone, continuous at 0 with $\omega(0) = 0$. In particular, for $y = y'$ we have $|c(x, y) - c(x', y)| \le \omega(||x - x'||_{\mathcal{X}})$. Thus, for $c^*(x) = \min_{y \in \text{Supp}(\mathbb{Q})} c(x, y)$, we have $|c^*(x) - c^*(x')| \le \omega(||x - x'||_{\mathcal{X}})$, see [62, Box 1.8]. This means that $c^* : \mathcal{X} \to \mathbb{R}$ is (uniformly) continuous. $\square$

*Proof of Theorem 1.* Since function $(x, y) \mapsto c(x, y)$ is continuous, there exists a measurable selection $T^* : \mathcal{X} \to \mathcal{Y}$ from the set-valued map $x \mapsto \arg\min_{y \in \text{Supp}(\mathbb{Q})} c(x, y)$, see [2, Theorem 18.19]. This map for all $x \in \mathcal{X}$ satisfies $c(x, T(x)) = \min_{y \in \text{Supp}(\mathbb{Q})} c(x, y) = c^*(x)$. As a result, $\int_{\mathcal{X}} c(x, T(x)) d\mathbb{P}(x) = \int_{\mathcal{X}} c^*(x) d\mathbb{P}(x)$ and (7) is tight. $\square$

**Lemma 1.** *(Distinctness) Let $\mu, \nu \in \mathcal{M}(\mathcal{Y})$. Then $\mu \le \nu$ holds if and only if for every $f \in \mathcal{C}(\mathcal{Y})$ satisfying $f \le 0$ it holds that $\int_{\mathcal{Y}} f(y) d\mu(y) \ge \int_{\mathcal{Y}} f(y) d\nu(y)$.*

*Proof of Lemma 1.* If $\mu \le \nu$, then the inequality $\int_{\mathcal{Y}} f(y) d(\mu - \nu)(y) \ge 0$ for every (measurable) $f \le 0$ follows from the definition of the Lebesgue integral. Below we prove the statement in the other direction.

Assume the opposite, i.e., $\int_{\mathcal{Y}} f(y) d(\mu - \nu)(y) \ge 0$ for every continuous $f \le 0$ but still $\nu \not\ge \mu$. The latter means there exists a measurable $A \subset \mathcal{Y}$ satisfying $\mu(A) > \nu(A)$. Let $\epsilon = \frac{1}{2}(\mu(A) - \nu(A)) > 0$. Consider the negative indicator function $f_A(y)$ which equals $-1$ if $y \in A$ and 0 when $y \notin A$. Consider a variation measure $|\nu - \mu| \in \mathcal{M}_+(\mathcal{Y})$. Thanks to [23, Proposition 7.9], the continuous functions $C(\mathcal{Y})$ are dense in the space $\mathcal{L}^1(|\mu - \nu|)$. Therefore, there exists a function $f_{A,\epsilon} \in \mathcal{C}(\mathcal{Y})$ satisfying $\int_{\mathcal{X}} |f_A(y) - f_{A,\epsilon}(y)| d|\mu - \nu|(y) < \epsilon$. We define $f_{A,\epsilon}^-(y) \stackrel{def}{=} \min\{0, f_{A,\epsilon}(y)\} \le 0$. This is a non-positive continuous function, and for $y \in \mathcal{Y}$ it holds that $|f_A(y) - f_{A,\epsilon}(y)| \ge |f_A(y) - f_{A,\epsilon}^-(y)|$ because $f_A$ takes only non-positive values $\{0, -1\}$. We derive

$$\int_{\mathcal{Y}} f_{A,\epsilon}^-(y) d(\mu - \nu)(y) = \overbrace{\int_{\mathcal{Y}} f_A(y) d(\mu - \nu)(y)}^{=\nu(A) - \mu(A) = -2\epsilon} + \int_{\mathcal{Y}} (f_{A,\epsilon}^-(y) - f_A(y)) d(\mu - \nu)(y) \le$$

$$-2\epsilon + \int_{\mathcal{Y}} |f_{A,\epsilon}^-(y) - f_A(y)| d|\mu - \nu|(y) \le -2\epsilon + \int_{\mathcal{Y}} |f_{A,\epsilon}(y) - f_A(y)| d|\mu - \nu|(y) \le$$

$$-2\epsilon + \epsilon = -\epsilon < 0,$$

which is a contradiction to the fact that $\int_{\mathcal{Y}} f(y) d(\mu - \nu)(y) \ge 0$ for every continuous $f \le 0$. Thus, $\mu \le \nu$. $\square$

*Proof of Proposition 2.* To begin with, we prove that $\Pi^w(\mathbb{P}, \mathbb{Q})$ is a weak-* compact set. Pick any sequence $\pi_n \in \Pi^w(\mathbb{P}, \mathbb{Q})$. It is bounded as all $\pi_n$ are probability measures ($||\pi_n||_1 = 1$). Hence by the Banach-Alaoglu theorem [62, Box 1.2], there exists a subsequence $\pi_{n_k}$ weakly-* converging to some $\pi \in \mathcal{M}(\mathcal{X} \times \mathcal{Y})$. It remains to check that $\pi \in \Pi^w(\mathbb{P}, \mathbb{Q})$. Let $(\mu_{n_k}, \nu_{n_k})$ denote the marginals

of $\pi_{n_k}$ and $(\mu, \nu)$ be the marginals of $\pi$. Pick any $f \in \mathcal{C}(\mathcal{Y})$ with $f \leq 0$. Since $\pi_{n_k} \in \Pi^w(\mathbb{P}, \mathbb{Q})$, it holds that $0 \leq \nu_{n_k} \leq w\mathbb{Q}$, $w \int_{\mathcal{Y}} f(y)d\mathbb{Q}(y) \leq \int_{\mathcal{Y}} f(y)d\nu_{n_k}(y) \leq 0$ (Lemma 1). We have

$$\int_{\mathcal{Y}} f(y)d\nu(y) = \int_{\mathcal{X} \times \mathcal{Y}} f(y)d\pi(x,y) = \lim_{k \to \infty} \int_{\mathcal{X} \times \mathcal{Y}} f(y)d\pi_{n_k}(x,y) = \lim_{k \to \infty} \int_{\mathcal{Y}} f(y)d\nu_{n_k}(y).$$

The latter limit is $\geq w \int_{\mathcal{Y}} f(y)d\mathbb{Q}(y)$ and $\leq 0$. As this holds for every continuous $f \leq 0$, we conclude that $0 \leq \nu \leq w\mathbb{Q}$. By the analogous analysis one may prove that $\mu = \mathbb{P}$ and $\pi \geq 0$. Thus, $\pi \in \Pi^w(\mathbb{P}, \mathbb{Q})$ and $\Pi^w(\mathbb{P}, \mathbb{Q})$ is a weak-* compact.

The functional $\pi \mapsto \int_{\mathcal{X} \times \mathcal{Y}} c(x,y)d\pi(x,y)$ is continuous in the space $\mathcal{M}(\mathcal{X} \times \mathcal{Y})$ equipped with weak-* topology because $c : \mathcal{X} \times \mathcal{Y} \to \mathbb{R}$ is continuous. Since $\Pi^w(\mathbb{P}, \mathbb{Q})$ is a compact set, there exists $\pi^* \in \Pi^w(\mathbb{P}, \mathbb{Q})$ attaining the minimum on $\Pi^w(\mathbb{P}, \mathbb{Q})$. This follows from the Weierstrass extreme value theorem and ends the proof. $\qquad\square$

**Bibliographical remark.** The results showing the existence of minimizers $\pi^*$ in partial OT (4) already exist, see [10, Lemma 2.2] or [22, §2]. They also provide the existence of minimizers in our IT problem (12). Yet, the authors study the particular case when $\mathbb{P}, \mathbb{Q}$ have densities on $\mathcal{X}, \mathcal{Y} \subset \mathbb{R}^D$. *For completeness*, we include a separate proof of existence as we do not require the absolute continuity assumption. The proof is performed via the usual technique based on weak-* compactness in dual spaces and is analogous to [62, Theorem 1.4] which proves the existence of minimizers for OT problem (2). Our proof is slightly more technical due to the inequality constraint.

*Proof of Proposition 3.* Let $\pi^* \in \Pi^w(\mathbb{P}, \mathbb{Q})$ be an IT plan. Consider the OT problem between $\mathbb{P}$ and $\pi_y^*$:

$$\min_{\pi \in \Pi(\mathbb{P}, \pi_y^*)} \int_{\mathcal{X} \times \mathcal{Y}} c(x,y)d\pi(x,y). \tag{18}$$

It turns out that $\pi^*$ is a minimizer here. Assume the contrary, i.e., that there exists a more optimal $\pi' \in \Pi(\mathbb{P}, \pi_y^*)$ satisfying

$$\int_{\mathcal{X} \times \mathcal{Y}} c(x,y)d\pi'(x,y) < \int_{\mathcal{X} \times \mathcal{Y}} c(x,y)d\pi^*(x,y) \qquad \left(= \text{Cost}_w(\mathbb{P}, \mathbb{Q})\right). \tag{19}$$

This plan by the definition of $\Pi(\mathbb{P}, \pi_y^*)$ satisfies $\pi_x' = \mathbb{P}$ and $\pi_y' = \pi_y^* \leq w\mathbb{Q}$, i.e., $\pi' \in \Pi^w(\mathbb{P}, \mathbb{Q})$. However, (19) contradicts the fact that $\pi^*$ is an IT plan in (18) as $\pi'$ provides smaller cost. Thus, $\min$ in (18) equals $\text{Cost}_w(\mathbb{P}, \mathbb{Q})$ in (12).

Thanks to [62, Theorem 1.33], problem (18) has the same minimal value as the $\inf$ in the Monge's problem

$$\inf_{T \sharp \mathbb{P} = \pi_y^*} \int_{\mathcal{X}} c\big(x, T(x)\big)d\mathbb{P}(x), \tag{20}$$

i.e., for every $\epsilon > 0$ there exists $T_\epsilon : \mathcal{X} \to \mathcal{Y}$ satisfying $T_\epsilon \sharp \mathbb{P} = \pi_y^*$ and $\int_{\mathcal{X}} c\big(x, T_\epsilon(x)\big)d\mathbb{P}(x) < \text{Cost}_w(\mathbb{P}, \mathbb{Q}) + \epsilon$. It remains to substitute this $T_\epsilon$ to Monge's IT problem (11) to get an $\epsilon$-close transport cost to Kantorovich's IT cost (12). As this works for every $\epsilon > 0$, we conclude that $\min$ in (12) is the same as $\inf$ in (11). $\qquad\square$

*Proof of Theorem 2.* The fact that $w \mapsto \text{Cost}_w(\mathbb{P}, \mathbb{Q})$ is non-increasing follows from the inclusion $\Pi^{w_1}(\mathbb{P}, \mathbb{Q}) \subset \Pi^{w_2}(\mathbb{P}, \mathbb{Q})$ for $w_1 \leq w_2$. This inclusion means that for larger values of $w$, the minimization in (12) is performed over a larger set of admissible plans. As for convexity, take any IT plans $\pi^{w_1} \in \Pi^{w_1}(\mathbb{P}, \mathbb{Q})$, $\pi^{w_2} \in \Pi^{w_2}(\mathbb{P}, \mathbb{Q})$ for $w_1, w_2$, respectively. For any $\alpha \in [0, 1]$ consider the mixture $\pi' = \alpha\pi^{w_1} + (1-\alpha)\pi^{w_2}$. Note that $\pi_y' = \alpha\pi_y^{w_1} + (1-\alpha)\pi_y^{w_2} \leq \alpha w_1\mathbb{Q} + (1-\alpha)w_2\mathbb{Q} = \big(\alpha w_1 + (1-\alpha)w_2\big)\mathbb{Q}$. Therefore, $\pi' \in \Pi^{\alpha w_1 + (1-\alpha)w_2}(\mathbb{P}, \mathbb{Q})$. We derive

$$\text{Cost}_{\alpha w_1 + (1-\alpha)w_2}(\mathbb{P}, \mathbb{Q}) \leq \int_{\mathcal{X} \times \mathcal{Y}} c(x,y)d\pi'(x,y) =$$

$$\alpha \int_{\mathcal{X} \times \mathcal{Y}} c(x,y)d\pi^{w_1}(x,y) + (1-\alpha) \int_{\mathcal{X} \times \mathcal{Y}} c(x,y)d\pi^{w_2}(x,y) =$$

$$\alpha\text{Cost}_{w_1}(\mathbb{P}, \mathbb{Q}) + (1-\alpha)\text{Cost}_{w_2}(\mathbb{P}, \mathbb{Q}), \tag{21}$$

which shows the convexity of $w \mapsto \mathrm{Cost}_w(\mathbb{P}, \mathbb{Q})$.

Now we prove that $\lim_{w\to\infty} \mathrm{Cost}_w(\mathbb{P}, \mathbb{Q}) = \mathrm{Cost}_\infty(\mathbb{P}, \mathbb{Q})$. For every $w \geq 1$ and $\pi \in \Pi^w(\mathbb{P}, \mathbb{Q})$, it holds that $\pi_y \leq w\mathbb{Q}$. This means that $\mathrm{Supp}(\pi_y) \subset \mathrm{Supp}(\mathbb{Q})$. As a result, we see that $\Pi^w(\mathbb{P}, \mathbb{Q}) \subset \Pi^\infty(\mathbb{P}, \mathbb{Q})$, i.e., $\mathrm{Cost}_w(\mathbb{P}, \mathbb{Q}) \geq \mathrm{Cost}_\infty(\mathbb{P}, \mathbb{Q})$. We already know that $w \mapsto \mathrm{Cost}_w(\mathbb{P}, \mathbb{Q})$ is non-increasing, so it suffices to show that for every $\epsilon > 0$ there exists $w = w(\epsilon) \in [1, +\infty)$ such that $\mathrm{Cost}_w(\mathbb{P}, \mathbb{Q}) \leq \mathrm{Cost}_\infty(\mathbb{P}, \mathbb{Q}) + \epsilon$. This will provide that $\lim_{w\to\infty} \mathrm{Cost}_w(\mathbb{P}, \mathbb{Q}) = \mathrm{Cost}_\infty(\mathbb{P}, \mathbb{Q})$.

Pick any $\epsilon > 0$. Consider the set $\mathcal{S} \overset{def}{=} \{(x, y) \in \mathcal{X} \times \mathcal{Y}$ such that $y \in \mathrm{NN}(x)\}$. It is a compact set. To see this, we pick any sequence $(x_n, y_n) \in \mathcal{S}$. It is contained in compact $\mathcal{X} \times \mathcal{Y}$. Therefore, it has a sub-sequence $(x_{n_k}, y_{n_k})$ converging to some $(x, y) \in \mathcal{X} \times \mathcal{Y}$. It remains to check that $(x, y) \in \mathcal{S}$. Since $\mathrm{Supp}(\mathbb{Q})$ is compact and $y_{n_k} \in \mathrm{NN}(x_{n_k}) \subset \mathrm{Supp}(\mathbb{Q})$, we have $y \in \mathrm{Supp}(\mathbb{Q})$ as well. At the same time, by the continuity of $c^*$ (Proposition 1) and $c$, we have

$$c(x, y) - c^*(x) = \lim_{k\to\infty} \{c(x_{n_k}, y_{n_k}) - c^*(x_{n_k})\} = \lim_{k\to\infty} 0 = 0,$$

which means that $y \in \mathrm{NN}(x)$ and $(x, y) \in \mathcal{S}$, i.e., $\mathcal{S}$ is compact.

Since $(x, y) \mapsto c(x, y) - c^*(x)$ is a continuous function, for each $(x, y) \in \mathcal{S}$ there exists an open neighborhood $U_x \times V_y \subset \mathcal{X} \times \mathcal{Y}$ of $(x, y)$ such that for all $(x', y') \in U_x \times V_y$ it holds that $c(x', y') - c^*(x') < \epsilon$ or, equivalently, $c(x', y') < c^*(x') + \epsilon$. Since $\bigcup_{(x,y)\in\mathcal{S}} U_x \times V_y$ is an open coverage of the compact set $\mathcal{S}$, there exists a finite sub-coverage $\bigcup_{n=1}^N U_{x_n} \times V_{y_n}$ of $\mathcal{S}$. In particular, $\mathcal{X} = \bigcup_{n=1}^N U_{x_n}$. For convenience, we simplify the notation and put $U_n \overset{def}{=} U_{x_n}$ and $V_n \overset{def}{=} V_{y_n}$. Now we put $U_1' \overset{def}{=} U_1$ and iteratively define $U_n' \overset{def}{=} U_n \setminus U_{n-1}'$ for $n \geq 2$. By the construction, it holds that the entire space $\mathcal{X}$ is a disjoint union of $U_n'$, i.e., $\mathcal{X} = \bigsqcup_{n=1}^N U_n'$. Some of $U_n'$ may be empty, so we just remove them from the sequence and for convenience assume that each $U_n'$ is not empty. Now consider the measure $\pi \in \mathcal{P}(\mathcal{X} \times \mathcal{Y})$ which is given by

$$\pi \overset{def}{=} \sum_{n=1}^N \Big[ \mathbb{P}|_{U_n'} \times \frac{\mathbb{Q}|_{V_n}}{\mathbb{Q}(V_n)} \Big]. \tag{22}$$

Here for $\mu, \nu \in \mathcal{M}_+(\mathcal{X}), \mathcal{M}_+(\mathcal{Y})$, we use $\times$ to denote their product measure $\mu \times \nu \in \mathcal{M}(\mathcal{X} \times \mathcal{Y})$. In turn, for a measurable $A \subset \mathcal{X}$, we use $\mu|_A$ to denote the restriction of $\mu$ to $A$, i.e., measure $\mu' \in \mathcal{M}(\mathcal{X})$ satisfying $\mu'(B) = \mu(A \cap B)$ for every measurable $B \subset \mathcal{X}$. Note that $\sum_{n=1}^N \mathbb{P}|_{U_n'} = \mathbb{P}$ and $\sum_{n=1}^N \mathbb{P}(U_n') = \sum_{n=1}^N \mathbb{P}|_{U_n'}(U_n') = 1$ by the construction of $U_n'$. At the same time, for each $n$ it holds that $\frac{\mathbb{Q}|_{V_n}}{\mathbb{Q}(V_n)}$ is a probability measure because of the normalization $\mathbb{Q}(V_n)$. Note that this normalization is necessarily positive because $V_n$ is a neighborhood of a point in $\mathrm{Supp}(\mathbb{Q})$. Therefore, since sets $U_n'$ are disjoint and cover $\mathcal{X}$, we have $\pi_x = \sum_{n=1}^N \mathbb{P}|_{U_n'} = \mathbb{P}$. Now let us show that there exists $w$ such that $\pi_y \leq w\mathbb{Q}$. It suffices to take $w = \sum_{n=1}^N \frac{\mathbb{P}(U_n')}{\mathbb{Q}(V_n)}$. Indeed, in this case for every measurable $A \subset \mathcal{Y}$ we have

$$\pi_y(A) = \sum_{n=1}^N \mathbb{P}(U_n') \frac{\mathbb{Q}(A \cap V_n)}{\mathbb{Q}(V_n)} \leq \sum_{n=1}^N \mathbb{P}(U_n') \frac{\mathbb{Q}(A)}{\mathbb{Q}(V_n)} \leq \mathbb{Q}(A) \sum_{n=1}^N \frac{\mathbb{P}(U_n')}{\mathbb{Q}(V_n)} \leq w\mathbb{Q}(A),$$

which yields $\pi_y \leq w\mathbb{Q}$ and means that $\pi \in \Pi^w(\mathbb{P}, \mathbb{Q})$ for our chosen $w$. Now let us compute the cost of $\pi$:

$$\int_{\mathcal{X}\times\mathcal{Y}} c(x, y) d\pi(x, y) = \sum_{n=1}^N \int_{U_n'} \Big\{ \frac{1}{\mathbb{Q}(V_n)} \int_{V_n} c(x, y) d\mathbb{Q}|_{V_n}(y) \Big\} d\mathbb{P}|_{U_n'}(x) \leq$$

$$\sum_{n=1}^N \int_{U_n'} \underbrace{\Big\{ \frac{1}{\mathbb{Q}(V_n)} \int_{V_n} \big(c^*(x) + \epsilon\big) d\mathbb{Q}|_{V_n}(y) \Big\}}_{=c^*(x)+\epsilon} d\mathbb{P}|_{U_n'}(x) = \sum_{n=1}^N \int_{U_n'} \big(c^*(x) + \epsilon\big) d\mathbb{P}|_{U_n'}(x) =$$

$$\int_{\mathcal{X}} \big(c^*(x) + \epsilon\big) d\mathbb{P}(x) = \mathrm{Cost}_\infty(\mathbb{P}, \mathbb{Q}) + \epsilon. \tag{23}$$

To finish the proof it remains to note that this plan is not necessarily a minimizer for (12), i.e., $\int_{\mathcal{X}\times\mathcal{Y}} c(x, y) d\pi(x, y)$ is an upper bound on $\mathrm{Cost}_w(\mathbb{P}, \mathbb{Q})$. Therefore, we have $\mathrm{Cost}_w(\mathbb{P}, \mathbb{Q}) \leq \mathrm{Cost}_\infty(\mathbb{P}, \mathbb{Q}) + \epsilon$ for our chosen $w = w(\epsilon)$. $\square$

**Bibliographical remark.** There exist seemingly related **but actually different results** in the fundamental OT literature, see [22, Lemma 2.1] or [10, §3]. There the authors study partial OT problem (4) and study how the partial OT plan and OT cost evolve when the marginals $w_0\mathbb{P}$ and $w_1\mathbb{Q}$ are fixed and the required mass amount to transport changes from 0 to $\min\{w_0, w_1\}$. In our study, the first marginal $w_0\mathbb{P}$ and the amount of mass to transport $m$ are fixed ($w_0 = m = 1$), and we study how the OT cost changes when $w_1 \to \infty$ in the IT problem.

*Proof of Theorem 3.* Note that $\mathcal{P}(\mathcal{X} \times \mathcal{Y})$ is (weak-*) compact. This can be derived from the Banach-Alaoglu theorem analogously to the compactness of $\Pi^w(\mathbb{P}, \mathbb{Q})$ in the proof of Theorem (2). Therefore, *any* sequence in $\mathcal{P}(\mathcal{X} \times \mathcal{Y})$ has a converging sub-sequence. In our case, for brevity, we assume that $\pi^{w_n} \in \Pi^{w_n}(\mathbb{P}, \mathbb{Q})$ itself weakly-* converges to some $\pi^* \subset \mathcal{P}(\mathcal{X} \times \mathcal{Y})$. Since $\pi_x^{w_n} = \mathbb{P}$ for all $n$, we also have $\pi_x^* = \mathbb{P}$. As $\lim_{n\to\infty} w_n = \infty$, we conclude from Theorem 2 that

$$\text{Cost}_\infty(\mathbb{P}, \mathbb{Q}) = \lim_{n\to\infty} \text{Cost}_{w_n}(\mathbb{P}, \mathbb{Q}) = \lim_{n\to\infty} \int_{\mathcal{X}\times\mathcal{Y}} c(x, y)d\pi^{w_n}(x, y) = \int_{\mathcal{X}\times\mathcal{Y}} c(x, y)d\pi^*(x, y), \tag{24}$$

where the last equality holds since $\pi^{w_n}$ (weakly-*) converges to $\pi^*$. From (24), we see that the cost of $\pi^*$ is perfect and it remains to check that $\text{Supp}(\pi_y^*) \subset \text{Supp}(\mathbb{Q})$. Assume the opposite and pick any $y^* \in \text{Supp}(\pi_y^*)$ such that $y^* \notin \text{Supp}(\mathbb{Q})$. By the definition of the support, there exists $\epsilon > 0$ and a neighborhood $U = \{y \in \mathcal{Y}$ such that $\|y - y^*\|_{\mathcal{Y}} < \epsilon\}$ of $y^*$ satisfying $\pi_y^*(U) > 0$ and $U \cap \text{Supp}(\mathbb{Q}) = \emptyset$. Let $h(y) \stackrel{def}{=} \max\{0, \epsilon - \|y - y^*\|_{\mathcal{Y}}\}$. From $\pi_y^{w_n} \leq w_n\mathbb{Q}$ (for all $n$), it follows that $\text{Supp}(\pi_y^{w_n}) \subset \text{Supp}(\mathbb{Q})$. Therefore, $\pi_y^{w_n}(U) = 0$ for all $n$. Since $\pi^{w_n}$ converges to $\pi^*$, we have

$$\lim_{n\to\infty} \int_{\mathcal{Y}} h(y)d\pi^{w_n}(y) = \int_{\mathcal{Y}} h(y)d\pi^*(y). \tag{25}$$

The left part is zero because $h(y)$ vanishes outside $U$ and $\int_U h(y)d\pi^{w_n}(y) = 0$ as $\pi_y^{w_n}(U) = 0$. The right part equals $\int_U h(y)d\pi^*(y)$ and is positive as $\pi^*(U) > 0$ and $h(y) > 0$ for $y \in U$. This is a contradiction. Therefore, $\text{Supp}(\pi_y^*) \subset \text{Supp}(\mathbb{Q})$. Now we see that $\pi^* \in \Pi^\infty(\mathbb{P}, \mathbb{Q})$ is a perfect plan as its cost matches the perfect cost. $\square$

*Proof of Corollary 1.* Assume the inverse. Then $\exists\varepsilon$ such that $\forall w(\varepsilon)$ $\exists w \geq w(\varepsilon)$ and $\exists$ IT plan $\pi^w \in \Pi_w(\mathbb{P}, \mathbb{Q})$ solving (12) such that $\forall$ ET plan $\pi^*$, it holds that $\mathbb{W}_1(\pi^w, \pi^*) \geq \varepsilon$. This means that there exists an increasing sequence $w_1, w_2, ..., w_n, \cdots \to \infty$ and the corresponding sequence of IT plans $\pi^{w_1}, \pi^{w_2}, ..., \pi^{w_n}, \ldots$ such that for every ET plan $\pi^*$ it holds that $\mathbb{W}_1(\pi^{w_n}, \pi^*) \geq \varepsilon$ for all $n$. At the same time, from Theorem 3, this sequence of plans must have a sub-sequence $\{\pi^{w_{n_k}}\}$ which is (weakly-*) converging to some ET plan $\pi^*$: $\pi^{w_{n_k}} \to \pi^*$. Recall that the convergence in $\mathbb{W}_1$ coicides with the weak-$*$ convergence (for compact $\mathcal{X}, \mathcal{Y}$), see [62, Theorem 5.9]. Hence, the sub-sequence should also converge to $\pi^*$ in $\mathbb{W}_1$ but it is not since $\mathbb{W}_1(\pi^{w_n}, \pi^*) \geq \varepsilon$. This is a contradiction. $\square$

*Proof of Theorem 4.* Let $\Pi(\mathbb{P}) \subset \mathcal{P}(\mathcal{X} \times \mathcal{Y})$ denote the subset of probability measures $\pi \in \mathcal{P}(\mathcal{X} \times \mathcal{Y})$ satisfying $\pi_x = \mathbb{P}$. Consider a functional $I : \Pi(\mathbb{P}) \to \{0, +\infty\}$ defined by $I(\pi) \stackrel{def}{=} \sup_{f\leq 0} \int_{\mathcal{Y}} f(y)d(w\mathbb{Q} - \pi_y)(y)$, where the sup is taken over non-positive $f \in \mathcal{C}(\mathcal{Y})$. From Lemma 1, we have $I(\pi) = 0$ when $\pi \in \Pi^w(\mathbb{P}, \mathbb{Q})$ and $I(\pi) = +\infty$ otherwise. Indeed, if there exists a non-positive function satisfying $\int_{\mathcal{Y}} f(y)d(w\mathbb{Q} - \pi_y)(y) > 0$, then function $Cf$ (for $C > 0$) also satisfies this condition and provides $C$-times bigger value which tends to $\infty$ with $C \to \infty$. We use $I(\pi)$ incorporate the right constraint $\pi_y \leq w\mathbb{Q}$ in $\pi \in \Pi^w(\mathbb{P}, \mathbb{Q})$ to the objective and obtain the equivalent to (12) problem:

$$\min_{\pi\in\Pi^w(\mathbb{P},\mathbb{Q})} \int_{\mathcal{X}\times\mathcal{Y}} c(x, y)d\pi(x, y) = \min_{\pi\in\Pi(\mathbb{P})} \left\{ \int_{\mathcal{X}\times\mathcal{Y}} c(x, y)d\pi(x, y) + I(\pi) \right\} =$$

$$\min_{\pi\in\Pi(\mathbb{P})} \left\{ \int_{\mathcal{X}\times\mathcal{Y}} c(x, y)d\pi(x, y) + \sup_{f\leq 0} \int_{\mathcal{Y}} f(y)d(w\mathbb{Q} - \pi_y)(y) \right\} =$$

$$\min_{\pi\in\Pi(\mathbb{P})} \sup_{f\leq 0} \left\{ \int_{\mathcal{X}\times\mathcal{Y}} c(x, y)d\pi(x, y) + \int_{\mathcal{Y}} f(y)d(w\mathbb{Q} - \pi_y)(y) \right\} = \tag{26}$$

$$\sup_{f \leq 0} \min_{\pi \in \Pi(\mathbb{P})} \left\{ \int_{\mathcal{X} \times \mathcal{Y}} c(x,y)d\pi(x,y) + \int_{\mathcal{Y}} f(y)d(w\mathbb{Q} - \pi_y)(y) \right\} = \quad (27)$$

$$\sup_{f \leq 0} \left\{ \min_{\pi \in \Pi(\mathbb{P})} \left\{ \int_{\mathcal{X} \times \mathcal{Y}} c(x,y)d\pi(x,y) - \int_{\mathcal{Y}} f(y)d\pi_y(y) \right\} + w \int_{\mathcal{Y}} f(y)d\mathbb{Q}(y) \right\} =$$

$$\sup_{f \leq 0} \left\{ \min_{\pi \in \Pi(\mathbb{P})} \left\{ \int_{\mathcal{X} \times \mathcal{Y}} c(x,y)d\pi(x,y) - \int_{\mathcal{X} \times \mathcal{Y}} f(y)d\pi(x,y) \right\} + w \int_{\mathcal{Y}} f(y)d\mathbb{Q}(y) \right\} = \quad (28)$$

$$\sup_{f \leq 0} \left\{ \min_{\pi \in \Pi(\mathbb{P})} \left\{ \int_{\mathcal{X}} \int_{\mathcal{Y}} c(x,y)d\pi(y|x) \underbrace{d\mathbb{P}(x)}_{=d\pi_x(x)} - \int_{\mathcal{X}} \int_{\mathcal{Y}} f(y)d\pi(y|x) \underbrace{d\mathbb{P}(x)}_{=d\pi_x(x)} \right\} + w \int_{\mathcal{Y}} f(y)d\mathbb{Q}(y) \right\} = \quad (29)$$

$$\sup_{f \leq 0} \left\{ \min_{\pi \in \Pi(\mathbb{P})} \left\{ \int_{\mathcal{X}} \int_{\mathcal{Y}} \big(c(x,y) - f(y)\big)d\pi(y|x)d\mathbb{P}(x) \right\} + w \int_{\mathcal{Y}} f(y)d\mathbb{Q}(y) \right\} \quad (30)$$

In transition from (26) to (27) we use the minimax theorem to swap sup and min [64, Corollary 2]. This is possible because the expression in (26) is a bilinear functional of $(\pi, f)$. Thus, it is convex in $\pi$ and concave in $f$. At the same time, $\Pi(\mathbb{P})$ is a convex and (weak-*) compact set. The latter can be derived analogously to the compactness of $\Pi^w(\mathbb{P}, \mathbb{Q})$ in the proof of Theorem 2. In transition from (28) to (29), we use the measure disintegration theorem to represent $d\pi(x,y)$ as the marginal $d\pi_x(x) = d\mathbb{P}(x)$ and a family of conditional measures $d\pi(y|x)$ on $\mathcal{Y}$. We note that

$$\min_{\pi \in \Pi(\mathbb{P})} \int_{\mathcal{X}} \int_{\mathcal{Y}} \big(c(x,y) - f(y)\big)d\pi(y|x)d\mathbb{P}(x) \big\} \geq \int_{\mathcal{X}} \underbrace{\min_{y \in \mathcal{Y}} \big(c(x,y) - f(y)\big)}_{=f^c(x)} d\mathbb{P}(x). \quad (31)$$

On the other hand, consider the measurable selection $T : \mathcal{X} \to \mathcal{Y}$ for the set-valued map $x \mapsto \arg\min_{y \in \mathcal{Y}} \big(c(x,y) - f(y)\big)$. It exists thanks to [2, Theorem 18.19]. As a result, for the deterministic plan $\pi^T = [\text{id}, T]\sharp\mathbb{P}$, the minimum in (31) is indeed attained. Therefore, (31) is the equality. We combine (30) and (31) and obtain

$$\text{Cost}_w(\mathbb{P}, \mathbb{Q}) = \min_{\pi \in \Pi^w(\mathbb{P}, \mathbb{Q})} \int_{\mathcal{X} \times \mathcal{Y}} c(x,y)d\pi(x,y) = \sup_{f \leq 0} \left\{ \int_{\mathcal{X}} f^c(x)d\mathbb{P}(x) + w \int_{\mathcal{Y}} f(y)d\mathbb{Q}(y) \right\}. \quad (32)$$

It remains to prove that sup in the right part is actually attained at some non-positive $f^* \in \mathcal{C}(\mathcal{Y})$. Let $f_1, f_2, \dots \in \mathcal{C}(\mathcal{Y})$ be a sequence of non-positive functions for which $\lim_{n \to \infty} \left\{ \int_{\mathcal{X}} f_n^c(x)d\mathbb{P}(x) + w \int_{\mathcal{Y}} f_n(y)d\mathbb{Q}(y) \right\} = \text{Cost}_w(\mathbb{P}, \mathbb{Q})$. For $g \in \mathcal{C}(\mathcal{X})$, we define the $(c, -)$-transform $g^{(c,-)}(y) \overset{def}{=} \min_{x \in \mathcal{X}} \big[ \min \big(c(x,y) - g(x)\big), 0 \big] \leq 0$. It yields a (uniformly) continuous non-positive function satisfying $|g^{(c,-)}(y) - g^{(c,-)}(y')| \leq \omega\big(\|y - y'\|_\mathcal{Y}\big)$, where $\omega$ is the modulus of continuity of $c(x,y)$. This statement can be derived analogously to the proof of Proposition 1.

Before going further, let us highlight two important facts which we are going to use below. Consider any $g \in \mathcal{C}(\mathcal{X})$ and $0 \geq h \in \mathcal{C}(\mathcal{Y})$ satisfying $g(x) + h(y) \leq c(x,y)$ for all $(x,y) \in \mathcal{X} \times \mathcal{Y}$. First, from the definition of $(c, -)$-transform, one can see that for all $(x,y) \in \mathcal{X} \times \mathcal{Y}$ it holds that $0 \geq g^{(c,-)} \geq h$ and

$$g(x) + h(y) \leq g(x) + g^{(c,-)}(y) \leq c(x,y), \quad (33)$$

i.e., $(g, g^{(c,-)})$ also satisfies the assumptions of $(g, h)$. Second, from the definition of $c$-transform, it holds that $h^c \geq g$ and

$$g(x) + h(y) \leq h^c(x) + h(y) \leq c(x,y), \quad (34)$$

i.e., the pair $(h^c, h)$ satisfies the same assumptions as $(g, h)$.

Now we get back to our sequence $f_1, f_2, \dots$. For each $n$ and $(x,y) \in \mathcal{X} \times \mathcal{Y}$, we have $f_n^c(x) + f_n(y) \leq c(x,y)$. Next,

$$f_n^c(x) + f_n(y) \leq f_n^c(x) + (f_n^c)^{(c,-)}(y) \leq \big((f_n^c)^{(c,-)}\big)^c(x) + (f_n^c)^{(c,-)}(y) \qquad \big[ \leq c(x,y) \big], \quad (35)$$

where we first used (33) with $(g, h) = (f_n^c, f_n)$ and then used (34) with $(g, h) = (f_n^c, (f_n^c)^{(c,-)})$. In particular, we have $f_n^c \leq ((f_n^c)^{(c,-)})^c$ and $f_n \leq (f_n^c)^{(c,-)}$. We sum these inequalities with weights 1 and $w$, and for all $(x, y) \in \mathcal{X} \times \mathcal{Y}$ obtain

$$f_n^c(x) + w f_n(y) \leq ((f_n^c)^{(c,-)})^c(x) + w(f_n^c)^{(c,-)}(y) = h_n^c(x) + w h_n(y), \tag{36}$$

where for convenience we denote $h_n \overset{def}{=} (f_n^c)^{(c,-)}$. Integrating (36) with $(x, y) \sim \mathbb{P} \times \mathbb{Q}$ yields

$$\int_{\mathcal{X}} f_n^c(x) d\mathbb{P}(x) + w \int_{\mathcal{Y}} f_n(y) d\mathbb{Q}(y) \leq \int_{\mathcal{X}} h_n^c(x) d\mathbb{P}(x) + w \int_{\mathcal{Y}} h_n(y) d\mathbb{Q}(y). \tag{37}$$

This means that potential $h_n$ provides not smaller dual objective value than $f_n$. As a result, sequence $h_1, h_2 \ldots$ also satisfies $\lim_{n \to \infty} \left\{ \int_{\mathcal{X}} h_n^c(x) d\mathbb{P}(x) + w \int_{\mathcal{Y}} h_n(y) d\mathbb{Q}(y) \right\} = \text{Cost}_w(\mathbb{P}, \mathbb{Q})$. Now we forget about $f_1, f_2, \ldots$ and work with $h_1, h_2, \ldots$.

All the functions $h_n$ are uniformly equicontinuous as they share the same modulus of continuity $\omega$ because they are $(c, -)$-transforms by their definition. Let $v_n(y) \overset{def}{=} h_n(y) - \max_{y' \in \mathcal{Y}} h_n(y')$. This function is also non-positive and uniformly continuous as well. Note that $v_n$ provides the same dual objective value as $h_n$. This follows from the definition of $v^c = h^c + \max_{y' \in \mathcal{Y}} h_n(y')$. Here the additive constant vanishes, i.e., $v_n^c(x) + v_n(y) = h_n^c(x) + h_n(y)$. At the same time, $v_n$ are all uniformly bounded. Indeed, let $y_n \in \mathcal{Y}$ be any point where $v_n(y_n) = 0$. Then for all $y \in \mathcal{Y}$ it holds that $|v_n(y)| = |v_n(y) - v_n(y_n)| \leq \omega(\|y - y_n\|_{\mathcal{Y}}) \leq \omega(\text{diam}(\mathcal{Y}))$. Therefore, by the Arzelà–Ascoli theorem, there exists a subsequence $v_{n_k}$ *uniformly* converging to some $f^* \in \mathcal{C}(X)$. As all $v_{n_k} \leq 0$, it holds that $f^* \leq 0$ as well. It remains to check that $f^*$ attains the supremum in (32).

To begin with, we prove that $v_{n_k}^c$ uniformly converges to $(f^*)^c$. Denote $\|v_{n_k} - f^*\|_\infty = \epsilon_k$. For all $(x, y) \in \mathcal{X} \times \mathcal{Y}$, we have

$$c(x, y) - f^*(y) - \epsilon_k \leq c(x, y) - v_{n_k}(y) \leq c(x, y) - f^*(y) + \epsilon_k \tag{38}$$

since $|v_{n_k}(y) - f^*(y)| \leq \|v_{n_k} - f^*\|_\infty < \epsilon$. We take $\min_{y \in \mathcal{Y}}$ in (38) and obtain $(f^*)^c(x) - \epsilon_k \leq v_{n_k}^c(x) \leq (f^*)^c(x) + \epsilon_k$. As this holds for all $x \in \mathcal{X}$, we have just proved that $\|v_{n_k}^c - (f^*)^c\|_\infty < \epsilon_k$. This means that $v_{n_k}^c$ uniformly converges to $(f^*)^c$ as well since $\lim_{k \to \infty} \|v_{n_k} - f^*\|_\infty = \lim_{k \to \infty} \epsilon_k = 0$. Thanks to the uniform convergence, we have

$$\text{Cost}_w(\mathbb{P}, \mathbb{Q}) = \lim_{k \to \infty} \left\{ \int_{\mathcal{X}} (v_{n_k})^c(x) d\mathbb{P}(x) + \int_{\mathcal{Y}} v_{n_k}(y) d\mathbb{Q}(y) \right\} =$$

$$\int_{\mathcal{X}} (f^*)^c(x) d\mathbb{P}(x) + \int_{\mathcal{Y}} f^*(y) d\mathbb{Q}(y). \tag{39}$$

We conclude that $f^*$ is a maximizer of (32) that we seek for. $\qquad\square$

**Bibliographical remark.** There exists a duality formula for partial OT (4), see [10, §2] which can be reduced to duality formula to IT problem (12). However, it is hard to relate the resulting formula with ours (13). We do not know how to derive one formula from the other. More importantly, it is unclear how to turn their formula to the computational algorithm. Our formula provides an opportunity to do this by using the saddle point reformulation of the dual problem which nowadays becomes standard for neural OT, see [43, 20, 61]. We will give further comments after the next proof. The second part of the derivation of our formula (existence of a maximizer $f^*$) is inspired by the [62, Proposition 1.11] which shows the existence of maximizers for standard OT (2).

*Proof of Theorem 5.* By the definition of $f^*$, we have

$$\text{Cost}_w(\mathbb{P}, \mathbb{Q}) =$$

$$\min_{T: \mathcal{X} \to \mathcal{Y}} \mathcal{L}(f^*, T) = \min_{T: \mathcal{X} \to \mathcal{Y}} \int_{\mathcal{X}} \left\{ c(x, T(x)) - f^*(T(x)) \right\} d\mathbb{P}(x) + w \int_{\mathcal{Y}} f^*(y) d\mathbb{Q}(y) \leq \tag{40}$$

$$\int_{\mathcal{X}} \left\{ c(x, T^*(x)) - f^*(T^*(x)) \right\} d\mathbb{P}(x) + w \int_{\mathcal{Y}} f^*(y) d\mathbb{Q}(y) = \tag{41}$$

$$\int_{\mathcal{X}} c(x, T^*(x)) d\mathbb{P}(x) - \int_{\mathcal{X}} f^*(y) d[T^* \sharp \mathbb{P}](y) + w \int_{\mathcal{Y}} f^*(y) d\mathbb{Q}(y) =$$

$$\text{Cost}_w(\mathbb{P}, \mathbb{Q}) + \underbrace{\int_{\mathcal{Y}} f^*(y)d\big[w\mathbb{Q} - T^*\sharp\mathbb{P}\big](y)}_{\leq 0 \text{ (Lemma 1)}} \leq \text{Cost}_w(\mathbb{P}, \mathbb{Q}). \quad (42)$$

This means that all the inequalities in (40)-(42) are equalities. Since (40) equals (41), we have $T^* \in \arg\min_{T:\mathcal{X}\to\mathcal{Y}} \mathcal{L}(f^*, T)$. $\qquad\square$

**Bibliographic remark (theoretical part).** The idea of the theorem is similar to that of [61, Lemma 4.2], [24, Lemma 3], [43, Lemma 4], [20, Theorem 2] which prove that their respective saddle point objectives $\max_f \min_T \mathcal{L}(f, T)$ can be used to recover optimal $T^*$ from *some* optimal saddle points $(f^*, T^*)$. Our functional (15) differs, and we have the constraint $f \leq 0$. We again emphasize here that not for all the saddle points $(f^*, T^*)$ it necessarily holds that $T^*$ is the IT map, see the discussion in limitations (Appendix A).

**Bibliographic remark (algorithmic part).** To derive our saddle point optimization problem (15), we use the $c$-transform expansion proposed by [53] in the context of Wasserstein GANs and later explored by [40, 43, 61, 20, 24, 31] in the context of learning OT maps. That is, our resulting algorithm 1 overlaps with the standard maximin neural OT solver, see, e.g., [24, Algorithm 1]. The difference is in the constraint $f \leq 0$ and the additional multiplier $w \geq 1$.

*Proof of Proposition 4.* From the proof of Theorem 5, we see that $\int_{\mathcal{Y}} f^*(y)d\big[w\mathbb{Q} - T^*\sharp\mathbb{P}\big](y) = 0$. Recall that $f^* \leq 0$. This means that $f(y) = 0$ for $y \in \text{Supp}\big(w\mathbb{Q} - T^*\sharp\mathbb{P}\big)$. Indeed, assume the opposite, i.e., there exists some $y \in \text{Supp}\big(w\mathbb{Q} - T^*\sharp\mathbb{P}\big)$ for which $f(y) < 0$. In this case, the same holds for all $y'$ in a small neighboorhood $U$ of $y$ as $f$ is continuous. At the same time, $\int_{\mathcal{Y}} f^*(y)d\big[w\mathbb{Q}-T^*\sharp\mathbb{P}\big](y) \leq \int_U f^*(y)d\big[w\mathbb{Q}-T^*\sharp\mathbb{P}\big](y) < 0$ since $\big[w\mathbb{Q}-T^*\sharp\mathbb{P}\big]$ is a non-negative measure satisfying $\big[w\mathbb{Q} - T^*\sharp\mathbb{P}\big](U) > 0$ by the definition of the support. This is a contradiction. To finish the proof it remains to note that $\text{Supp}(\mathbb{Q}) \setminus \text{Supp}(T^*\sharp\mathbb{P}) \subset \text{Supp}(w\mathbb{Q} - T^*\sharp\mathbb{P})$, i.e., $f(y) = 0$ for $y \in \text{Supp}(\mathbb{Q}) \setminus \text{Supp}(T^*\sharp\mathbb{P})$ as well. $\qquad\square$

**Bibliographical remark.** Treating functional $\mathcal{L}(f, T)$ in (16) as a Lagrangian, Proposition 4 can be viewed as a consequence of the complementary slackness in the Karush-Kuhn-Tucker conditions [37].

## G   Additional Experimental Illustrations

### G.1   Comparison with the Closed-form Solution for ET

In this section, we conduct a *Swiss2Ball* experiment in 2D demonstrating that for the sufficiently large parameter $w$, IT maps become fine approximations of the ground-truth ET map. We define source measure $\mathbb{P}$ as a uniform distribution on a swiss roll centered at $(0, 0)$. Target measure $\mathbb{Q}$ is a uniform distribution on a ball centered at $(0, 0)$ with radius $R = 0.5$, i.e., $\text{Supp}(\mathbb{Q}) = B(0, 0.5)$. We note that the supports of source and target measures are partially overlapping. In the proposed setup, the solution to ET problem (5) has a closed form: $T(x) = x \cdot 1_{x \in B((0,0),0.5)} + x \cdot \frac{R}{\|x\|_2} \cdot 1_{x \notin B((0,0),0.5)}$, see Fig. 20f. We provide the learned IT maps for $w \in \{1, 3/2, 2, 32\}$, see Fig. 20b-20e. The qualitative and quantitative results show that with the increase of $w$ our IT maps become closer and closer to the ground-truth ET one, see Table 8.

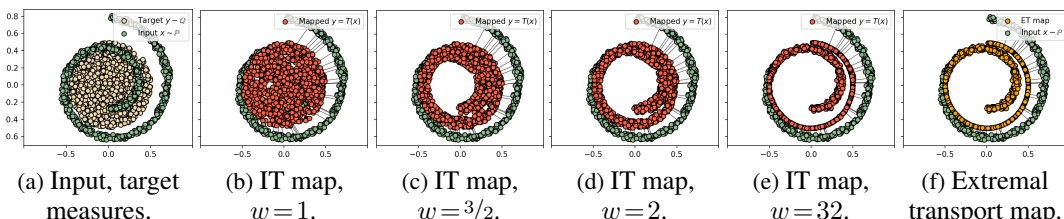

(a) Input, target measures.

(b) IT map, $w = 1$.

(c) IT map, $w = 3/2$.

(d) IT map, $w = 2$.

(e) IT map, $w = 32$.

(f) Extremal transport map.

Figure 20: Incomplete Transport (IT) maps learned with $c(x, y) = \|x - y\|_2^2$ transport cost and ground-truth Extremal Transport (ET) map in 'Swiss2Ball' experiment.

|  | $w = 1$ | $w = {}^{3}/_{2}$ | $w = 2$ | $w = 32$ |
|---|---|---|---|---|
| MSE($\hat{T}, T^*$) | 0.0136 | 0.0026 | 0.0009 | 7.98e-06 |

Table 8: MSE($\hat{T}, T^*$) between learned IT maps $\hat{T}$ ($w \in \{1, {}^{3}/_{2}, 2, 32\}$) and ground-truth ET map $T^*$.

## G.2 Solving Fake Solutions Issue with Weak Kernel Cost

As we discuss in Appendix A, saddle-point neural OT methods (including our IT algorithm) may suffer from *fake solutions* issue. As it is proved in [42], this issue can be eliminated by considering OT with the so-called *weak kernel cost* functions. In this section, we test the effect of using them in our IT algorithm. Specifically, we demonstrate the example where IT algorithm with the cost function $c(x, y) = \|x - y\|_2$ struggles from fake solutions, while IT with the same cost endowed with weak kernel regularization, i.e., kernel cost [42, Equation (16)] with parameters $\alpha = 1, \gamma = 0.4$, resolves the issue. Following [42], we consider stochastic version of IT map $T(x, z)$ using noise $z \sim \mathcal{N}(0, \mathcal{I})$ as an additional input.

We design the *Ball2Circle* example in 2D, where input $\mathbb{P}$ is a uniform distribution on a ball and target $\mathbb{Q}$ − on a ring embracing the ball. The solution of ET problem is an internal circuit of a ring, see Fig. 21a. We learn IT maps for $c(x, y) = \|x - y\|_2$, with ($\gamma = 0.4$) or without ($\gamma = 0$) kernel regularization for weights $w \in \{1, 2, 32\}$.

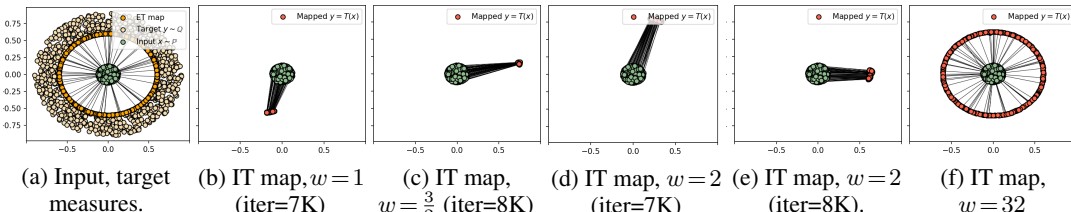

(a) Input, target measures.  (b) IT map, $w = 1$ (iter=7K)  (c) IT map, $w = \frac{3}{2}$ (iter=8K)  (d) IT map, $w = 2$ (iter=7K)  (e) IT map, $w = 2$ (iter=8K).  (f) IT map, $w = 32$

Figure 21: Incomplete Transport (IT) maps learned with $c(x, y) = \|x - y\|_2$ transport cost in *'Ball2Circle'* experiment. We observe that training is highly unstable for $w \in \{1, 2\}$, see the solutions for nearby iterations of training - (b-c) and (d-e) respectively. Increase of the weight $w$ helps to improve the stability (f).

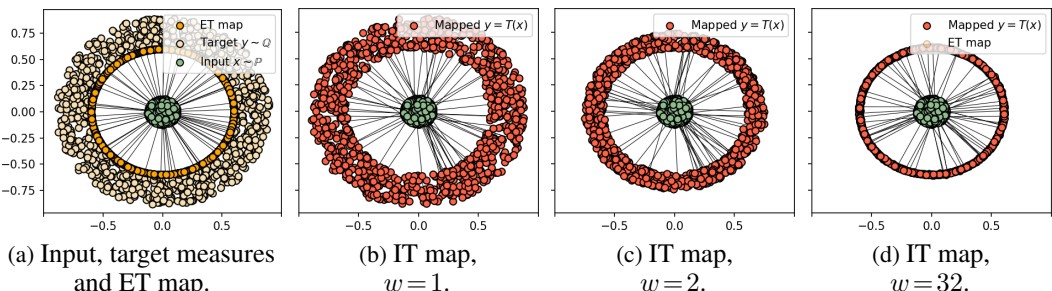

(a) Input, target measures and ET map.  (b) IT map, $w = 1$.  (c) IT map, $w = 2$.  (d) IT map, $w = 32$.

Figure 22: Incomplete transport (IT) maps learned with weak kernel cost in *'Ball2Circle'* experiment for a fixed noise $z$. Usage of the kernel regularization + stochastic map $T(x, z)$ helps to overcome instability issues for weights $w \in \{1, 2\}$.

| method / weight | $w = 1$ | $w = 2$ | $w = 32$ |
|---|---|---|---|
| w/o kernel regularization | - | - | 0.0022 |
| with kernel regularization | 0.1377 | 0.0730 | 0.0495 |

Table 9: MSE($\hat{T}, T^*$) between IT maps ($w \in \{1, 2, 32\}$) learned with $c(x, y) = \|x - y\|_2$ (with and without weak *kernel regularization*) and the ground-truth ET map. MSE for IT map learned with kernel regularization is larger (see $w = 32$) than without the regularization since it introduces small bias to the optimization.

**Discussion.** We observe that without kernel regularization training of IT method is highly unstable (for $w \in \{1, 2\}$), see Fig. 21b-21d. Interestingly, with the increase of the weight $w$, the issue disappears and for $w = 32$, IT map is close to the ground-truth ET one (Fig. 21f).

The usage of kernel regularization helps to improve stability of the method, see Fig. 22. However, while MSE between learned and ground-truth ET map drops with the increase of $w$, for $w = 32$ it is bigger than that of learned IT maps without regularization. It is expected, since using regularizations usually leads to some bias in the solutions.

Thus, the example shows that (a) fake solutions may be a problem, (b) kernel regularization from [42] may help to deal with them. However, further studying this aspect is out of the scope of the paper.

### G.3 Perceptual Cost

In this section, we show that the stated conclusions hold true for the transport costs other than $\ell^2$. For this purpose, we perform additional experiments using *perceptual* transport cost from [24]. We use the same hyperparameters as in our experiments with $\ell^2$ cost.

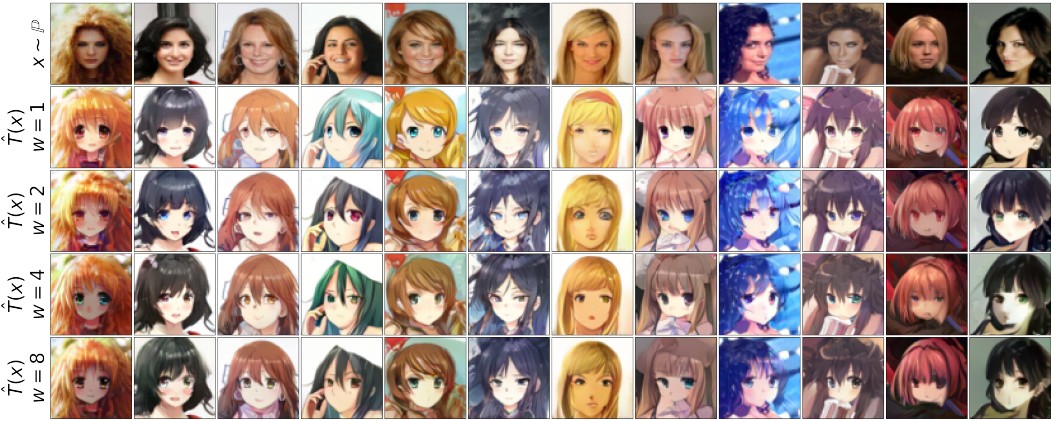

Figure 23: *Celeba* (female) $\rightarrow$ *anime* (64$\times$64 image size, perceptual cost).

| **Metrics** | $w = 1$ | $w = 2$ | $w = 4$ | $w = 8$ |
|:---:|:---:|:---:|:---:|:---:|
| *Test FID* | 9.21 | 12.98 | 17.24 | 22.08 |
| *Test perceptual cost* | 0.954 | 0.794 | 0.667 | 0.545 |
| *Test $\ell^2$ cost* | 0.303 | 0.209 | 0.153 | 0.103 |

Table 10: Test FID and $\ell^2$, perceptual transport costs of our IT maps (learned with perceptual transport cost).

**Experimental results.** Qualitative results show that that similarity of input images $x$ and the images $\hat{T}(x)$ translated by our IT method trained with perceptual cost grows with the increase of the parameter $w$, see Figure 23. In 10, we quantitatively verify these observations by showing that both perceptual and $\ell^2$ mean transport costs between input and translated images decrease with the increase of $w$. Interestingly, we see that IT trained with perceptual cost yields smaller FID than IT method trained with $\ell^2$ cost.

### G.4 Bigger Weight Parameters

In this section, we present the results of IT method trained with $\ell^2$ cost and parameters $w = 16, 32$. We see that in contrast to CycleGAN which does not translate face images to anime images in case of big parameters $\lambda$, our IT method continues to translate faces to anime even for big parameters $w$. We quantify the obtained results in Table 11. As expected, mean transport costs are decreasing with the increase of $w$, while FID is slightly increasing. We present the qualitative results for additional weights in Figure 24.

| Metrics | Main results | | | | Additional results | |
|---|---|---|---|---|---|---|
| | $w = 1$ | $w = 2$ | $w = 4$ | $w = 8$ | $w = 16$ | $w = 32$ |
| *Test FID* | 14.65 | 20.79 | 22.18 | 22.84 | 24.86 | 28.28 |
| *Test $\ell^2$ cost* | 0.297 | 0.154 | 0.133 | 0.094 | 0.091 | 0.083 |

Table 11: Test FID and $\ell^2$ transport costs of our IT maps.

## G.5  Additional Results for $\ell^2$ Cost

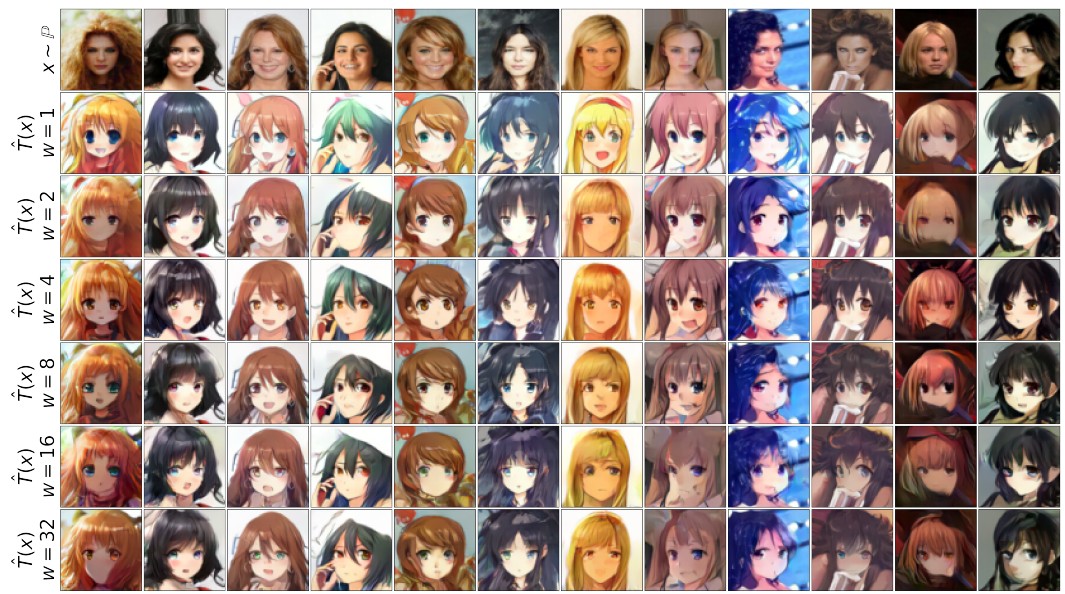

Figure 24: *Celeba* (female) → *anime* (64×64).

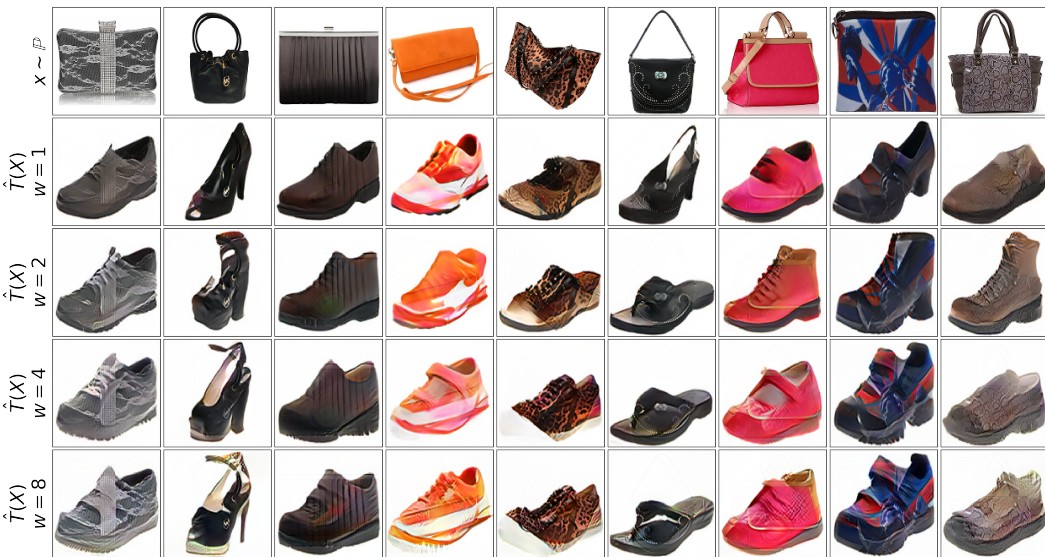

Figure 25: *Handbag* → *shoes* (128×128).

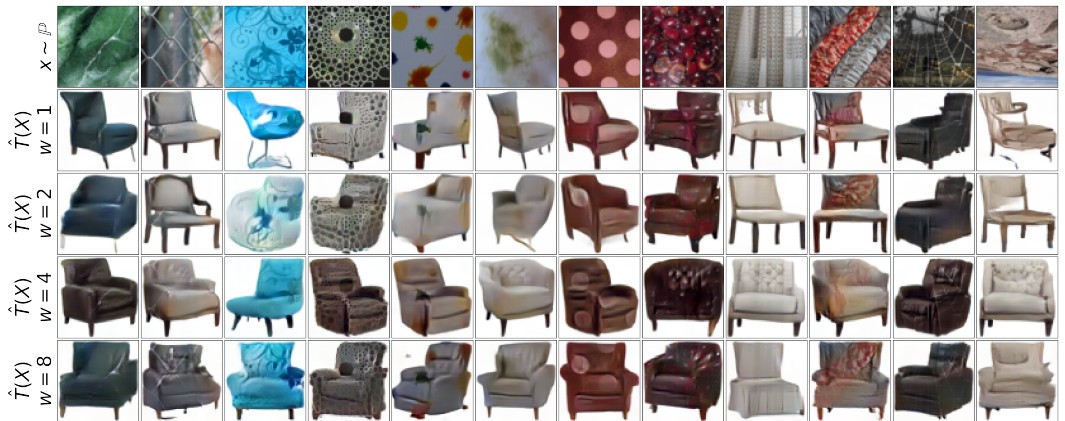

Figure 26: *Textures → chairs* (64×64).

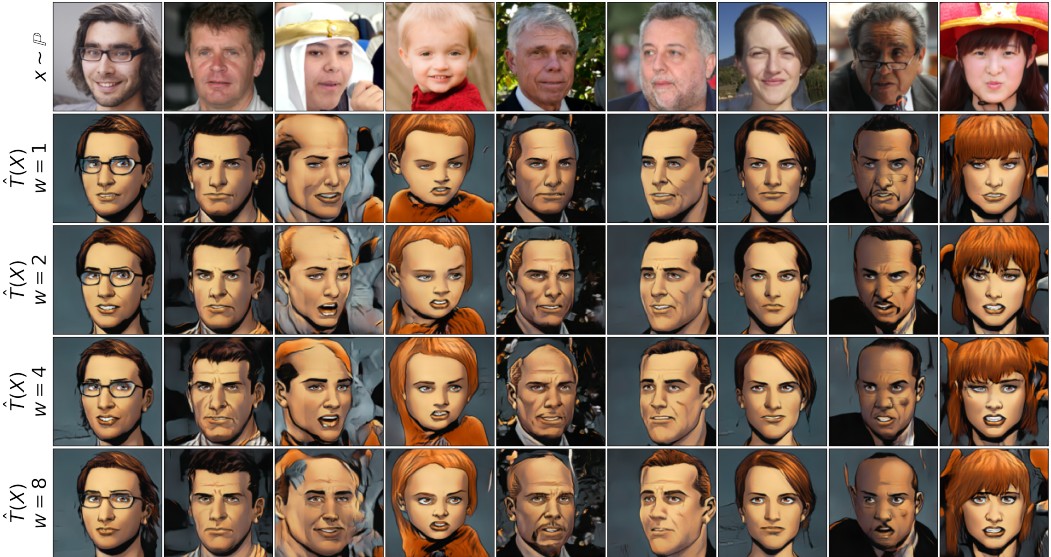

Figure 27: *Ffhq → comics* (128×128).

