# OpenReview forum: "Extremal Domain Translation with Neural Optimal Transport"
_NeurIPS.cc/2023/Conference — NeurIPS 2023 poster_

### Official Review · Reviewer_3fqr · 2023-07-04

**Soundness:** 3 good
**Presentation:** 4 excellent
**Contribution:** 3 good
**Rating:** 7
**Confidence:** 3

**Summary:**

This paper presents a novel OT problem, extremal transport (ET), in the context of domain translation. The authors propose an incomplete transport (IT) problem as a surrogate optimization problem to obtain an approximate solution to the ET problem. The theoretical convergence between IT and ET costs (plans) is proven conditionally. The effectiveness of ET and IT is showcased through experiments on toy examples and public datasets, while also demonstrating the relationship between ET and IT.

**Strengths:**

- The optimal transport formulations in this work, i.e., ET and IT, are something that I have not seen. These formulations, for me, are inspiring and interesting.

- The paper provides a thorough discussion on the technical differences and relationship with other existing methods.

- The overall presentation of the main paper is generally clear. Furthermore, the full paper, including the appendices, appears to be comprehensive and well-prepared. The theoretical results are presented in a comprehensive and self-contained manner.

**Weaknesses:**

- The scope of and motivation for the work are a little unclear from the abstract and introduction. In particular, the motivation to incorporate ET in the context of domain translation is not clear, which could make readers confused. It should be highlighted at least in the abstract and introduction sections.

- Certain statements lack sufficient explanations or supporting intuitions, which should be explained and provided in the main text. To list a few:
  - In the lines 85-91, a variant of OT problem is established. The authors mentioned "we say that the target domain is the part of Y where the probability mass of Q lives". However, the intuition behind this statement is not well-presented and should be elaborated upon.
  - In the lines 76-78, the intuition to establish the ET framework in the context of domain translation should be demonstrated. It should also be highlighted in the abstract and introduction sections. Specifically, readers may hope to see why it is essential to construct ET for domain translation task, given thorough related works on weak OT, partial OT, etc... Although you have formulated the technical difference with them, I hope to see analysis on "why ET outperforms them in your application".
  - In the lines 91-92, the authors relaxes the mass preservation condition for the target domain. Why not relax the condition for both the target and source domain, likewise UOT? A concise and simple comparison with current weak / partial OT should be clarified here, to **explain the advantages of ET in domain translation compared with peer methods, (rather than purely formulating the technical differences)**, which is critical for identifying and evaluating ET's role in domain translation.**


### ************ Minors ***************

- Figure 3-4 are kind of redundant and largely inherit from current literatures. As such, it is not appropriate to take too much blank space for them. A more proper way would be consendate them in one line or move them to the appendix.
- In the line 104, the NN could lead to unexpected confusion since it usually represents neural networks.
- In the lines 80-109, the authors establish an optimization problem to seek for the nearest neighbors of $x\in\mathcal{X}$ in Supp(Q). The narrative in this section is kind of redundant.
- It is essential for readers to acknowledge the technical differences with other OT methods. As such, I suggest authors moving the section of "Related works" after the section of "background on optimal transport", and highlight your technical difference with each genre of OT methods concisely.

My current score is a reflection of the weaknesses above as they stand. Since these seem like rather correctable issues, I expect to increase my score pending a positive response from the authors.

**Questions:**

- The authors noted "the task is ill-posed as there might exist multiple suitable T." It is confusing for me due to two reasons: (1) you just mention that there could be multiple maps between source and target; however, the optimization problem has not been formulated here, and it is not proper to say it is ill-posed problem since the word ill-posed seems to be suitable for optimization problems; (2) the optimization problem, such as the Kantorovich problem, could be not ill-posed in some cases given certain metrics, is it right? Please consider to improve the narrative to make it more rigorous and self-contained.
- The authors noted "challenge here is that the correspondence between vailable data samples a from the source and y from target domains is not given." What if the correspondence is given? Can we get a new optimization problem with intriguing properties?

**Limitations:**

The limitations have been well discussed.

---

> ### Author Rebuttal · Authors · 2023-08-09
>
> Dear Reviewer, thank you for your comments. Here are the answers to your questions.
>
> **(1) The motivation to incorporate ET in the context of domain translation is not clear <...> in the abstract and introduction.**
>
> The motivation to incorporate ET in the context of domain translation is implicitly stated in lines 25-31 of the introduction. Indeed, ET allows to perform domain translation while achieving best possible similarity between source and generated images. Such property is needed in various particular downstream image-to-image problems including, e.g., image inpainting or super-resolution (lines 222-225). *We will state the motivation more clearly in the abstract and introduction in the final version of our paper.*
>
> **(2) The intuition behind "we say that the target domain is the part of $\mathcal{Y}$ where the probability mass of $\mathbb{Q}$ lives" (lines 85-91) is not clear.**
>
> Consider a typical example $\mathcal{Y}=[-1,1]^{D}$. Following the standard manifold hypothesis [1], real data distribution $\mathbb{Q}$ (e.g., distribution of images of faces) is usually supported on a small-dimensional manifold $M=$ Supp$(\mathbb{Q})\subset [-1,1]^{D}$ occupying a tiny part of the ambient space $[-1,1]^{D}$. Thus, the intuition behind the mentioned statement is that the probability distribution $\mathbb{Q}$ lives on this manifold $M$ which represents a target domain. *We will add clarifications to the final version.*
>
> **(3) Demonstrate the intuition to establish the ET framework in the context of domain translation. Why it is essential to construct ET for domain translation task, given thorough related works on weak OT, partial OT, etc... <...> "why ET outperforms them".**
>
> Our experiments with domain translation do not pretend to show that ET construction is essential to solve domain translation task, but only demonstrates that proposed numerical algorithm are applicable to the problem with real data. In particular, it can recover very good similarity of translated samples to input samples which may be of high importance in certain image-to-image translation tasks (lines 222-233) while existing (GAN-based) methods encounter their limitations (Appendix). To the best of our knowledge, there are also no works that apply weak OT or partial OT to solve the same problem of ET (nearest neighbours computation with the out-of-sample generalization).
>
> **(4) In the lines 91-92, the authors relaxes the mass preservation condition for the target domain. Why not relax the condition for both the target and source domain, likewise UOT?**
>
> In a task of domain translation, we need to translate the sample from the source domain to the target one. Relaxing the condition for target domain helps us to achieve the 'best possible similarity' between source and generated images which is a primary goal of our IT approach. At the same time, relaxing source distribution is counter-intuitive for domain translation task where all the **test** samples will come from the source distribution $\mathbb{P}$.
>
> **(5a) The NN abbreviation is confusing.**
>
> We agree with the reviewer that NN abbreviation is commonly employed to denote neural networks. At the same time, it is also common abbreviation for nearest neighbors, e.g., like in $k$-NN. To avoid misunderstanding, we explain the abbreviation several times in the paper, see line 32 and 104-106.
>
> **(5b) In the lines 80-109, the authors establish an optimization problem to seek for the nearest neighbors of x$\in$X in Supp(Q). The narrative in this section is kind of redundant.**
>
> We kindly ask the reviewer to specify which parts of the section seem to be redundant. We will be happy to improve the section narrative.
>
> **(5c) Acknowledge the technical differences with each genre of OT methods. Move the "Related works" section after "background on OT".**
>
> We highlight technical differences with conventional neural OT methods in Appendix F (lines 906-910), discrete partial OT methods in Appendix D (lines 650-657), discrete unbalanced OT in Appendix B (lines 554-560). Unfortunately, we could not move related work section to the beginning of the paper, since it uses some ET/IT notions and properties which are introduced in prior sections. *We will move the technical differences with other OT methods from Appendix to the related work section.*
>
> **(6) The statement "the task is ill-posed as there might exist multiple suitable T" is confusing. <...> the Kantorovich problem, could be not ill-posed in some cases given certain metrics, is it right?**
>
> (a) We agree with the reviewer's comment that maybe 'ill-posed' is not an appropriate designation for the problem before it is strictly mathematically formulated. *We will change the word 'ill-posed' to 'ambiguous' in line 15 in the final version of our paper.*
>
> (b) You are correct in your understanding that Kantorovich problem is not an ill-posed problem.
>
> **(7) What if the correspondence between available data samples from the source and y from target domains is given?**
>
> If the correspondence is given, i.e., **paired** samples $(x_n,y_n)$ are available for training, the problem can be straightforwardly solved by common supervised learning techniques such as Pix2Pix [2]. This setup is out of the scope of our paper. We consider the **unpaired** setup where the is no supervision.
>
> **Concluding remarks**. Please respond to our post to let us know if the clarifications above suitably address your concerns about our work. We are happy to address any remaining points during the discussion phase; if the responses above are sufficient, we kindly ask that you consider raising your score.
>
> **References.**
>
> [1] Fefferman, C. et al. Testing the manifold hypothesis.
>
> [2] Isola, P. et al. Image-to-image translation with conditional adversarial networks.

---

> > ### Comment · Reviewer_3fqr · 2023-08-20
> > **Acknowledgement**
> >
> > Thank you very much for your detailed response. The author response has mitigated my concerns, especially (1-4). I believe the authors could consider my suggestions in preparing their final version. After reading the comments from other reviewers and author responses, I am pleased to adjust my initial rating to support acceptance of this work more definitely.

---

### Official Review · Reviewer_Cqh5 · 2023-07-05

**Soundness:** 3 good
**Presentation:** 3 good
**Contribution:** 3 good
**Rating:** 7
**Confidence:** 3

**Summary:**

The paper proposes extremal transport (ET), a mathematical framework for achieving the best possible unpaired translation between two domains based on a given similarity function, and proving that ET maps can be learned as a limit of specific partial optimal transport (OT) problem. The contributions of the paper include a formalization of ET as a rigorous mathematical task, the characterization of ET maps and plans through a connection to nearest neighbors, and the derivation of an efficient computational algorithm based on partial optimal transport. The proposed algorithm is evaluated using 2D examples and the unpaired image-to-image translation task.

**Strengths:**

1.The logic of this paper is well organized, and also the mathematical formulas are well defined.
2.The work demonstrates that it is possible to relax equality constraints to inequality constraints especially for some situations where enforcing rigorous equal constraints are difficult, and proposes incomplete transport (IT) to approximate ET maps using partial optimal transport (OT) problems, which is very novel.
3.The experimental part is quite sufficient complete and proves the effectiveness of the proposed method.

**Weaknesses:**

It is not clarified the motivations of defining ET problem and the relationship between specific task such as unpaired image translation and the mathematical ET problem.The process from abstraction of practical problems to theoretical derivation is too steep.

**Questions:**

Why the ET problem is defined in the compact polish spaces？Could it transfer to other spaces?

**Limitations:**

The ℓ2 and FID metrics have some limitations in characterizing the effectiveness of the method.

---

> ### Author Rebuttal · Authors · 2023-08-09
>
> Dear Reviewer, thank you for your comments. Here are the answers to your questions.
>
> **(1) It is not clarified the motivations of defining ET problem and the relationship between specific task such as unpaired image translation and the mathematical ET problem.The process from abstraction of practical problems to theoretical derivation is too steep.**
>
> We thank reviewer for pointing this. Our choice to consider unpaired image translation problem in the context of ET was motivated by the fact that recovering good similarity of translated samples to input samples may be of high importance in certain image-to-image translation tasks (lines 222-233).
>
> **(2) Why the ET problem is defined in the compact Polish spaces. Could it transfer to other spaces?**
>
>
> **Compactness.** We set the spaces $\mathcal{X}$ and $\mathcal{Y}$ to be compact as it is a natural property which holds for various real-world distributions, e.g., images. At the same time, the compactness assumption notably shortens the derivation of the theoretical results, e.g., $\inf$/$\sup$ can be automatically replaced by $\min$/$\max$ everywhere simplifying derivations, etc. We think that most of our results can be generalized to non-compact spaces but we leave this for future theoretical studies.
>
> **Polish spaces.** Recall that a Polish space is a *separable completely metrizable topological space* (and $\mathbb{R}^{D}$ is an example). Its metrizable property yields the equivalence between compactness and sequential compactness which is simpler to work with (we use it, e.g., in our Theorem 3). Its separability is required in Banach-Alaouglu theorem which we use in our Proposition 2. Completeness holds automatically due to the compactness assumption.
>
> **(3) The $\ell^2$ and FID metrics have some limitations in characterizing the effectiveness of the method.**
>
> We use the FID metric simply because there are no principally different alternatives. In fact, all the metrics for generative models which we have seen in the related papers evaluate set-to-set similarity rather than the quality of individual samples. If you can suggest popular metrics based on individual images, we are happy to include them to our evaluation.
>
> Regarding the $\ell^2$ metric, we used as a cost fuction during trining of our models and, as a consequence, employ it to evaluate the similarity between input and generated images.
>
> **Concluding remarks**. Please respond to our post to let us know if the clarifications above suitably address your concerns about our work. We are happy to address any remaining points during the discussion phase; if the responses above are sufficient, we kindly ask that you consider raising your score.

---

### Official Review · Reviewer_URfJ · 2023-07-05

**Soundness:** 4 excellent
**Presentation:** 4 excellent
**Contribution:** 4 excellent
**Rating:** 9
**Confidence:** 3

**Summary:**

This paper introduces a mathematical formalization called "extremal transport" that aims to achieve optimal translation between unpaired domains based on a given similarity function. Additionally, the paper proposes a scalable algorithm that utilizes neural optimal transport to approximate extremal transport mappings. The algorithm is tested on toy examples and the image-to-image translation task, yielding promising results.

**Strengths:**

The paper introduces a novel mathematical formalization called extremal transport for achieving optimal translation between unpaired domains. Additionally, the concept of neural optimal transport is introduced and applied in the algorithm. These theoretical foundations provide a solid basis for the algorithm design.

The algorithm is tested on toy examples and the image-to-image translation task, demonstrating good results. These experiments validate the effectiveness and scalability of the algorithm.

Image-to-image translation is an important problem in computer vision, and the proposed algorithm can be applied to address this problem. Furthermore, the paper mentions other potential application areas, such as single-cell data analysis in biology.

The algorithm proposed in this paper can be widely applied to solve translation problems between unpaired domains, making it highly generalizable. Additionally, a new method called partial optimal transport is introduced, which can be used to address alignment problems between imbalanced measures, further expanding its potential for generalization.

**Weaknesses:**

Based on the information provided, the paper primarily focuses on the common main features of existing approaches rather than encompassing all of them. It's possible that due to space constraints, some concepts or details may not be extensively elaborated in the main paper. However, substantial explanations are provided in the Appendix.

**Questions:**

In lines 121-124, how the lower bound is defined during the process of analyzing and constructing upper and lower bounds to simplify the objective is not mentioned. The reason why value(8) admits more minimizers is also not explained.

In lines 133-136, an explanation is needed for the replacement mentioned here and why using a finite parameter w can achieve the desired replacement of Supp(T#P) belonging to Supp(Q).

In lines 218-220, it is asked whether the method presented in the paper scales to high dimensions. If images are defined as high-dimensional, this should be clarified from the beginning of the paper.

**Limitations:**

The paper has conducted extensive discussions.

---

> ### Author Rebuttal · Authors · 2023-08-09
>
> Dear Reviewer, thank you for your comments. Here are the answers to your questions.
>
> **(1a) In lines 121-124, how the lower bound is defined during the process of analyzing and constructing upper and lower bounds to simplify the objective is not mentioned.**
>
> Equation (10) states that for any $\pi\in\Pi^{\infty}(\mathbb{P},\mathbb{Q})$  it holds (for $\mathbb{P}$-almost all $x\in\mathcal{X}$) that: $c^{\*}(x)=\min_{y\in Supp(\mathbb{Q})}c(x,y)\leq \int_{\mathcal{Y}}c(x,y)d\pi(y|x)$.
> If we now integrate the equation (10) with respect to $x\sim\mathbb{P}=\pi_{x}$ and take $\inf$ over all admissible plans, we get the following inequality:
> $\int_{\mathcal{X}}c^{\*}(x)d\mathbb{P}(x)\leq \inf_{\pi\in\Pi^{\infty}(\mathbb{P},\mathbb{Q})}\int_{\mathcal{X}\times\mathcal{Y}}c(x,y)d\pi(x,y)$. It shows that indeed $\int_{\mathcal{X}}c^{\*}(x)d\mathbb{P}(x)$ is a lower bound for equation (8).
> At the same time, as we write in lines 102-103, $\int_{\mathcal{X}}c^{\*}(x)d\mathbb{P}(x)$ is also a lower bound for equation (5). From our Theorem 1 we see that there exists measurable map $T^\*$ minimizing (5) such that the bound is tight. It automatically yields existence of a deterministic plan $\pi^\*(y|x)=\delta_{T^\*(x)}$ such that (8)=$\int_{\mathcal{X}}c^{\*}(x)d\mathbb{P}(x)$=(5). *We will add these details to the final version of our paper.*
>
> **(1b) The reason why value (8) admits more minimizers is also not explained.**
>
> This fact follows from the definitions of Monge's and Kantorovich's Extremal Transport (ET) formulations. Indeed, like in the general OT theory, Kantorovich's ET formulation (8) is an extension of Monge's one (5). If there exists a minimizer $T^\*$ of equation (5), then it yields the corresponding deterministic minimizer $\pi^\*(y|x)=\delta_{T^\*(x)}$ of (8). However, Kantorovich's formulation allows to split the mass between nearest neighbors and, therefore, may potentially admit more minimizers.
>
> *Example.* Consider $\mathcal{X}=\mathcal{Y}=[-1,1]$. Let $\mathbb{P}=\delta_0$ and $\mathbb{Q}=\frac{1}{2}\delta_{-1}+\frac{1}{2}\delta_1$ be distributions concentrated at $\{0\}$ and $\{-1, 1\}$ respectively. Let $c(x, y)=\frac{1}{2}\|x-y\|^{2}$ be the quadratic cost. Then there are obviously two extremal OT maps $T^\*$ delivering minimum to Monge's ET  problem: $T^\*(0)=-1$ and $T^\*(0)=1$. At the same time, there are infinitely many minimizers $\pi^{\*}$ of the Kantrorovich's problem (8). Indeed, all the plans distributing the mass of $\mathbb{P}$ between points $\{-1\}$, $\{1\}$, i.e., satisfying $\pi(0, -1)=m$, $\pi(0, 1)=1-m$ are minimizers for $m\in [0,1]$.
>
> **(2) In lines 133-136, an explanation is needed for the replacement mentioned here and why using a finite parameter $w$ can achieve the desired replacement of $Supp(T\sharp\mathbb{P})$ belonging to $Supp(\mathbb{Q})$.**
>
> For an arbitrary finite parameter $w$, the condition $Supp(T_{\sharp}\mathbb{P}) \subset Supp(\mathbb{Q})$ implies $T_{\sharp}\mathbb{P}\leq w \mathbb{Q}$. As we show in our paper, in the limit $w\rightarrow\infty$, the solutions to IT problem converge to the ET in a certain sense. At the same time, the replacement of ET with IT is desired as the latter one admits the dual formulation (which we derive) that can be efficiently solved with neural networks. How to directly enforce the ET condition $Supp(T_{\sharp}\mathbb{P}) \subset Supp(\mathbb{Q})$ in practice is an open problem.
>
> **(3) In lines 218-220, it is asked whether the method presented in the paper scales to high dimensions. If images are defined as high-dimensional, this should be clarified from the beginning of the paper.**
>
> Thank you for noting this aspect. *We will add the clarifications to the final version of our paper.*
>
> **Concluding remarks**. Please respond to our post to let us know if the clarifications above suitably address your concerns about our work. We are happy to address any remaining points during the discussion phase.

---

> > ### Comment · Reviewer_URfJ · 2023-08-16
> >
> > The authors answered questions and will also add details. Their work is greatly appreciated.

---

### Official Review · Reviewer_Mz3K · 2023-07-18

**Soundness:** 2 fair
**Presentation:** 2 fair
**Contribution:** 2 fair
**Rating:** 4
**Confidence:** 4

**Summary:**

This paper introduces the concept of extremal Optimal Transport (OT) and proposes the use of incomplete OT as a solution to the extremal OT problem. The authors present a duality method to address the incomplete OT problem and validate this approach using a toy 2D dataset and image translation tasks.

**Strengths:**

The authors offer the analysis of the proposed problems and algorithms, including the existence of Extremal Transport (ET), the presence of Incomplete Transport (IT), and the convergence of IT plans to the ET plans.

They test their algorithms on several image translation datasets, demonstrating the ability to manipulate the similarity between the source image and the generated image by adjusting the weight parameter $w$.

Their algorithm exhibits the capability to map the data towards a portion of the target distribution's support.


**Weaknesses:**

## Motivation

Firstly, the paper's logic doesn't fully convince me. The authors initially propose to solve the Extremal Transport (ET) problem, but then shift to solving the Incomplete Transport (IT) problem, presumably because the ET problem is too challenging. However, based on their experimental results, it seems more logical to propose solving the IT problem directly, as most of their visualizations demonstrate that adjusting the weight parameter $w$ controls the similarity between the source image and the generated image. If the authors are indeed intent on solving the ET problem, could they provide experimental evidence to support this? For instance, they could design some distributions that have a closed-form solution for ET, and then compare the ground truth minimum value of problem (5) with their simulation results.

Regarding the motivation for problem (11): the authors state that "In practice, solving the extremal problem (5) is challenging because it is hard to enforce Supp($T \sharp P$) $\subset$ Supp($Q$)", but isn't this relatively straightforward for image tasks? Since images are typically scaled to a certain range like [-1,1], we can simply scale the images to meet this support requirement.

I believe that the application of incomplete OT or extremal OT could be limited in image tasks, especially extremal OT, as multiple data points could be mapped to the same data point, resulting in generated images with limited diversity.

## Method

Regarding the relaxation from problem 5 to 11, it seems to me that the constraint in 11 is not a softened version, but rather a stronger one. To satisfy the constraint in (11), one must also satisfy the constraint in (5). This is because in the constraint of (11), if Q=0, then regardless of the value of $w$, $T\sharp P$ must also be zero, which aligns with the constraint in (5). Why is it not feasible to design a dual formula for ET directly?

The authors mention the issue of fake solutions. Could they consider borrowing ideas from the Kernel Neural OT paper? It seems that using a similar cost from Kernel Neural OT could enhance this paper, and it seems not that difficult?

## Results

Given that the motivation of this paper is to solve the extremal OT problem, could the authors clarify under what conditions their method can recover the solution to extremal OT? Specifically, how large does the value of $w$ need to be? It's possible that this information is already included, and I may have overlooked it.

The authors acknowledge that the FID is not particularly meaningful when $w>1$, yet it appears to be the primary metric used in this paper. If the authors believe that FID is not representative, why do they present so many FID results? They also claim that the image quality of the translated image does not decrease with increasing $w$. Is there a way to verify this quantitatively, perhaps with a metric based on the image itself rather than the image distribution?

Some of the results presented in this paper seem less than satisfactory. For instance, in Figure 24, even when $w$ is large, some images fail to preserve hair colors. In Figure 22, as $w$ increases, some images exhibit unrealistic artifacts (e.g., the first grey shoe and the black saddle have strange bands on them), likely because the generated image is forced to share more similarity with the source image. Could this be a limitation of the proposed method?

**Questions:**

Figure 5b may not be as illustrative as intended. In this case, it seems that pi would still uniquely correspond to a certain deterministic transport map. Perhaps it would be more instructive to provide an example where (8) admits multiple minimizers, such as a scenario where the support is not convex?

**Limitations:**

.

---

> ### Author Rebuttal · Authors · 2023-08-09
>
> Dear Reviewer, thank you for your comments. Here are the answers to your questions.
>
> **(1) Design distributions that have a closed-for ET solution. Compare simulation results with the ground truth.**
>
> We conduct *Swiss2Ball* experiment in 2D, see Fig. 1, Table 1 in **the attached PDF file**. Here the supports of source and target measures are partially overlapping and the solution to ET problem (5) has a closed form:
> $ T(x) = x \cdot 1_{x\in B((0,0), 0.5)} + x \cdot \frac{R}{\|x\|\_2} \cdot 1_{x\notin B((0,0), 0.5)} $, see Fig. (1a). We provide the learned IT maps for $w\in \\{1, \frac{3}{2}, 2, 32\\}$, see Fig. (1b-1e). The quantitative results show that with the increase of $w$ our IT maps become closer and closer to the  grund-truth ET one (Table 1).
>
> **(2) Isn't "Supp$(T\sharp\mathbb{P}) \subset$ Supp$(\mathbb{Q})$" relatively straightforward for image tasks?**
>
> We use the standard mathematical definition of the support (lines 39-40), i.e., the support of a non-negative mesure $\mu$ is a closed set consisting of all points $x \in X$ for which every open neighbourhood $A\ni x$ satisfies $\mu(A)>0$. Following the standard manifold hypothesis [1], for a real data distribution $\mathbb{Q}$ (e.g., distribution of images of faces) and $\mathcal{Y}=[-1,1]^{D}$, one may think of Supp$(\mathbb{Q})\subset [-1,1]^{D}$ as of a small-dimensional manifold occupying a tiny part of the ambient space $[-1,1]^{D}$. In general, this manifold is unknown and we observe only some random data samples lying on it. This is the reason why enforcing the constraint Supp($T\sharp \mathbb{P}$) $\subset$ Supp($\mathbb{Q}$) is tricky in practice.
>
> **(3) The application of IT or ET could be limited in image tasks due to limited diversity issue.**
>
> We discussed this issue as a limitation of our method in Appendix A (lines 503-509). However, we emphasize that it is a dataset-dependent problem which may appear only in specific dataset cases like in our *texture* to *chair* translation example, see Fig. 9A. For clarity, we additionally demonstrated it on a specially designed toy example, see Fig. 10A.
>
> **(4.1) The constraint in (11) is not a softened version of (5).**
>
> Indeed, 'softened version' may be not the best formulation in this case since the constraint $T\sharp\mathbb{P}\leq \mathbb{Q}$ is indeed stronger. However, the constraint $T_{\sharp}\mathbb{P} \leq \mathbb{Q}$ is more computationally feasible in practice and our answer to your question below explains why. *We will change the word 'soften' in section 3.2.*
>
> **(4.2) Why is it not feasible to design a dual formula for ET directly?**
>
> Most of the duality formulas in OT field are derived using the ideas of the Largrange multiplier method. It is applicable to equality contraints such as $T_{\sharp}\mathbb{P}=\mathbb{Q}$ or inequality constraints like $T_{\sharp}\mathbb{P}\leq w \mathbb{Q}$. It is not clear how to incorporate the set inclusion constraints such as $Supp(T\sharp\mathbb{P})\subset Supp(\mathbb{Q})$. This is why we need the transition from ET to IT.
>
> **(5) You mention fake solutions issue. Consider using ideas from the Kernel NOT paper.**
>
> We perform additional *Ball2Circle* experiment, see **the attached PDF file**, demonstrating the effect of using kernel cost function for alleviating fake solutions issue. We learn IT maps for $c(x, y)=\|x-y\|_2$ with or w/o regularization for weights $w\in\\{1,2,32\\}$.
>
> Without regularization, we observe that the method is unstable (for $w\in\{1,2\}$), see Fig. 2b-e. In contrast, with  kernel regularization (+stochastic map $T(x,z)$) [2] the method always converges, see Fig. 3.
>
> Our example shows (a) fake solutions may be a problem and (b) regularization may help to deal with them. Further studying this aspect is out of the scope of the paper.
>
> **(6) Under which conditions the method can recover the ET solution? How large does the value of w need to be?**
>
> If we correctly understand, you ask for a concrete convergence rate of IT plans $\pi^{w}$ to ET plans $\pi^{*}$ as a function of $w$. We do not specify the rate of convergence and leave this aspect for future studies (see lines 168-170).
>
> **(7) FID and metrics based on images itself.**
>
> We use the FID metric simply because there are no principally different alternatives. In fact, all the metrics for generative models which we have seen evaluate set-to-set similarity rather than the quality of individual samples. Moreover, in the *unpaired* translation task the evaluation based on individual images (e.g., paired metrics) seems to be not applicable.
>
> **(8) In Figure 24, some images fail to preserve hair colors. In Figure 22, some images exhibit unrealistic artifacts. Could this be a limitation?**
>
> We agree with the reviewer that in Figure 24, the hair colors are not preserved. However, we need to explain that it is the **expected behaviour** since we aim to solve the problem of *'mapping to the nearest neighbor in a target dataset'*. Thus, if this target dataset does not contain samples with the same hair color as in input image, our model is not intented and should not keep the color. Regarding artifacts in Figure 22, we note that they occur since the dataset is rather challenging.
>
> **(9) Figure 5b may not be as illustrative as intended. <...> Provide an example where (8) admits multiple minimizers, e.g. when support is not convex?**
>
> *We will replace the picture in the final version of our paper with the new one where $\mathbb{Q}$'s support is non-convex.*
>
> **Concluding remarks**. Please respond to our post to let us know if the clarifications above suitably address your concerns about our work. We are happy to address any remaining points during the discussion phase; if the responses above are sufficient, we kindly ask that you consider raising your score.
>
> **References.**
>
> [1] Fefferman, et al. (2016). Testing the manifold hypothesis.
>
> [2] Korotin, A. et al. (2022). Kernel neural optimal transport.

---

> > ### Comment · Area_Chair_YdJW · 2023-08-14
> >
> > Dear reviewer Mz3K, does the author's rebuttal address your concerns? In particular, can you comment on whether your concerns on formulation and method are addressed? Do you have further comments/questions for the authors?

---

> > ### Comment · Reviewer_Mz3K · 2023-08-14
> >
> > Thank you for your reply! I have some follow-up questions.
> >
> > (6) "We do not specify the rate of convergence and leave this aspect for future studies (see lines 168-170)."
> >
> > Can you discuss this even empirically? For example, for those face image style transfer and handbag <-> shoes datasets, what range is enough?
> >
> >
> > (7) FID and metrics based on images itself.
> >
> > Could you try to add the following baseline? For each image in the source dataset, determine its closest match in the target dataset using a 1-nearest-neighbor approach. This method will allow you to create a new dataset. Then, calculate the FID concerning this newly established dataset instead of the initial target dataset. I foresee a reduction in the FID value as $w$ increases.

---

> > > ### Author Response · Authors · 2023-08-16
> > > **Additional response**
> > >
> > > Dear Reviewer Mz3K, please find the answers to your follow-up questions below.
> > >
> > > **(1) Convergence rate: empirically, which values of $w$ are enough to make IT maps close to ET?**
> > >
> > > We may use the $\ell^2$ transport cost of learned IT maps to determine $w$ for which IT maps become close enough to ET. Indeed, after a certain value of $w$ it is expected that $\mbox{Cost}_{w}$ will stop rapidly changing. Hence, intuitively, one may expect that IT map is close enough to ET as well.
> > >
> > > For *celeba*$\rightarrow$*anime* case, the $\ell^{2}$ cost decreases rapidly for weights $w\in\\{1,2,4,8\\}$, see Table 1(a) in our paper. However, as we discussed in Appendix G.2, further increase of the weight (there we tested additional weights $w\in\\{16,32\\}$) does not lead to any significant cost decrease. Therefore, for this pair of datasets, $w=8$ can be considered as a sufficient value to get the IT map which is close enough to ET.
> > >
> > > In *handbag*$\rightarrow$*shoes* experiment, the difference in $\ell^2$ cost between $w=4$ and $w=8$ in Table 1(a) seems insignificant. Thus, in this case, for $w=4$ the IT map may be treated as close enough to some ET map.
> > >
> > > **(2) FID for the 1-nearest-neighbors.**
> > >
> > > To address the reviewer's question, we calculate FID values between our IT maps and 1-nearest-neighbors of input samples in target dataset (test parts) for *celeba*$\rightarrow$*anime* and *handbag*$\rightarrow$*shoes* experiments, see the Table below.
> > >
> > > |                                             | $w=1$ | $w=2$ | $w=4$ | $w=8$ |
> > > |---------------------------------------------|-------|-------|-------|-------|
> > > | *celeba*$\rightarrow$*anime*  | 53.21 | 44.81 | 39.77 | 43.03 |
> > > | *handbag*$\rightarrow$*shoes* | 73.35 | 68.31 | 73.44 | 80.61 |
> > >
> > > We see that there is no obvious dependence between the weight $w$ and calculated FID values.
> > > This is quite expected since the comparison with discrete nearest neighbors is irrelevant in our case. Indeed, in our paper we show that minimizer $T^*$ of ET problem (which we seek for when $w\rightarrow\infty$) maps each point $x\sim\mathbb{P}$ to its nearest neighbors *in the support* (Supp$(\mathbb{Q})$) of the target distribution, see lines 104-113. However, nearest neighbors in the **empirical** dataset are significantly **biased** over the desired nearest neighbors in the support.
> > >
> > > The fact that the **empirical nearest neighbors are a poor replacement of the true nearest neighbors** (in the support) was already indirectly illustrated in the Appendix D of our paper. There we regressed (distilled) a neural network $T_{\theta}$ to predict discrete (empirical) nearest neighbors in the train dataset. From Figure 18 we can make two valuable conclusions:
> > >
> > > - While the learned network generates high-quality images on the training dataset (Figure 18(b), $w=\infty$), it struggles to generalize well to the unseen test samples (Figure 18(a), $w=\infty$).
> > >
> > > - The empirical nearest neighbors may not have sufficient similarity with the input images. This can be seen from Figure 18(b) (the 1st and last lines). Despite the fact that the target empirical nearest neighbors are not shown there, our trained network ($w=\infty$) almost perfectly reproduces them (the train loss was almost zero in that experiment). Hence, the images in the last line can be viewed as the empirical nearest neighbors to the images on the first line.
> > >
> > > To conclude, FID values estimated by using the empirical replacement of nearest neighbors can not be considered as the representative metric.

---

> > > > ### Comment · Reviewer_Mz3K · 2023-08-17
> > > >
> > > > Thank you for your prompt response. I am willing to increase my score. However, upon reviewing the supplementary material, I've observed a few areas that might benefit from further refinement. Specifically, in Figure 23, columns five and six, it appears that as $w$ increases, the generated chair seems to deviate from the input pattern. Additionally, in certain columns, such as the penultimate and final ones, the generated chair doesn't seem to closely match the input pattern. Moving on to Figure 24, in the first column, the generated face at $w=8$ seems to lack the teeth detail, and in the fifth column, the representation at $w=1$ appears to be more in line with the input because it does not show the teeth. I believe there's potential for further enhancement, perhaps by integrating methods like the kernel neural OT.

---

> > > > > ### Author Response · Authors · 2023-08-18
> > > > > **Response to your official comment**
> > > > >
> > > > > Dear Reviewer, we appreciate your thorough attention to the details of our paper. Please find below the clarifications for the concerns which you raised.
> > > > >
> > > > > **(1a) Figure 23. In columns 5 and 6, as $w$ increases, the generated chair seems to deviate from the input pattern. In penultimate (11th) and final (12th) columns, the generated chair doesn't seem to closely match the input pattern.**
> > > > >
> > > > > **(1b) Figure 24. In the 1st column, the generated face at $w=8$ seems to lack the teeth detail. In the 5th column, the representation at $w=1$ appears to be more in line with the input because it does not show the teeth.**
> > > > >
> > > > > In order to address your concerns about the patterns and details, we additionally explain several points about our results.
> > > > >
> > > > > - Particular samples are not good representatives of the properties of the entire learned transport map $\hat{T}$. According to the average $\ell^2$ cost between the generated and respective input images, the **average input-output dissimilarity** $\frac{1}{N_{test}}\sum_{n=1}^{N_{test}}\ell^{2}(x_n,\hat{T}(x_n))$ indeed decreases for all the dataset pairs with the increase of the weight $w$, see Table 1 in our paper.
> > > > >
> > > > > - Moreover, we additionally calculate $\ell^2(x,\hat{T}(x))$ for the **particular face/texture samples** $x$ to which you pointed to. We see that their input-output dissimilarity also decreases with the increase of $w$ in majority of the cases, see the Table below.
> > > > >
> > > > > |                  | $w=1$ | $w=2$ | $w=4$ | $w=8$ |
> > > > > |------------------|-------|-------|-------|-------|
> > > > > | Fig. 23, col. 5  | 0.635 | 0.526 | 0.522 | 0.482 |
> > > > > | Fig. 23, col. 6  | 0.524 | 0.378 | 0.323 | 0.277 |
> > > > > | Fig. 23, col. 11 | 0.798 | 0.724 | 0.610 | 0.591 |
> > > > > | Fig. 23, col. 12 | 0.410 | 0.420 | 0.291 | 0.226 |
> > > > > | Fig.24, col. 1   | 0.258 | 0.239 | 0.213 | 0.198 |
> > > > > | Fig.24, col. 5   | 0.154 | 0.131 | 0.105 | 0.103 |
> > > > >
> > > > > - It is a generic knowledge that $\ell^2$ cost is mostly sensitive to low-frequency image components (big image parts) rather than high-frequency components (tiny details). For example, as your noted, in the 1st column of Figure 24, IT map for $w=8$ lacks the fine teeth detail. However, here we observe the presence of the long hair (like in the input image). Note that the hair was not generated for smaller weights. In the *textures*$\rightarrow$*chairs* experiment, the situation is also expainable. Here with the increase of $w$, the IT method with $\ell^{2}$ focuses more on the reducing the white background area. At the same time, preservation of the fine details of the texture may become less important. We have partially mentioned this aspect in lines 508-509 of Appendix A (limitations).
> > > > >
> > > > > To conclude, when using $\ell^{2}$ cost, it is not correct to assess similarity of the images focusing only on fine/tiny image details. In order to focus on such details, one may consider using more advanced cost functions, e.g., perceptual ones from computer vision. While this is a promising research avenue, it is beyond the scope of our paper which focuses on developing the theoretical formulation and the computational algorithm. Still to ensure that our IT algorithm may work with costs beside $\ell^{2}$, we have already tested it with a VGG-based perceptual cost, see Appendix G.1.
> > > > >
> > > > > **(2) Further enhancement, e.g., using kernel neural OT.**
> > > > >
> > > > > We agree that many potential modifications to our methodology can be considered. For example, apart from the advanced cost functions mentioned above, one indeed may employ kernel regularization [1]. We note that the goal of the current work is to establish the **basic fundament** to which such modifications can be applied. Studying of all possible modifications is a separate interesting topic which is out of the scope of our paper. At the same time, we have already provided evidence that such modifications are applicable and sometimes useful. In Appendix G.1, there are our experiments with a perceptual cost and in the attached PDF $-$ your requested experiment with kernel regularization.
> > > > >
> > > > > **Concluding remarks.** We hope that our clarifications have resolved your concerns regarding our method performance and potential enhancement. If the responses above are sufficient, we kindly ask that you consider raising your score.
> > > > >
> > > > > **References**
> > > > >
> > > > > [1] Korotin, A. et al. (2023). Kernel neural optimal transport.

---

### Official Review · Reviewer_iXGC · 2023-07-26

**Soundness:** 3 good
**Presentation:** 4 excellent
**Contribution:** 3 good
**Rating:** 6
**Confidence:** 3

**Summary:**

This paper proposes a novel notion of extremal transport, which relaxes the optimal transport problem by only requiring the support of the pushed-forward distribution to be a subset of the support of the target distribution. To solve this problem, the authors propose a novel approximation approach to find the solution from a subsequence of a series of incomplete OT problems (where T#P = Q is relaxed to T#P < wQ, w >= 1, when w=1 this is the traditional OT problem). Then the author derives novel dual formulations of the incomplete OT problems, which the author proposes to solve by using neural networks to approximate the potential and c-transform solutions. The authors apply the method to toy sets and image translation examples to demonstrate the scalability of the method. FID results on the image translation example prove the concept of the method.

The authors carefully compare their results with related works and establish their theoretic novelty. The presentation of the results are clear and potential impacts are well discussed.

**Strengths:**

The paper proves several strong theoretical results on the relaxed versions of OT problems (i.e. ET and IT). The theoretic study of the solutions to these problems is fruitful. This opens up a new domain of study area in the optimal transport area.

**Weaknesses:**

Overall the theoretic results are rigorous and novel. To solve the solution of the dual formulation of IT, the authors propose to use neural networks to approximate the transport map (necessary for solving large-scale problems). One concern I have is that in the traditional OT problem, the OT map is in general discontinuous. This proposes difficulties in using DNN to approximate the OT map. It would be very welcomed if results on the regularity of the IT maps are presented, and/or how well the neural network can approximate such maps.

**Questions:**

One question: in theorem 3, it only guarantees the existence of a subsequence. In practice, if we found a sequence of solutions by Alg. 1, how can we make sure whether it is the desired ET solution?

**Limitations:**

The limitation of the paper is adequately addressed by the authors.

---

> ### Author Rebuttal · Authors · 2023-08-09
>
> Dear Reviewer, thank you for your comments. Here are the answers to your questions.
>
> **(1) In the traditional OT problem, the OT map is in general discontinuous. This proposes difficulties in using DNN to approximate the OT map. It would be very welcomed if results on the regularity of the IT maps are presented, and/or how well the neural network can approximate such maps.**
>
> **Regularity.** In the proof of Proposition 3 (Appendix F, line 772), we derive an auxiliary statement that if $\pi^* \in\Pi^w(\mathbb{P},\mathbb{Q})$ is an IT plan between $\mathbb{P}$ and $\mathbb{Q}$, then it is an OT plan between $\mathbb{P}$ and $\pi_{y}^{\*}$. This also leads to the fact that IT maps $T^{*}$ between $\mathbb{P}$ and $\mathbb{Q}$ are the OT maps between $\mathbb{P}$ and $T^{\*}_{\sharp}\mathbb{P}$. Hence, it seems like the results on their regularity may be potentially derived from the general regularity properties of OT maps, see [1], but this requires further studies.
>
> **DNN approximation.** Fortunately, from the practical point of view, we can show that IT maps **can** be approximated with neural networks. As it is shown in [2, Theorem 1], assuming that the target distribution has finite second moment, neural networks can arbitrary well approximate the OT map (and hence our IT map) w.r.t. the $\mathcal{L}^2(\mathbb{P})$ norm. The authors provided a concise proof for stochastic OT maps but there are no significant changes when we turn to the deterministic case.
>
> To conclude, neural networks can arbitrarily well approximate IT maps.
>
> **(2) In theorem 3, it only guarantees the existence of a subsequence. In practice, if we found a sequence of solutions by Alg. 1, how can we make sure whether it is the desired ET solution?**
>
> Indeed, different sub-sequences might converge to different ET plans and whole sequence might be non-converging itself. To address your question, we provide a short corollary from our Theorem 3 showing that with the increase of weight $w$, elements of any such sequence become closer to **set** of ET plans.
>
> *Corollary.* $\forall\varepsilon>0$ $ \exists w(\varepsilon)\in[1,\infty)$ such that $\forall w \geq w(\varepsilon)$ and $\forall$ IT plan $\pi^w\in\Pi_w(\mathbb{P}, \mathbb{Q})$ solving Kantorovich's IT problem (equation (12) in our paper), there exists a ET plan $\pi^\*$ which is $\varepsilon$-close to $\pi^w$ in $\mathbb{W}_1$, i.e., $\mathbb{W}_1(\pi^\*, \pi^w)\leq \varepsilon$.
>
> *Proof.* Assume the inverse. Then $\exists \varepsilon$ such that $\forall w(\varepsilon)$ $\exists w \geq w(\varepsilon)$ and $\exists$ IT plan $\pi^w\in \Pi_w(\mathbb{P}, \mathbb{Q})$ solving (12) such that $\forall$ ET plan $\pi^\*$, it holds that $\mathbb{W}\_1 (\pi^w, \pi^\*)\geq \varepsilon$. Pick a sequence $w_1, w_2, ..., w_n \rightarrow \infty$ and the corresponding sequence of IT plans $\pi^{w_1}, \pi^{w_2}, ..., \pi^{w_n}$.  From Theorem 3, it has a sub-sequence (weakly-*) converging to some ET plan $\pi^\*$: $\pi^{w^{n_k}}\rightarrow \pi^\*$. However, $\forall n_k$ $\exists w \geq w_{n_k}$, such that $\mathbb{W}\_1(\pi^{w^{n_k}}, \pi^{\*}) \geq \mathbb{W}\_1(\pi^{w}, \pi^{\*}) \geq \varepsilon$. Hence, the sub-sequence does not converge to $\pi^{\*}$ in $\mathbb{W}\_{1}$.
> Recall that the convergence in $\mathbb{W}\_{1}$ coicides with the weak-$\*$ convergence (for compact $\mathcal{X},\mathcal{Y}$), see [3, Theorem 5.9]. Hence, the subsequence also does not weakly-* converge to $\pi^{\*}$ which is a contradiction. $\square$
>
>
> **Concluding remarks**. Please respond to our post to let us know if the clarifications above suitably address your concerns about our work. We are happy to address any remaining points during the discussion phase; if the responses above are sufficient, we kindly ask that you consider raising your score.
>
> **References.**
>
> [1] De Philippis, G., & Figalli, A. (2015). Partial regularity for optimal transport maps. Publications mathématiques de l'IHÉS, 121(1), 81-112.
>
> [2] Korotin, A., Selikhanovych, D., & Burnaev, E. (2022, September). Neural Optimal Transport. In The Eleventh International Conference on Learning Representations.
>
> [3] Santambrogio, F. (2015). Optimal transport for applied mathematicians. Birkäuser, NY, 55(58-63), 94.

---

### Author Rebuttal · Authors · 2023-08-09

Dear reviewers,

thank you for your thorough and detailed feedback! We are highly inspired by the fact that you agree on the novelty of the proposed Extremal and Incomplete Transport (ET/IT) formulations (Reviewers iXGC, URfJ, Cqh5, 3fqr), find our theoretical results to be valuable (Reviewer iXGC, 3fqr), acknowledge that our algorithm is widely applicable (Reviewer URfJ), and its effectiveness is proved by complete and sufficient experimental evaluation (Reviewers URfJ, Cqh5). We are glad that you positively highlight clear presentation and comprehensiveness (Reviewers iXGC, 3fqr, Cqh5) of our paper. We hope that our IT algorithm would be easy to use in practical applications.

We will incorporate the changes suggested by the reviewers in the final version of our paper. We list the changes below:

(a) Main text (**minors**) $-$ replacement of the Figures (3-5) by those where the support is non-convex (Reviewers Mz3K, 3fqr) plus minor requested clarifications here-and-there;

(b) New **Appendix** section$-$ additional experiments (Reviewer Mz3K): *Swiss2Ball* experiment where ET maps have *closed-form*, *Ball2Circle* experiment testing advanced version of our method with weak kernel cost;

(c) **Addition** to Appendix F $-$ small corollary showing the closeness of IT problem solution to the set of ET plans (Reviewers iXGC).

Please find Figures for experiments requested by reviewer Mz3K in the **attached PDF file**.

Please find the answers to your questions below.

---

> ### Comment · Area_Chair_YdJW · 2023-08-18
> **Rebuttal Acknowledgment**
>
> Dear authors,
>
> Thank you for your efforts in writing a rebuttal. Unfortunately not all reviewers have acknowledged or responded to it, but rest assured that I have read it and will bring it up in further discussion with the reviewers, and will take it into account for the final recommendation.
>
> Best,
>
> AC

---

### Decision · Program_Chairs · 2023-09-21

**Decision:**

Accept (poster)

**Comment:**

This paper studies generalized versions of the optimal transport problem where the usual mass conservation marginal constraint is replaced by weaker / relaxed constraints. The reviewers in general appreciated the formulation, theoretical rigor, and strong empirical results.

But the paper has important weaknesses too. For example, I tend to agree with reviewer Mz3K that the way the paper is presented puts too much emphasis on a formulation of the problem (the "Extremal Transport" problem, Eq 5.) that is not properly motivated and is not ultimately what is solved. In particular, it is not clear why one would be interested in a map that has absolutely no mass preservation constraints (only partial support overlap). E.g., any map that puts zero mass on the complement of the support of Q but is otherwise unconstrained is feasible for Eq. 5 -- this is an extremely large (and frankly, not too interesting) class of maps! In contrast, the "Incomplete Transport" problem (Eq. 11), which the authors settle for almost begrudgingly as an "easier alternative" to (5) seems much more natural and easier to motivate, since it optimizes over a much more interesting class of maps. Confusingly, they refer to the transition from (5) to (11) as a "relaxation" (or "softening"), even though it's the exact opposite -- as is clear from the example above, the constraint set of (5) is much larger than that of (11). This to me muddies the message of the paper and makes the ET formulation feel almost forced and superfluous.

Another aspect where the paper falls short is in contextualizing these formulations in the context of Partial Optimal Transport, a very related and well-studied variant of OT, of which the formulation proposed here is a particular case. While there is a discussion on the related POT work in the Appendix, and sporadic mentions throughout the main text, this connection should be made earlier than Page 5 -- for the sake of the reader's understanding and contextualization of the contributions of this paper.

All things taken into account, I believe this is a solid paper that has not lived up to its full potential due to some important issues of presentation and narrative, but which is nevertheless (and in spite of its weaknesses) likely to be of interest to the ML+OT community in NeurIPS. For that reason, I am ultimately recommending acceptance, but I sincerely hope the authors take the combined feedback from reviews and metareviews to clean up the paper's presentation.